



# Improved SWAT vegetation growth module for tropical ecosystem

Tadesse Alemayehu[1*], Ann van Griensven[1,2] and Willy Bauwens[1]

[1]Vrije Universiteit Brussel (VUB), Department of Hydrology and Hydraulic Engineering, Brussel, Belgium
[2]UNESCO-IHE, Department of Water Science and Engineering, Delft, The Netherlands

*Correspondence: tabitew@vub.ac.be; Tel.: +32-488979027

**Abstract.** The Soil and Water Assessment Tool (SWAT) is a globally applied river basin eco-hydrological simulator in a wide spectrum of studies, ranging from land use change and climate change impacts studies to research for the development of best water management practices. However, SWAT has limitations in simulating the seasonal growth cycles for trees and perennial vegetation in tropics, where the major plant growth controlling factor is the rainfall (via soil moisture) rather than temperature. Our goal is to improve the vegetation growth module of the SWAT model for simulating the vegetation parameters such as the leaf area index (LAI) for tropics. Therefore, we present a modified SWAT version for the tropics (SWAT-T) that uses of a simple but robust soil moisture index (SMI) - a quotient of the rainfall (P) and reference evapotranspiration (PET) - to initiate a new growing season after a defined dry season. Our results for the Mara Basin (Kenya/Tanzania) show that the SWAT-T simulated LAI corresponds well with the Moderate Resolution Imaging Spectroradiometer (MODIS) LAI for evergreen forest, savanna grassland and shrubs, indicating that the SMI is a reliable proxy to dynamically initiate a new growing cycle. The water balance components (evapotranspiration and flow) simulated by the SWAT-T exhibit a good agreement with remote sensing-based evapotranspiration (RS-ET) and observed flow. The SWAT-T simulator with the proposed improved vegetation growth module for tropical ecosystem could be a robust tool for several applications including land use and climate change impact studies.

## 1. Introduction

The Soil and Water Assessment Tool (SWAT; Arnold et al., 1998) is a process-oriented, spatially semi-distributed and time-continuous river basin simulator. SWAT is one of the most widely applied eco-hydrological simulators for simulating hydrological and biophysical processes under a range of climate and management conditions (Arnold et al., 2012; Bressiani et al., 2015; Gassman et al., 2014; van Griensven et al., 2012; Krysanova and White, 2015). Many studies used SWAT in tropical Africa, to investigate the basin hydrology (e.g. Dessu and Melesse, 2012; Easton et al., 2010; Mwangi et al., 2016; Setegn et al., 2009) as well as to study the hydrological impacts of land use change (e.g. Gebremicael et al., 2013; Githui et al., 2009; Mango et al., 2011) and climate change (Mango et al., 2011; Mengistu and Sorteberg, 2012; Setegn et al., 2011; Teklesadik et al., 2017). Notwithstanding the high number of SWAT model applications in tropical catchments, only a few studies underscored the limitation of its plant





growth module for simulating the growth cycles of trees, perennials and annuals in this region of the world (Mwangi
et al., 2016; Strauch and Volk, 2013; Wagner et al., 2011).
It is worthwhile to note that phenological changes in the vegetation affect the biophysical and hydrological process-
es in the basin hydrology and thus play a key role in integrated hydrologic and ecosystem modeling (Jolly and
Running, 2004; Shen et al., 2013; Strauch and Volk, 2013; Yang and Zhang, 2016; Yu et al., 2016). The Leaf Area
Index (LAI), a vegetation variable commonly used in hydrological modeling, strongly correlates with the phenologi-
cal development. Thus, an improved representation of this variable may improve the predictive capability of hydro-
logic models, as noted in several studies (Andersen et al., 2002; Yu et al., 2016; Zhang et al., 2009). Arnold *et al.*
(2012) underscored the need for a realistic representation of the local and regional plant growth processes  in SWAT
due to its effect on the water balance, on the erosion, and on the nutrient yields.
SWAT utilizes a simplified version of the Environmental Policy Impact Climate (EPIC) crop growth module to
simulate the phenological development of plants, based on accumulated heat units (Arnold et al., 1998; Neitsch et
al., 2011). It uses dormancy, a function of daylength and latitude, to repeat the annual growth cycle for trees and
perennials. Admittedly, this approach is suitable for temperate climate zones. However, Strauch and Volk (2013)
showed that the LAI temporal dynamics are not well represented for perennial vegetation (savanna and shrubs) and
evergreen forest in Brazil. Likewise, Wagner et al. (2011) reported a shift in the growth cycle of deciduous forest in
the Western Ghats  (India).
Unlike temperate regions where the vegetation growth dynamics are mainly controlled by the temperature, the pri-
mary controlling factor in tropical regions is the rainfall (i.e. the water availability)  (Jolly and Running, 2004;
Lotsch, 2003; Pfeifer et al., 2012, 2014; Zhang, 2005). A study of Zhang et al. (2005) explored the relationship be-
tween the rainfall seasonality and the vegetation phenology across Africa. They showed that the onset of the vegeta-
tion green-up can be predicted using the cumulative rainfall as a criterion to indicate the season change. Jolly and
Running (2004) determined the timing of leaf flush in an ecosystem process simulator (BIOME-BGC) after a de-
fined dry season in the Kalahari , using events where the daily rainfall (P) exceeded the reference evapotranspiration
(PET). They showed that the modeled leaf flush date compared well with the leaf flush dates estimated from the
Normalized Vegetation Index (NDVI), indicating that precipitation and PET are good proxy's to pinpoint the season
change for tropical ecosystems. Strauch and Volk (2013) used SWAT simulated soil moisture in the top soil layers
with a certain minimum threshold to indicate the start of rainy season (SOS) and thus new vegetation growth cycle
after a defined dry season. Their results showed improvements in the SWAT simulated LAI seasonal dynamics and
reproduced well the Moderate Resolution Imaging Spectroradiometer (MODIS) LAI. However, their approach re-
quires calibrating the SWAT parameters for a realistic representation of the soil water balance dynamics often using
observed streamflow. Recently, Yu *et al.* (2016)  concluded uncertainty in soil moisture is significantly greater than
streamflow simulations of a calibrated hydrologic model.





Sacks et al. (2010) studied the relationships between crop planting dates and temperature, P and PET, using 30-year
average climatological values. They noted that in rainfall limited regions the ratio of P to PET is a better proxy for
the soil moisture status than is P alone. Therefore, we employ a simple soil moisture index (SMI) that is based on the
major inputs of SWAT (P and PET) for indicating the SOS. The SMI is determined using a quotient of a 5-day (pen-
tad) P to PET. A major advantage of this approach is the fact that the SMI is known *a priori* and so are the SOS and
the associated start of new vegetation growth cycle.
Remotely sensed information provides crucial information about the dynamics of vegetation (Adole et al., 2016;
Bobée et al., 2012; Zhang, 2005; Zhang et al., 2006). Zhang et al. (2006) produced global maps at 1-km spatial reso-
lution of key phonological metrics -such as the start of the growing season- using MODIS. They reported a good
correspondence of the retrieved phenological metrics with in situ measurements. Also, Bobée *et al*. (2012) found a
good match between the start and the end of the growth season as derived from remote sensing with ground-truth
observations for Sahelian vegetation. Therefore, we use remote sensing-based LAI products to explore the seasonal
LAI dynamics and evaluate the LAI simulated by the modified SWAT model.
In summary, this paper presents a methodology to improve the temporal dynamics of SWAT simulated LAI. The
performance of the modified SWAT simulator to simulate LAI and evapotranspiration (ET) will be evaluated using
MODIS LAI timeseries and remote sensing-based ET while the flow simulation skill will be evaluated using ob-
served flow.
**2. Materials and methods**
**2.1. The study area**
The Mara River, a transboundary river shared by Kenya and Tanzania, drains an area of 13,750 km$^2$ (Figure 1a).
This river originates from the forested Mau Escarpment (about 3000 m.a.s.l.) and meander through diverse agroeco-
systems and subsequently crosses the Masai-Mara Game Reserve in Kenya and the Seregenti National Park in Tan-
zania and finally feeds the Lake Victoria. The Amala River and the Nyangores River are the only perennial tributar-
ies draining the head water region. The Talek River and the Sand River are the two most notable seasonal rivers
stemming from Loita Hills.
Rainfall is highly variable in the Mara Basin. This is mainly due to its equatorial location and its topography. The
rainfall pattern in most part of the basin is bimodal, with a short rainy season (October-December) driven by con-
vergence and southward migration of the Intertropical Convergence Zone (ITCZ) and long rainy season (March-
May) driven by southeasterly trades. In general, rainfall decreases from west to east across while temperature in-
creases southwards in the basin. The Mara basin is endowed with significant biodiversity features through a se-
quence of zones from moist montane forest on the escarpment through dry upland forest to scattered woodland and
then the extensive savanna grasslands (Figure 1b). Dark volcanic origin soils are common on the escarpment and
rangelands while shallow soils that drain freely are found lower down. Poorly drained soils cover the plateau.





**2.2. A brief overview of SWAT**
The SWAT (Arnold et al., 1998, 2012; Neitsch et al., 2011) is a comprehensive, process-oriented, semi-distributed
and physically-based eco-hydrological simulator at a river basin scale. The major components include weather,
hydrology, soil temperature and properties, plant growth, nutrients, pesticides, bacteria and pathogens, and land
management.
SWAT uses a GIS based interface that allows using spatial information such as a digital elevation model (DEM) and
land use/land cover and soil maps. In SWAT a basin is partitioned into several sub-basins using topographic infor-
mation and the sub-basins, in turn, are subdivided into several Hydrological Response Units (HRUs) with a unique
combination of land use, soil and slope class. Each hydrologic process are simulated at HRU level on a daily or sub-
daily time step and aggregated into sub-basin level for routing into a river network (Neitsch et al. 2011). SWAT
considers five storages: snow, canopy storage, the soil profile with up to ten layers, a shallow aquifer and a deep
aquifer to calculate the water balance (Neitsch et al., 2011) using the following equation:

$$\Delta S = \sum_{t=1}^{t} \left( P - Q_{total} - ET - Losses \right) \tag{1}$$

where $\Delta S$ is the change in water storage and $t$ is time in days. $P, Q_{total,} ET$ and $Losses$ are the daily amounts of precip-
itation, the total water yield, the evapotranspiration and the groundwater losses, respectively. The total water yield
represents an aggregated sum of the surface runoff, the lateral flow and the return flow. In this study, the surface
runoff is computed using the SCS curve number procedure (USDA SCS, 1972). SWAT simulates ET i based on the
PET from soil and plants  as described in Ritchie (1972). Therefore, the aggregated ET refers the sum of evaporation
from the canopy and the soil as well as plant transpiration. The reader is referred to Alemayehu *et al.* (2015) and
Neitsch *et al.* (2011) for the PET formulations in SWAT. PET is calculated using the Penman-Monteith equation.
**2.2.1.  The vegetation growth and Leaf Area Index modeling in SWAT**
SWAT simulates the annual vegetation growth based on the simplified version of the EPIC plant growth model
(Neitsch et al., 2011). The potential plant phenological development is hereby simulated on the basis of daily accu-
mulated heat units under optimal conditions; however, the actual growth is constrained by temperature, water, nitro-
gen or phosphorous stress. The potential biomass production is based on Monteith's approach while the yield is
computed using a harvest index (Arnold et al., 2012; Neitsch et al., 2011).
Plant growth is primarily based on temperature and hence each plant has its own temperature requirements (i.e.
minimum, maximum and optimum). Plant growth is maintained while the daily mean temperature exceeded and/or
equalled the base temperature with a  rate of growth directly proportional to heat unit (HU) accumulation.HU is
computed as:



$$HU = T_m - T_{base} \text{ when } T_m > T_{base} \tag{2}$$

Where $T_m$ is the mean daily temperature ($^0$C) and $T_{base}$ is the plant's minimum temperature for growth ($^0$C).
The fundamental assumption in heat unit theory is that plants have a heat unit requirements that can be quantified
and linked to the time of planting to maturity (Neitsch et al., 2011). The total number of heat units required for a
plant to reach maturity must be provided by the user that is calculated:

$$PHU = \sum_{d=1}^{n} HU \tag{3}$$

where $PHU$ is the total heat units required for a plant maturity (heat units), $HU$ is the number of heat units accumu-
lated on day $d$ where $d=1$ on the day of planting and $n$ is the number of days required for a plant to reach maturity.
For trees and perennials, the time that the plants begin to develop buds and seeds reach maturation are considered as
the beginning and endo of the growing season. The fraction of PHU ($fr_{PHU}$) accumulated on a given date is calculat-
ed:

$$fr_{PHU} = \frac{\sum_{i=1}^{d} HU_i}{PHU} \tag{4}$$

The plant growth modeling in SWAT includes simulation of the leaf area development, light interception and con-
version of intercepted light into biomass assuming a plant species-specific radiation-use efficiency (Neitsch et al.,
2011). The optimal leaf area development during the initial period of the growth is modeled as:

$$fr_{LA\text{Im}x} = \frac{fr_{PHU}}{fr_{PHU} + \exp(l_1 - l_2 \cdot fr_{PHU})} \tag{5}$$

where $fr_{LAImx}$ is the fraction of the plant's maximum leaf area index corresponding to a given fraction of potential
heat units for the plant, , and $l_1$ and $l_2$ are shape coefficients. Once the maximum leaf area index is reached, LAI will
remain constant until the leaf senescence begins to exceed the leaf growth. Afterwards, the leaf senescence becomes
the dominant growth process and hence the LAI follows a linear decline (Neitsch et al., 2011). However, Strauch
and Volk (2013) showed the advantage of using a logistic decline curve, to avoid that the LAI drops to zero before
dormancy occurs. Therefore, we adopted this change to SWAT2012 whereby the LAI during leaf senescence for
perennials is calculated as:

$$LAI = \frac{LAI_{mx} - LAI_{\min}}{1 + \exp(-t)} \tag{6}$$





$$with \quad t = 12(r - 0.5) \quad and \quad r = \frac{1 - fr_{PHU}}{1 - fr_{PHU,sen}} \quad , fr_{PHU} \geq fr_{PHU,sen}$$

where the term used as exponent is a function of time and t varies from 6 to -6, $LAI$ is the leaf area for a given day
and declines using r as a decline rate, $LAI_{mx}$ and $LAI_{min}$ are the maximum and minimum (i.e. during dormancy) leaf
area index, respectively, $fr_{PHU,sen}$ is the fraction of growing season (PHU) at which senescence becomes the domi-
nant growth process.
As detailed in Neitsch *et al.* (2011), the daily LAI calculation for perennials and trees are slightly different.
For perennials, the leaf on day *i* is calculated as:

$$\Delta LAI_i = \left( fr_{LAImx,i} - fr_{LAImx,i-1} \right) LAI_{mx} .$$
$$\left( 1 - \exp(5.(LAI_{i-1} - LAI_{mx})) \right) \tag{7}$$


While for trees, the leaf area added on day *i* is calculated:

$$\Delta LAI_i = \left( fr_{LAImx,i} - fr_{LAImx,i-1} \right) \left( \frac{yr_{cur}}{yr_{fulldev}} \right) . LAI_{mx} . \tag{8}$$
$$\left( 1 - \exp \left( 5. \left( LAI_{i-1} - \left( \frac{yr_{cur}}{yr_{fulldev}} \right) . LAI_{mx} \right) \right) \right)$$

The total leaf area index is calculated:

$$LAI_i = LAI_{i-1} + \Delta LAI_i \tag{9}$$

where $\Delta LAI_i$ is the leaf area added on day *i*, $LAI_i$ and $LAI_{i-1}$ are the leaf area indices for day *i* and *i-1* respectively,
$fr_{LAImx,i}$ and $fr_{LAImx,i-1}$ are the fraction of the plant's maximum leaf area index for day *i* and *i-1*, $LAI_{mx}$ is the maximum
leaf area index for the plants, $yr_{cur}$ is the age of the tree (years), and $yr_{fulldev}$ is the number of years for tree species to
reach full development (years).
**2.2.2. The annual vegetation growth cycle in SWAT and its limitation for the tropics**
SWAT assumes that trees and perennial vegetation can go dormant as the daylength nears the minimum daylength
for the year. Dormancy, a function of latitude and daylength, during which plants do not grow, is used to repeat the





growth cycle each year for trees and perennials. At the beginning of the dormant period, a fraction of the biomass is
converted to residue and the leaf area index is set to the minimum value. Both the fraction of the biomass converted
to residue and the minimum LAI are defined in the plant growth database (Neitsch et al., 2011). Temperature is the
main controlling factor for vegetation growth in temperate region and thus, the dormancy strategy suitable as a
proxy for initiating new growth cycle annually.  In the tropics, however, plants growth dormancy is primarily con-
trolled by precipitation (Bobée et al., 2012; Jolly and Running, 2004; Lotsch, 2003; Zhang et al., 2010; Zhang, 2005)
and hence the standard SWAT growth module cannot realistically represent the seasonal growth dynamics for trees
and perennials. In fact, to address this limitation, SWAT offers several management operations to improve the sea-
sonality of trees and the perennial growth cycle using either heat units (the default) or calendar date scheduling. The
default management operation in SWAT is scheduled using heat unit fractions, whereby planting (start of growing
season) and kill (end of growing season) operations occur at $FR_{PHU}$ values of 0.15 and 1.2, respectively.

### 174    2.3. A soil moisture index-based vegetation growth cycle for the tropics

Several studties have demonstrated water availability in the soil profile is one of the primary governing factors of
vegetation growth in tropics (Jolly and Running, 2004; Lyamchai et al., 1997; Sacks et al., 2010; Strauch and Volk,
2013; Zhang, 2005; Zhang et al., 2006). The moisture availability (i.e. linked to rainfall) is therefore a realistic proxy
to pinpoint the onset of the new growing season for forest and perennials as noted in Strauch and Volk (2013) as
well. Nonetheless, the soil moisture estimates are not readily available from measurements, while model estimates
of the moisture are also not known *a priori*. Additionally, Yu *et al.* (2016) observed a higher uncertainty in soil
moisture simulations than in streamflow simulations. Thus, a simple SMI based on the major inputs of SWAT such
as P and PET could be a viable alternative. Figure 2 presents the SMI pattern for stations across the Mara Basin
using long-term climatological P and PET. It is apparent from Figure 2 that the dry season (mostly from June - Sep-
tember) shows low SMI values (less than 0.5). Additionally, these patterns resemble well the long-term monthly
average LAI for the savanna ecosystem (the dominant cover in the mid-section of the Mara Basin). In areas with a
humid climate (i.e. the head water regions of the basin), the SMI values are high and the rainfall regime is different,
yet in the relatively drier months (January and February) the SMI is low. Therefore, we suggest to use the SMI as a
proxy for the SOS and hence to reset the annual vegetation growth cycle. This approach enables SWAT to simulate
the vegetation cycle dynamically without the need for management setting ("plant" and "kill").
To avoid false starts during the dry season, the end of the dry season and the beginning of the rainy season ($SOS_1$
and $SOS_2$, respectively) are determined using a long-term monthly climatological P to PET ratio (Figure 2). For
river basins with a single rainfall regime, a single set of SOS months can be used across the basin. However, in ba-
sins with different rainfall regimes, different SOS months need to be set at sub-basin level. In our study area two
distinct rainfall regimes are observed and therefore two different SOS values were needed. For the major part of the
sub-basins October ($SOS_1$) and November ($SOS_2$) were used as transitions (Figure 2). Additionally, we used the
pentad ratio instead of a single day ratio, to assure the availability of sufficiently high soil moisture content for the
start of a new vegetation growth cycle.





### 2.4. SWAT-T: the adaptation of the SWAT plant growth module

Based on the rationale elaborated in the preceding sections, we modified the standard SWAT2012 (version 627) plant growth subroutine for basins located between $20^0$ N and $20^0$ S:

i)   If the simulation day is within $SOS_1$ and $SOS_2$ for a given HRU and a new growing cycle is not initiated yet, the SMI is calculated as the ratio of the pentad P to PET;

ii)  If the SMI exceeds or equals 0.5, a new growing cycle for trees and perennials is initiated. Subsequently, $FR_{PHU}$ is set to 0 and the LAI is set to the minimum value (ALAI_MIN). Plant residue decomposition and nutrient release is calculated as if dormancy would occur.

iii) In case the SMI is still below the threshold (i.e. 0.5) at the end of month $SOS_2$, a new growing cycle is initiated immediately after the last date of $SOS_2$.

It is worth noting that SMI threshold could be raised or lowered depending on the climatic condition of the basin.

### 2.5. Model set up, calibration and evaluation

### 2.5.1. The model set up and data used

The Mara River Basin was delineated using a high resolution (30 m) digital elevation model (DEM) in ArcSWAT2012 (revision 627). The basin was subdivided into 89 sub-basins to spatially differentiate areas of the basin dominated by different land use and or soil with dissimilar impact on hydrology. Each sub-basin was further discretized into several HRUs, which represent unique combinations of soil, land use and slope classes. The model was set up for conditions representing the period 2002-2009. The land cover classes for the basin were obtained from FAO-Africover project (FAO, 2002). Generally speaking, as shown in Figure 1b, the dominant portion of the basin is covered by natural vegetation including savanna grassland (RNGE), shrubland (RNGB) and evergreen forest (FRSE). We extracted the soil classes for the basin from the Harmonized Global Soil Database (FAO, 2008). A soil properties database for the Mara River Basin was established using the soil water characteristics tool (SPAW, http://hydrolab.arsusda.gov/soilwater).

Table 1 presents the list of hydro-climatological and spatial data used to derive, calibrate and evaluate the SWAT model. In situ measurements of rainfall and other climate variables are sparse and thus bias-corrected TMPA satellite rainfall data (Roy et al., 2017) were used. The bias-correction involves using historical gauge measurements and a downscaling to a 5 km resolution. Detailed information on the bias-correction and downscaling procedures can be found in Roy *et al.* (2017). Weather data needed to compute the PET was obtained from the Global Land Data Assimilation System (GLDAS) (Rodell et al., 2004). To improve the consistency of the PET estimates we adjusted at sub-basin level the solar radiation on average by 1.4% based on a method suggested in Alemayehu *et al.* (2017).



**2.5.2. Data for model evaluation**
*The Leaf Area Index*
The MOD15A2 LAI data used in this work is based on the MODIS TERRA sensor (version 5). This 8-day compo-
site product is provided at a 1 km$^2$ spatial resolution. The theoretical basis of the MODIS LAI product algorithm and
the validation results are detailed in Myneni et al. (2002). The LAI product is based on biome-specific algorithms,
involving several constants (leaf angle distribution, optical properties of soils and wood, and canopy heterogeneity)
(Myneni et al., 2002). Kraus (2008) reported a fair agreement of MOD15A2 LAI data with field measurements for
two East African forest biomes.
To reduce the effect of land cover mix on the LAI magnitude, we selected a representative homogenous sample sites
for evergreen forest, tea, savanna grassland and shrub land cover classes (see Figure 1b) using the Africover classes
and Google Earth images. Subsequently, the MOD15A2 LAI was masked using polygons of the sample covers. To
minimize the impacts of clouds, we used pixels with quality flag 0 as well as removed pixels with LAI values less
than 1.5 during the peak growing season. In the presence of gaps in the LAI time series, gaps were filled using linear
interpolation. Subsequently, we extracted the 8-day median LAI time series for each land cover for 2002-2009. Due
to the high frequency variability and the inevitable signal noise, the progression of LAI development from the start
of the growing season to the dry season are often influenced by sudden breaks. Verbesselt et al. (2010) developed
the Breaks For Additive Seasonal and Trend (BFAST) method that decomposes the Normalized Vegetation Index
(NDVI) time series into trend, seasonal, and remainder components. The trend and seasonal components comprise
information pertinent to phenological developments as well as gradual and abrupt changes whereas the reminder
component comprises noise and error information of the time series. This method has been applied to tropical eco-
systems to identify phenological cycles as well as abrupt changes (DeVries et al., 2015; Verbesselt et al., 2010,
2012). In our study, we used the BFAST tool to extract the seasonal development pattern of LAI while excluding the
noise and error information from the LAI time series. Figure 3 demonstrates the smoothed 8-day LAI time series
using BFAST along with the raw-median LAI values. It is apparent from the smoothed LAI time series that the
intra-annual variation of the LAI development is consistent with the seasonal rainfall pattern. Therefore, the
smoothed LAI time series  were used in two ways: i) to calibrate and evaluate the SWAT-T model for simulating the
LAI ii) incorporating the 8 years average of the first week (8-day) LAI for each month (i.e. prescription) to initialize
the LAI in SWAT-TRS month-by-month.
*The evapotranspiration*
ET is one of the major components in a basin water balance and closely linked with land cover classes and their
growth cycle. Thus, remote sensing-based ET estimates can be used to evaluate (calibrate) the SWAT-T model.
Alemayehu et al. (2017) estimated ET for the Mara River basin using several MODIS thermal imageries and the
GLDAS global weather dataset from 2002 to 2009 at a 8-day temporal resolution based on the Simplified Surface
Energy Balance operational (SSEBop) algorithm (Senay et al., 2013). The SSEBop mainly depends on the remotely



sensed land surface temperature and the grass reference evapotranspiration (Senay et al., 2013). Alemayehu et al.
(2017) demonstrated that the SSEBop ET explained about 52%, 63% and 81% of the observed variability in the
NDVI at 16-day, monthly and annual temporal resolution. Also, they suggested that the estimated ET can be used
for hydrological model parameterization. We note the resemblance in the seasonal pattern of the MODIS LAI ana-
lyzed in this study with the SSEBop ET, hereafter referred as remote sensing-based ET (RS-ET). Therefore, we used
this dataset to evaluate the SWAT simulated ET at land cover level.
*Flow*
Due to the limited availability of observed flow, the SWAT model was calibrated (2002 - 2005) and validated (2006
- 2008) for the head water region only, using daily flow. The selected periods for the calibration and validation peri-
od have about 11% missing gaps.
**2.6. The model performance metrics**
The main purpose of this study is to explore the potential of the SMI as a proxy to repeat the annual vegetation
growth cycle for the tropical ecosystem. The parameters related to the simulation of the LAI, the ET and the flow
are calibrated manually by trial-and-error and expert knowledge. Both the Pearson correlation coefficient (r) and the
Percent of PBIAS (%bias) were used to evaluate the agreement between the simulated and the remote sensing-based
estimates of LAI and ET for each land cover class and the flow at the Bomet gauge station. Additionally, the models
performance was evaluated using the Kling-Gupta Efficiency (KGE) (Gupta et al., 2009), which provides a com-
pressive assessment by taking into account of the variability, the bias and the correlation in a multi-objective sense.
**3. Results and discussion**
**3.1. The characterization of the vegetation growth dynamics**
**3.1.1. The vegetation seasonality based on MODIS data**
Figure 4 presents the seasonality of the evergreen forest, tea, savanna grass and shrub cover types using 8-day
MODIS LAI time series in the Mara Basin. The long-term mean annual LAI for evergreen forest is about 2.6 m$^2$/m$^2$
with peaks in April and August. As shown in Figure 4a, the seasonal LAI dynamics show (to some extent) a season-
al variation with an amplitude (peak-to-trough difference) equal to 31% of the annual mean LAI. This seasonal vari-
ation is comparable with the results of Myneni *et al*. (2007) who noted 25% seasonal variation in the Amazon forest.
We note that the seasonal LAI dynamics of the evergreen forest reflects well the seasonal rainfall pattern, with a low
LAI during the dryer months. Our results are in agreement with Kraus (2008), that reported similar findings for
forest sites located in Kenya and Uganda. Additionally, Kinyanjui (2010) analyzed the NDVI in the Mau forest
complex, that includes the forested part of this study, and marked the association of the rainfall pattern and the
NDVI.





In the part of the basin where there is a marked dry season, the seasonal LAI dynamics exhibit a notable variation,
with amplitude (i.e. peak-to-trough difference) that is 85% of the mean annual LAI of 1.4 for savanna grass. As
shown in Figure 4 c and d, low LAI values correspond with the dry months of July - Sept. These observations are
consistent with Zhang *et al.* (2005) who observed a vegetation growth seasonality that reflects the seasonal rainfall
pattern in East Africa.
**3.1.2. The vegetation seasonality simulated by SWAT-T**
As described in section 2.3, the vegetation growth cycle (and thus the LAI) in the SWAT-T model is simulated dy-
namically by using a SMI to annually trigger a new growing season. Hereby the evolution of the LAI follows a
sigmodal pattern, mainly controlled by the daily accumulated heat unit. Table 2 summarizes the list of SWAT model
parameters that control the vegetation growth dynamics. The shape coefficients for the LAI curve (*FRGW₁, FRGW₂,*
*LAIMX₁, LAIMX₂* and *DLAI*) are adjusted by a trail-and-error process such that the SWAT-T simulated LAI mimics
the MODIS LAI. In reality, the minimum LAI (ALAI_MIN) for each cover type varies inter-annually, depending on
the climatic condition; however, this value is fixed in SWAT and need to be provided for each plant (in the plant
database). Thus, ALAI_MIN is set to 2.0, 0.75 and 0.75 for FRSE, RNGE and RNGB, respectively based on the
long-term MODIS LAI (Table 2). Additionally, the optimal temperature and the base temperature in the plant data-
base are adjusted, as shown in Table 2.
Figure 5 presents the average seasonal variation of LAI as simulated by the SWAT-T model between 2002-2009.
The SWAT-T simulated LAI shows a higher seasonal variation as compared to the variation observed from MODIS
LAI for evergreen forest and tea. The amplitude of the evergreen forest is about 47.7% of the average annual
MODIS LAI.
The  SWAT-T simulated LAI for RNGE (RNGB) peaks in April with amplitude range of about 77% (82%) of the
average annual MODIS LAI of 1.4 (1.3) m$^2$/m$^2$ (Figure 5). Overall, the LAI values simulated by the modified
SWAT model tend to reflect the rainfall seasonality pattern. Our results are in agreement with several studies that
noted that the LAI dynamics for natural ecosystem in the Sub-Saharan Africa are associated with the rainfall distri-
bution pattern (Bobée et al., 2012; Kraus et al., 2009; Pfeifer et al., 2014).
One of the advantages of the SMI as a proxy to pinpoint the SOS is not only to trigger a new growth cycle dynami-
cally (i.e. without any management setting) but also the fact that it accounts for the year-to-year shifts in the SOS
due to climatic variations. This is particularly important for long-term land use change and climate change impact
studies. Figure 6 demonstrates the year-to-year shifts as well as the spatial variation in the SOS dates for part of the
Mara River Basin dominated by savanna grassland. Generally, the season change tends to occur in the month of
October (i.e. Julian date 278-304).





### 3.2. The assessment of the improvements of the Leaf Area Index simulation module

The improvement in the modified SWAT model to simulate the vegetation growth cycle and LAI progression for trees and perennials were assessed in two ways. Firstly, we compared the daily LAI as simulated by the standard SWAT2012 (revision 627) under different management settings with the modified version. Secondly, an evaluation was carried out using remotely sensed MOD15A2 LAI time series at 8-day scale.

### 3.2.1. Evaluation of the vegetation growth module improvement

Figure 7 and Figure 8 present the simulated daily LAI for FRSE and RNGE for different management operations along with the rainfall. The purpose of this comparison is to highlight the effect of the model structure changes on the simulated LAI with the default SWAT parameters. The PHU requirement for FRSE and RNGE are set to 3570 and 4100, respectively. The default management setting in SWAT is scheduled using heat unit fractions (Heat unit), whereby planting and kill operations occur at $FR_{PHU}$ 0.15 and 1.2, respectively. With this operation, the simulated LAI is zero at the beginning of each simulation year for all types of vegetation cover (which does not coincide with the dry season). As shown in Figure 7 and Figure 8, this can be partly improved using a date scheduling (Date) for the plant and kill operations (i.e. instead of Heat unit). Additionally, all the setting are removed (no mgt) and the land covers are set to land cover growing (IGRO=1) mode. As a result, the growth cycle resets every year on June, 28 (Figure 7 and Figure 8).

The forested head-water region experiences a unimodal rainfall regime, with March-August being the rainy season. In contrast, a bimodal rainfall regime prevails (March-May and October-December) on the remaining part of the basin. The LAI that is simulated with an uncalibrated model that uses the standard version of SWAT vegetation growth module does not reflect well the seasonality of rainfall in the basin. In contrast, the simulated LAI using the SWAT-T model (i.e. the modified vegetation growth module) tends to follow the seasonal rainfall pattern well (see Figure 5).

Figure 9 depicts the comparison of SWAT and SWAT-T simulated daily potential transpiration timeseries for grassland based on the Penman-Monteith approach. The limitations of the LAI growth cycle in the standard SWAT model also influences the simulation of potential plant transpiration, where a zero potential transpiration is observed during the growing season. In this regard, we observe 14% (12%) zero daily potential transpiration for evergreen forest (grassland) between 2002-2009 using the standard SWAT whilst this reduces to about 2% (0) using SWAT-T. and hence better realism. These results indicate the structural improvements in the plant growth module and hence better realism and significantly reduced inconsistent zero potential transpiration values. We also notice the SWAT-T simulated potential transpiration is consistent while changing the PET method to Hargreaves method in SWAT (results not shown here). Several studies have shown the effect of PET method selection in SWAT on simulated ET and other water balance components (Alemayehu et al., 2015; Maranda and Anctil, 2015; Wang et al., 2006). Alemayehu et al. (2015) reported significant differences in both potential and actual transpiration with the choice of PET method using calibrated SWAT model, which partly ascribed to the unrealistic LAI growth cycle.





Therefore, the improved vegetation growth cycle in the SWAT-T will reduce the uncertainty arising from the mod-
ule structure and thus minimize the uncertainty in simulated ET and runoff.

### 3.2.2. Performance of the LAI simulation

Figure 10 presents the comparison of 8-day MODIS LAI with the LAI simulated by the calibrated SWAT-T aggre-
gated over several land cover classes. We evaluated the degree of agreement qualitatively -by visual comparison-
and quantitatively -by statistical measures. From the visual inspection it is apparent that the intra-annual LAI dy-
namics (and hence the annual growth cycle of each land cover class) from the SWAT-T model correspond well with
the MODIS LAI data. This indicates that the SMI can indeed be used as a proxy to dynamically trigger a new grow-
ing season. This is further supported by a high correlation and a minimal average bias, as shown in Table 3, for most
of the cover types.

### 3.3. The spatial simulation of the evapotranspiration

Table 4 presents the list of SWAT parameters related to flow and evapotranspiration that were adjusted during the
manual calibration. Figure 11 presents the 8-day ET-RS and SWAT-T simulated for the calibration (2002 - 2005)
and validation (2006 - 2009) periods for evergreen forest, tea, grassland and shrubs. Visually, the ET simulated by
the SWAT-T fairly agrees with the RS-ET for all the covers. As shown in Table 3, the statistical performance indi-
ces show a modest performance in simulating ET for the dominant cover types in the basin. The average model
biases for the simulated ET ranges from 7.8% (grassland) to 1.2% (shrub) during the calibration period. Additional-
ly, the correlation between 8-day ET from the SWAT-T and the RS-ET varies from 0.67 (tea) to 0.72 (grassland).
Overall, we mark similar performance measures during the calibration and validation period, suggesting a fair repre-
sentation of the processes pertinent to ET.
The variability of the evapotranspiration is controlled by several -biotic and abiotic- factors. The 8-day ET time
series as simulated by the SWAT-T model illustrates the variation in the temporal dynamics of ET in the study area.
For land cover types located in the humid part of the basin (evergreen forest and tea) there is no clear temporal pat-
tern (Figure 11). In contrast, the areas covered by evergreen forest and shrubs show a clear seasonality in the simu-
lated ET. These observations are consistent with the seasonality of the simulated LAI, as shown in Figure 5.
The SWAT model parameters were adjusted by trial and error with the objective of improving the agreement be-
tween the SWAT-T simulated ET and the RS-ET. Perhaps, this may not be as robust as an automatic calibration as
the latter explores a larger parameters space. However, the manual calibration is sufficient to illustrate the impact of
the modification on the vegetation growth cycle and its effect on the water balance components. The higher water
use by evergreen forest as compared to other land cover classes is reflected by a lower ESCO, and a higher
GW_REVAP and GSI (Table 4). The lower ESCO indicates an increased possibility of extracting soil water to satis-
fy the atmospheric demand at a relatively lower soil depth. The higher GW_REVAP points to an increased extrac-





tion of water by capillary rise and deep-rooted plants from the shallow aquifer. Similar findings were reported by
Strauch and Volk (2013).
The improvements in the seasonality of the annual growth cycle in the SWAT-T model is also noted by a realistic
spatial and temporal representation of ET and LAI (Figure 12 and Figure 13). Figure 12 (upper row) exhibits the
monthly ET at HRU level for the wettest month (April) and driest month (August) in 2002. The lower portion of the
basin, with dominant savanna cover, experiences a monthly ET between 16 and 63 mm in August and between 41
and 93 mm in April. These estimates are also well reflected in the spatial distribution of the average monthly simu-
lated LAI (Figure 12 lower row). We notice that the linear relationship between ET and LAI is stronger, in general,
for grassland and shrubs than for evergreen forest and tea. The lower correlation for tea and evergreen forest could
be partly attributed to the high evaporation contribution of the wet soil, as the upper portion of the basin receives
ample rainfall year round. In this part, it is worth noting the tea harvest operation and hence low transpiration and
high evaporation contribution. We also note that during the wet month the spatial variability of ET is higher than
that of the LAI (Figure 12).
**3.4. The performance of the flow simulations**
Figure 14 presents the comparison of daily SWAT-T simulated flow with observation for the calibration and valida-
tion periods. Visually, the simulated hydrograph fairly reproduced the observation. The average biases of the
SWAT-T model simulated daily flow compared observations are 3.5 and 15.5% during the calibration and valida-
tion periods, respectively (Table 3). The degree of correspondence between daily observed and simulated flows
results a good correlation during calibration 0.72 and validation 0.76 periods. Additionally, the overall comprehen-
sive assessment using KGE reveals a good performance of the SWAT-T model in simulating the daily flows. Gener-
ally, the model tends to underestimate the baseflow and this is more pronounced during the validation period. This is
probably associated with the overestimation of the ET for evergreen forest (6.6%) during the validation, since ET
has a known effect on the groundwater flow.
**4. Summary and conclusions**
We presented an innovative approach to improve the simulation of the annual growth cycle for trees and perennials -
and hence improve the representation of the evapotranspiration- for tropical conditions in SWAT. The robustness of
the changes made to the standard SWAT2012 version 627 have been assessed by comparing the model outputs with
remotely sensed 8-day composite LAI data, as well as with RS-ET data. Towards this, we presented a simple but
robust soil moisture index (SMI), a quotient of rainfall (P) and reference evapotranspiration (PET), to trigger a new
growing season after a defined dry season. The new growing season starts when the SMI index exceeds or equals
0.5, meaning 50% or more of the atmospheric water demand is satisfied. To assure the availability of sufficient soil
water for a new growing season, we used the pentad P and PET to compute the SMI. Therefore, we have modified
the plant growth model of the standard SWAT model (SWAT-T) to simulate the vegetation growth cycle and hence





the LAI dynamically (with no management setting) using the SMI as a proxy for the season change. The Moderate
Resolution Imaging Spectroscopy (MODIS) LAI time series (2002-2009) at 8-day has been used to evaluate the LAI
simulated by the SWAT-T. Additionally, the overall performance of the SWAT-T model for simulating flow and
evapotranspiration (ET) has been compared with observed flow and remote sensing-based ET (RS-ET).
The structural improvements in the LAI simulation have been demonstrated by comparing simulation of LAI using
standard SWAT and SWAT-T with default parameters. The results indicated that the modified module structure for
the vegetation growth exhibits temporal progression patterns that are consistent with the seasonal rainfall pattern.
Further, we note better consistency in the simulated potential transpiration for perennial and trees regardless of the
choice of the PET method, suggesting the usefulness of the improved LAI temporal dynamics in reducing the model
structural uncertainty.
Our results show that the calibrated SWAT-T simulated LAI corresponds well with the MODIS LAI for various land
cover classes in the Mara Basin, indicating the realistic representation of the start of the new growing season using
the SMI. Our results also demonstrated the year-to-year variation of the start of the new growing seasons, due to the
variability in the P and PET.
The improvement in the vegetation growth cycle in SWAT is conformed with a good agreement of simulated ET
with RS-ET, particularly for the grassland. Additionally, the daily flow simulated with the SWAT-T mimics well the
observed flows for the Nyangores River. In general, the SWAT-T model shows a good skill in simulating the major
water balance components. Previous SWAT modeling studies, e.g. Mango *et al.* (2011) reported poor performance
of SWAT in the study area for the same location and period. Therefore, we believe that the good performance
demonstrated in this paper is partly attributed to the improvement in the vegetation growth cycle.
This research used bias-corrected satellite P and PET derived from global weather data as forcing. Given the inher-
ent errors in the input data, we acknowledge the inevitable influence on the model performance and simulation out-
puts. However, we believe that the quality of the input data used is sufficient to evaluate the plant growth module
modifications in SWAT on the LAI seasonal development and the water balance components. The SWAT-T devel-
oped in this study could be a robust tool for simulating water and carbon fluxes as well as various land use and cli-
mate change impact studies in tropical ecosystems.
**5. Acknowledgments**
We would like to thank Thirthankar Roy, the University of Arizona, for providing bias-corrected satellite rainfall
products. We also would like to thank the Water Resource Management Authority (WRMA) of Kenya for provision
of streamflow data. The technical help on FORTRAN coding from Befekadu Woldegeorgis, Vrije Universiteit Brus-
sel, is very much appreciated.



## 6. Data Availability

The modified SWAT simulator for Tropics is available upon request from the first author.

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

**Table 1 Summary of the inputs of the SWAT model and the evaluation datasets.**

|  | Spatial/temporal resolution | Source | Description |
|---|---|---|---|
| Rainfall | 5 km / 1-day | Roy *et al.* (2017) | Bias-corrected satellite rainfall for Mara basin |
| Climate | 25 km / 3-hour | Rondell *et al.* (2004) | Max. and min. temperature, relative humidity, wind, solar radiation |
| Land cover classes | 30 m | FAO (2002) | Land cover classes for East Africa |
| DEM | 30 m | NASA | Elevation model |
| Soil classes | 1 km | FAO (2009) | Global soil classes |
| Discharge | daily | local ministry | River discharge at Bomet |
| ET | 1 km / 8-day | Alemayehu *et al.* (2017) | ET maps for Mara basin |
| MOD15A2 | 500 m / 8-day | NASA | Global LAI |


**Table 2 Summary of the SWAT parameters that control the vegetation growth and the LAI with their default and cali-**
**brated values.**

| Parameter definition (unit) | | Default (calibrated) | | |
|---|---|---|---|---|
|  |  | FRSE | RNGE | RNGB |
| *BIO_E* | Radiation-use efficiency $((kg/ha)/(MJ/m^2))$ | 15 (17) | 34 (10) | 34 (10) |





| | | | | |
|---|---|---|---|---|
| *BLAI* | Maximum potential leaf area index ($m^2/m^2$) | 5 (4.0) | 2.5 (3.5) | 2 (3.5) |
| *FRGW₁* | Fraction of PHU corresponding to the 1st point on the optimal leaf area development curve | 0.15 (0.06) | 0.05 (0.2) | 0.05 (0.2) |
| *LAIMX₁* | Fraction of BLAI corresponding to the 1st point on the optimal leaf area development curve | 0.7 (0.15) | 0.1 (0.1) | 0.1 (0.1) |
| *FRGW₂* | Fraction of PHU corresponding to the 2nd point on the optimal leaf area development curve | 0.25 (0.15) | 0.25 (0.5) | 0.25 (0.5) |
| *LAIMX₂* | Fraction of BLAI corresponding to the 2nd point on the optimal leaf area development curve | 0.99 (0.30) | 0.7 (0.99) | 0.7 (0.99) |
| *DLAI* | Fraction of total PHU when leaf area begins to decline | 0.99 (0.30) | 0.35 (0.99) | 0.35 (0.99) |
| *T_OPT* | Optimal temperature for plant growth (ºC) | 30 (25) | 25 (30) | 25 (30) |
| *T_BASE* | Minimum temperature for plant growth (ºC) | 0 (5) | 12 (5) | 12 (5) |
| *ALAI_MIN* | Minimum leaf area index for plant during dormant period ($m^2.m^2$) | 0.75 (2.0) | 0 (0.75) | 0 (0.75) |
| *PHU* | Total number of heat units needed to bring plant to maturity | 1800 (3570) | 1800 (4100) | 1800 (4100) |


**Table 3 Summary of the performance metrics for the SWAT-T for simulating LAI, ET and flow. Note that the for LAI**
**and ET the performance is at 8-day whilst daily for flow.**

| | LAI calibration (validation) | | | | ET calibration (validation) | | | | Flow calibration (validation) |
|---|---|---|---|---|---|---|---|---|---|
| | **FRSE** | **Tea** | **RNGE** | **RNGB** | **FRSE** | **Tea** | **RNGE** | **RNGB** | **Flow** |
| **r** | 0.94 (0.93) | 0.83 (0.83) | 0.89 (0.86) | 0.92 (0.88) | 0.71 (0.68) | 0.67 (0.64) | 0.72 (0.77) | 0.66 (0.72) | 0.72 (0.76) |
| **%bias** | 1.5 (0) | 0.1 (0.2) | -3.7 (-0.4) | -1.3 (4.6) | 3.7 (6.6) | -1.7 (0.5) | 7.8 (11) | 1.2 (2.9) | 3.5 (15.5) |
| **KGE** | 0.50 (0.62) | 0.42 (0.44) | 0.86 (0.85) | 0.88 (0.86) | 0.71 (0.67) | 0.62 (0.62) | 0.69 (0.74) | 0.66 (0.72) | 0.71 (0.71) |


**Table 4  List of the manually calibrated SWAT parameters.**





| Parameter | Definition (unit) | Initial (calibrated) | | |
|---|---|---|---|---|
| | | FRSE | RNGE | RNGB |
| $SOL\_Z^1$ | Soil layer depths (mm) | 300 [1000] (480 [1600]) | 300[1000] (480 [1600]) | 300[1000] (480 [1600]) |
| $SOL\_AWC^2$ | Soil available water (mm) | 0.26-0.31 [0.27-0.29] (0.18-0.21 [0.18-0.20]) | 0.26-0.31 [0.27-0.29] (0.18-0.21 [0.18-0.20]) | 0.26-0.31 [0.27-0.29] (0.18-0.21 [0.18-0.20]) |
| ESCO | Soil evaporation compensation factor (-) | 0.95 (0.88) | 0.95 (1) | 0.95 (1) |
| EPCO | Plant uptake compensation factor (-) | 1 (1) | 1 (1) | 1 (1) |
| GSI | Maximum stomatal conductance at high solar radiation and low vapor pressure deficit (m.s$^{-1}$) | 0.002 (0.006) | 0.005 (0.0035) | 0.005 (0.004) |
| REVAPMN | Depth of water in the aquifer for revap (mm) | 750 (100) | 750 (100) | 750 (100) |
| $CN2^3$ | Initial SCS curve number II value (-) | 55 [70] (38 [48]) | 69 [79] (81 [92]) | 61 [74] (71 [87]) |
| SURLAG | Surface runoff lag time (day) | 4(0.01) | 4(0.01) | 4(0.01) |
| ALPHA_BF | Baseflow recession constant (day) | 0.048 (0.2) | 0.048 (0.2) | 0.048 (0.2) |
| GWQMN | Shallow aquifer minimum level for base flow | 1000 (50) | 1000 (50) | 1000 (50) |
| GW_REVAP | Groundwater 'revap' coefficient (-) | 0.02 (0.1) | 0.02 (0.02) | 0.02 (0.02) |
| RCHRG_DP | Deep aquifer percolation fraction (-) | 0.05 | 0.05 | 0.05 |

|  |  |  |
|---|---|---|
| (0.3) | (0.1) | (0.1) |

[1]SOL_Z values for the top [and lower] soil layers depth
[2]SOL_AWC values range for the top [and lower] soil layers depending on soil texture and bulk density
[3]CN2 values for soil hydrologic group B[C]

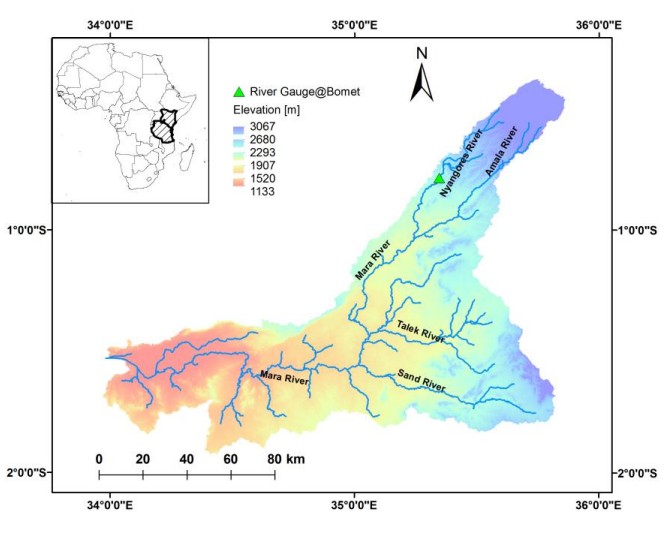


625                      (a)

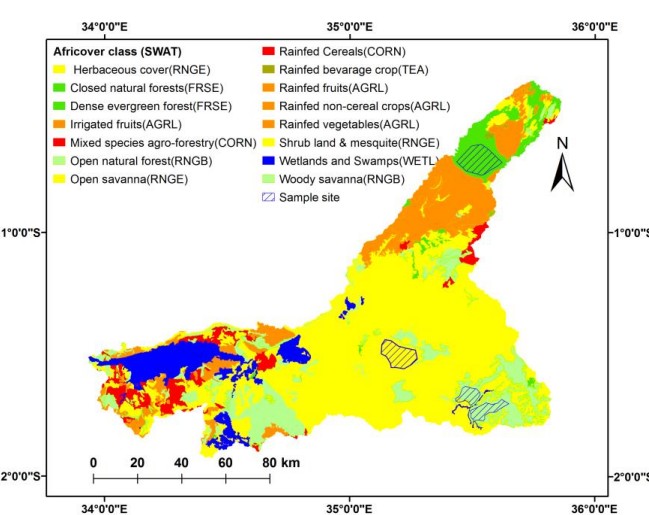





(b)
**Figure 1 Location of the Mara Basin (a) and its land cover classes (b). Note the sample sites location for the major natural**
**vegetation classes that are used to mask the Moderate Resolution Imaging Spectroradiometer (MODIS) Leaf Area Index**
**(LAI).**

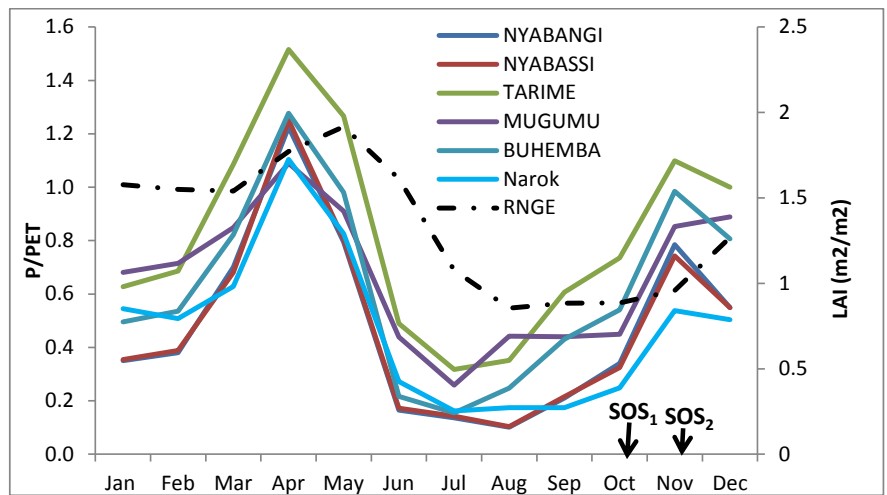


**Figure 2 The climatological moisture index (SMI) for meteorological stations across the Mara Basin and the mean Leaf**
**Area Index (LAI) for the savanna ecosystem (dotted line). SOS₁ and SOS₂ represent the start-of-months (SOS) to trigger**
**growth whenever 50% of the atmospheric demand is exceeded or equalled.**

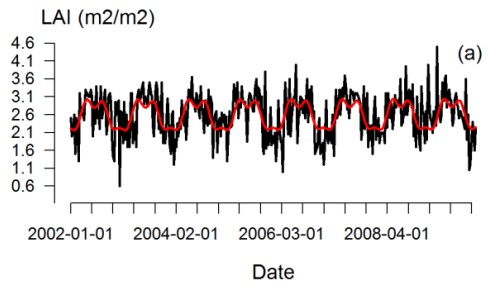
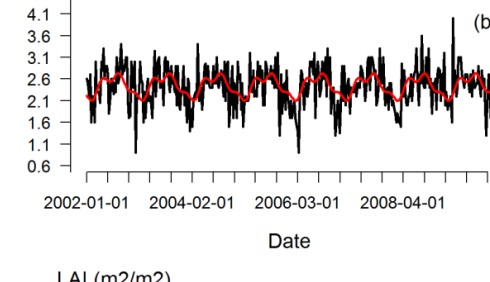
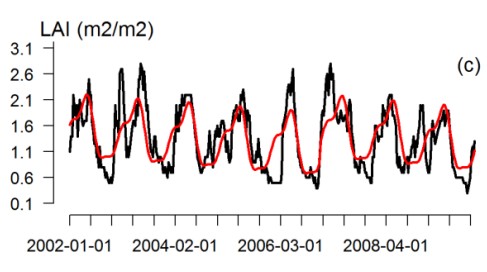
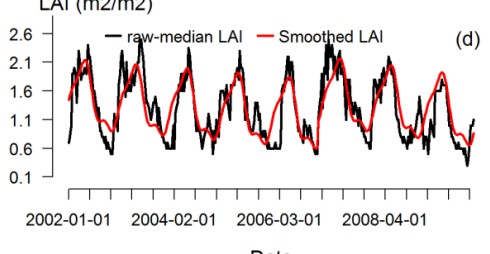






Figure 3 The 8-day raw-median LAI time series for evergreen forest (a), tea (b), grass (c) and shrub (d) sample sites. The
raw-median LAI is smoothed using the Breaks For Additive Seasonal and Trend (BFAST) method (Verbesselt et al.,
639    2010).

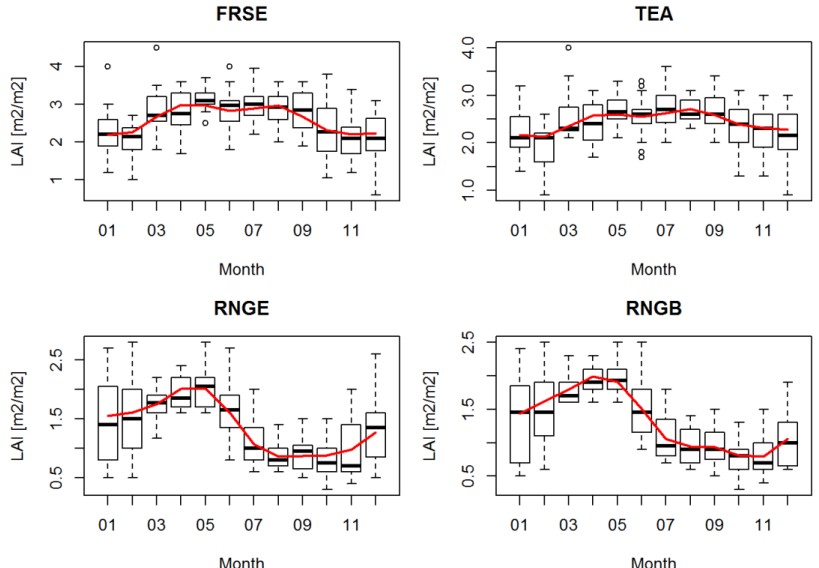


Figure 4 The seasonal variability of the LAI using the 8-day MOD15A2 time series for 2002-2009. The boxplots present
the median LAI and Interquartile Range for each month; the solid lines depict the smoothed seasonal LAI.

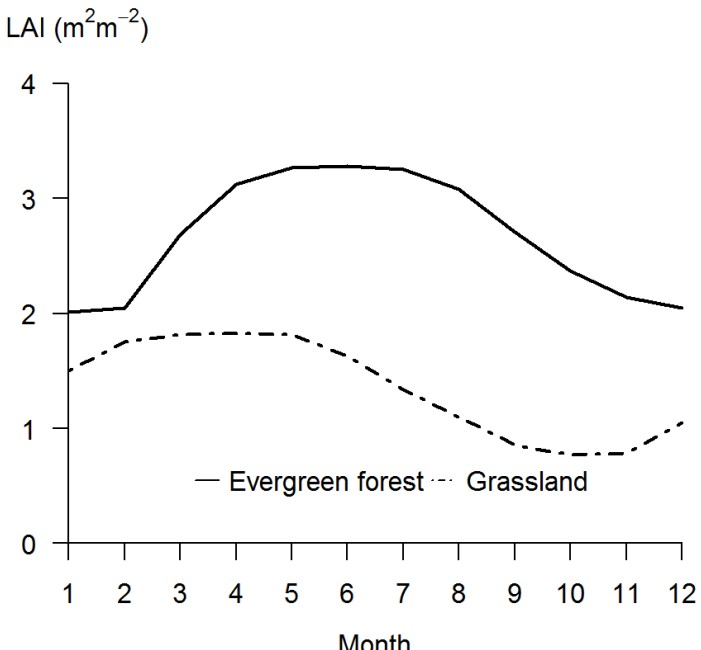






**Figure 5 The seasonal pattern of the SWAT-T simulated LAI (2002-2009) for evergreen forest and grassland.**

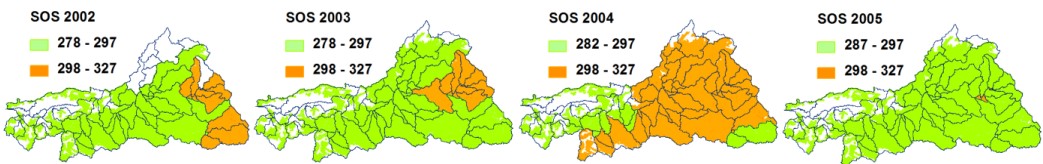


**Figure 6 The inter-annual and spatial variation of the start of the rainy season for the savanna vegetation in the Mara**
**River basin for 2002-2005. Note that Julian dates are used and the mapping is done at HRU scale.**

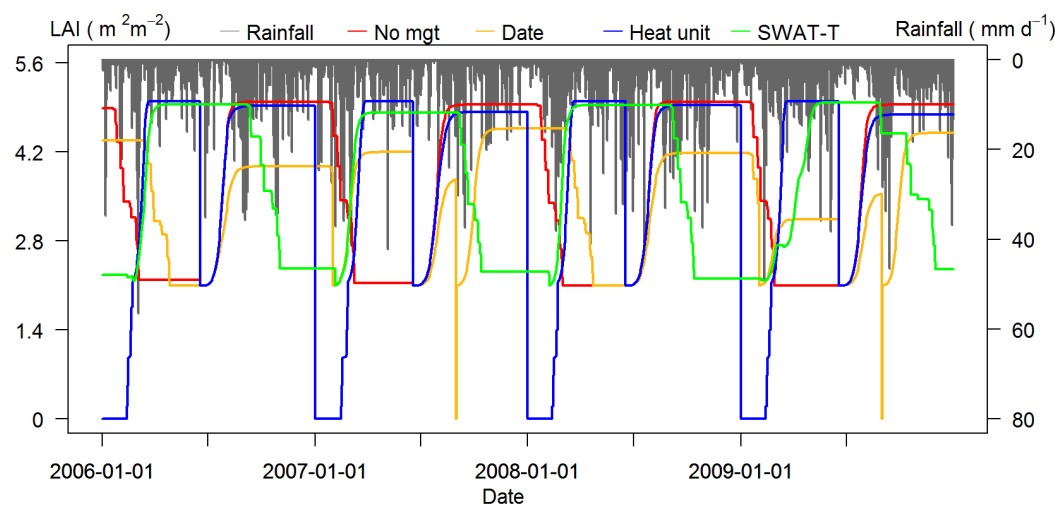


**Figure 7 The LAI as simulated by the SWAT-T and the standard SWAT models for different management setting for**
**evergreen forest using default SWAT parameter values. See management setting explanations in the texts.**

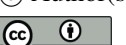


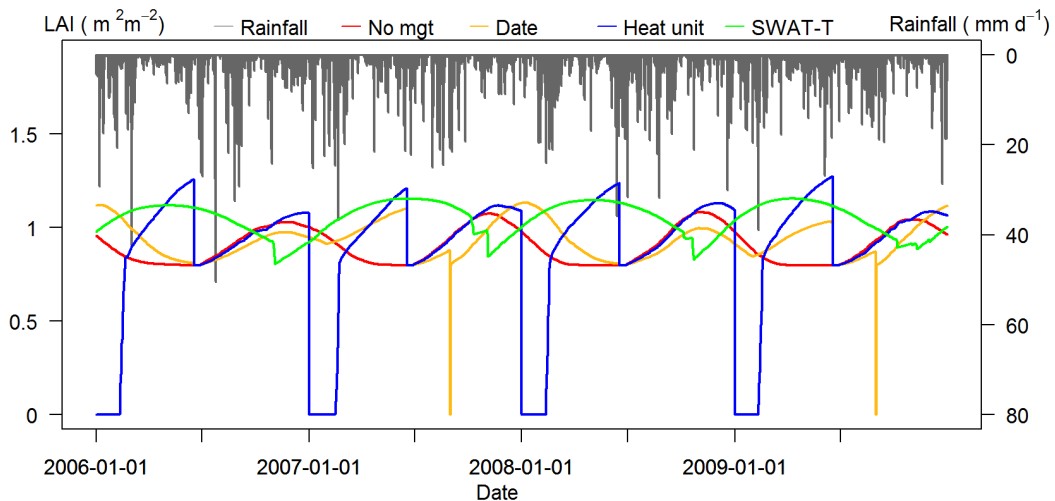


**Figure 8 The LAI as simulated by the SWAT-T and the standard SWAT models for different management setting for grassland using default SWAT parameter values. See management setting explanations in the texts.**


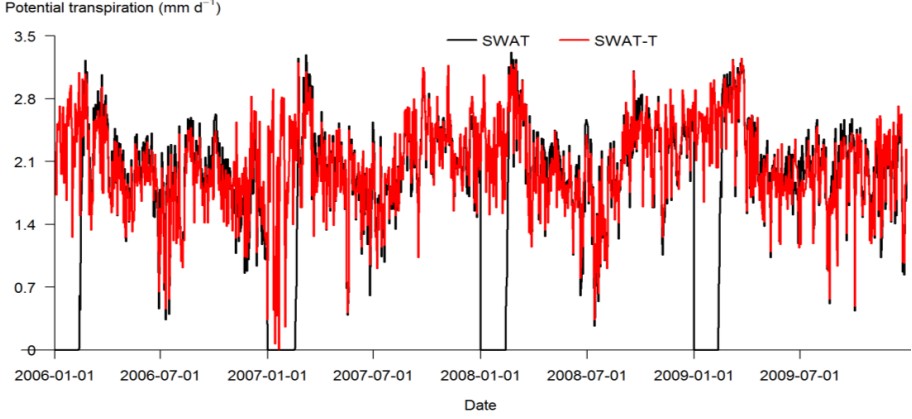


**Figure 9 Inter-comparison of Penman-Monteith-based daily potential transpiration simulated by SWAT-T and SWAT models for grassland. Note that the heat unit scheduling is used in SWAT model.**






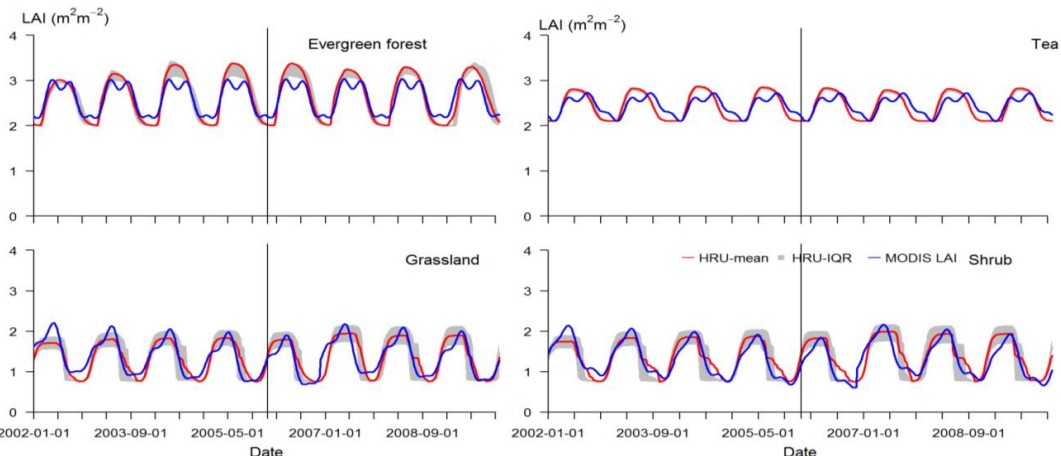


**Figure 10 The MODIS LAI and the SWAT-T model simulated HRU weighted aggregated 8-day LAI time series (2002-**
**2009). The gray sheds indicate the boundaries of the 25th and 75th percentiles. The vertical line marks the end of the cali-**
**bration period and the beginning of the validation period.**

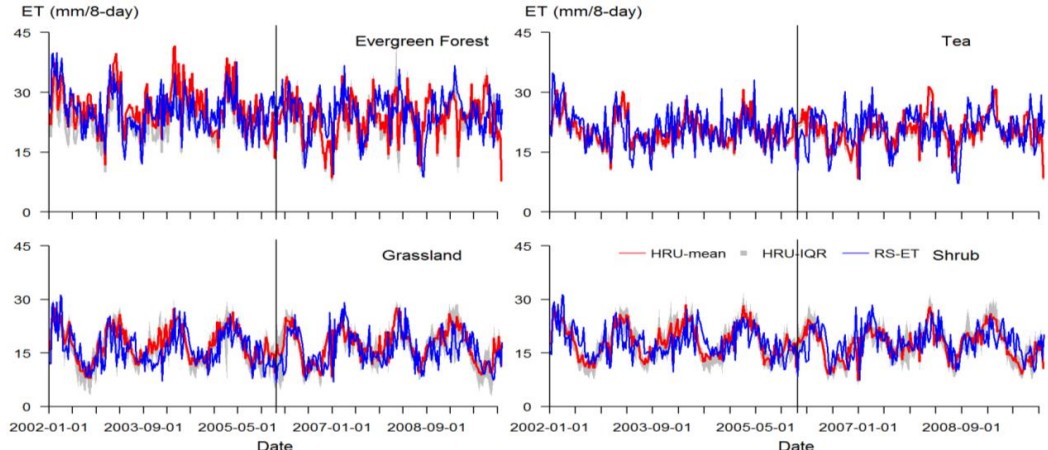


**Figure 11 The comparison of RS-ET and SWAT-T simulated ET. Note that for SWAT-T HRU level ET is aggregated per**
**landcover. The gray sheds indicate the boundaries of the 25th and 75th percentiles. The vertical line marks the end of the**
**calibration period and the beginning of the validation period.**





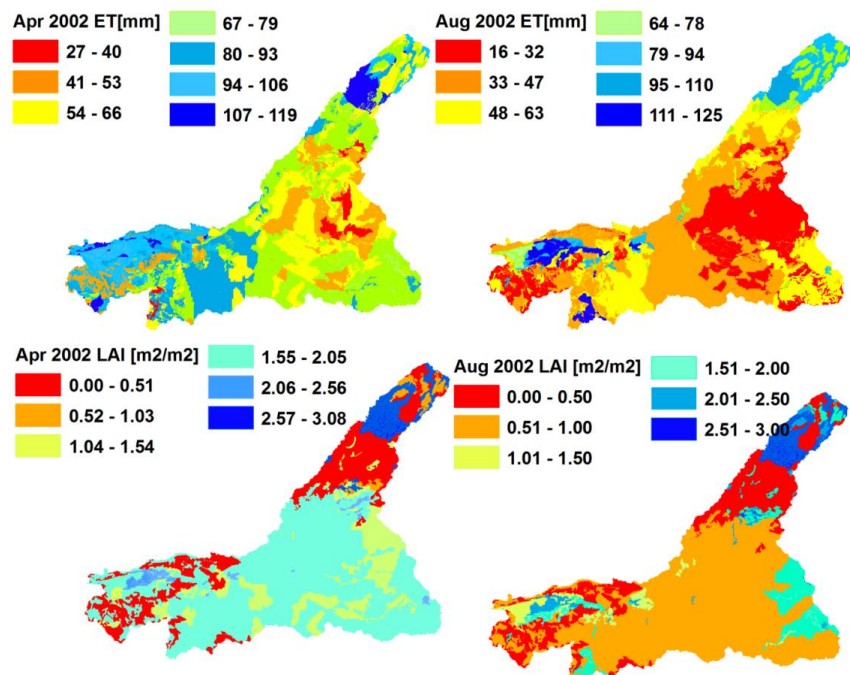


**Figure 12 SWAT-T simulated monthly ET (upper row) and LAI (lower row) for April (wet) and August (dry) 2002 at HRU level.**






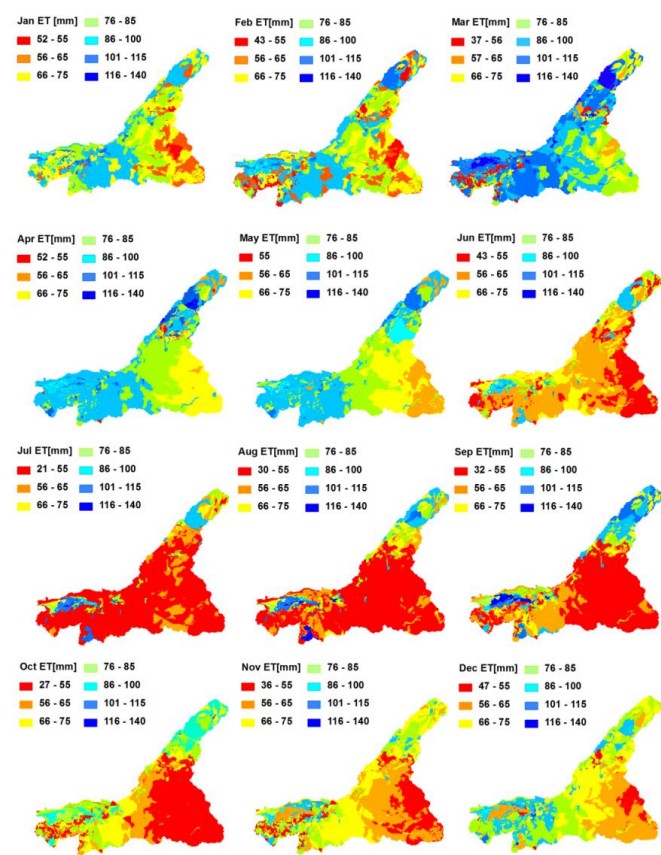


**Figure 13 The average seasonal and spatial distribution of ET (2002-2009) in the Mara Basin, as simulated by the SWAT-T model at HRU level.**





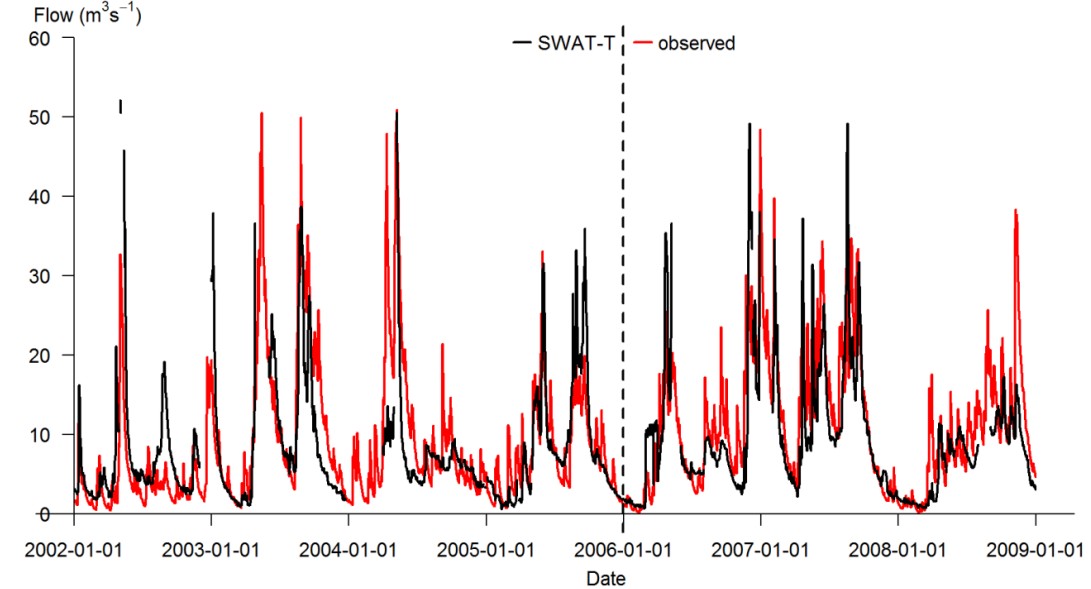


**Figure 14 Observed and simulated flows for the Nyangores River at Bomet.**

678