# Peer review of "An improved SWAT vegetation growth module and its evaluation for four tropical ecosystems"

_Hydrology and Earth System Sciences, 2017_

## Short Comment (SC1) · 9 Mar 2017

| Line | Comment |
|---|---|
| 10-11 | *"where the major plant growth controlling factor is the rainfall (via soil moisture) rather than temperature."* – it seems as if you mean to say that temperature is the preferred plant growth controlling factor, maybe you can cut the sentences up into two sentences: **1)** However, SWAT has limitations in simulating the seasonal growth cycles for trees and perennial vegetation in tropics. **2)** In the tropics plant growth is mainly controlled by rainfall (via soil moisture), whereas in SWAT plant growth is temperature controlled. |
| 57 | *"Normalized Vegetation Index (NDVI)"* – shouldn't this be: "Normalized Vegetation Difference Index (NDVI)"? |
| 97 | *"poorly drained soils cover the plateau"* I was wondering what you meant with "plateau". I guess the Mau escarpment? |
| 192-193 | Does this mean that there can be set two starts of the rainfall seasons (SOS) for a bimodal rainfall regime? : So there is an end of the dry season [SOS1] and a beginning of the rainy season [SOS2] for the long rains (for the Mara for example) and there is another end of the dry season [SOS3] and a beginning of the rainy season for the short rains [SOS4] ? |
| 196 | *"pentad ratio"* – I had never heard of this, I don't know whether it is a common term (maybe it's because I am a non-native speaker of English), but to make it easier to read you might also just say " five day ratio". |
| 302-306 | This trial-and-error process was it done manually or with for example SWAT-CUP? And if so, did you have some sort of a steps that you followed in this procedure?

I am curious because personal experience taught me that altering these five LAI parameters in SWAT-CUP or directly in the input .mgt or .plant files, could give pretty random outcomes in terms of LAI curves or PET, and results of altering multiple LAI parameters at the same time are difficult to predict. |
| 308-309 | Do you know why Kilonzo (2014) [Penmann-Monteith] and Mwangi (2016)[Hargreaves] recommend using a minimum LAI for FRSE of respectively 3 and 4? For Mwangi this worked very well. For tropical forest in Brasil this is reasonable, but looking at the mean annual LAI in the FRSE of the Mau escarpment of 2.6 this seems too high of an estimate.

I also saw in **figure 7** That you had set the minimum LAI for SWAT-T to about 2.2 and maximum LAI to 5. Was this just for the purpose of giving an example at the same setting as the default or was this also the value as used in your simulations? |
| 334-336 | "We also notice the SWAT-T simulated potential transpiration is consistent while changing the PET method to Hargreaves method in SWAT (results not shown here)."

Interesting! Is this also the case for the PET at times where LAI > 3 ?
Did you also try using the Priestley-Taylor (P-T)?

Personal modelling experience in the region taught me that the annual PET using the P-T method is often lower then when using the Hargreaves or P-M, thus giving a lower AET, thus implicating that there is more water in the catchment system to "play with" as in comparison to the P-M or Hargreaves. |

---

## Referee Comment (RC1) · Anonymous Referee #1 · 24 Mar 2017

In this manuscript, the authors modify the phenology algorithm of SWAT in simulating tropical vegetation. This work provided some interesting discussion of limitations in SWAT plant module. I have several major concerns about this work. First, writing of this work should be further improved to make it publishable. Second, organization of the introduction and method sections should be changed following requirement of a scholarly journal. line 47, what does this mean? do you mean that SWAT could not represent changes deciduous forest? line 55, extra space. line 58, for -> of. to simulate line 59, a rainy season line 67-68ïijŇthis sentence should be moved to the method section line 71-81, I expected to see objectives of this work, but the authors are describing their methodology, which should be moved to the next section (method) line 92, a long rainy season line 93, across what? section 2.2.1 was copied from the SWAT manual. I suggest to condense this part significantly, or move it to a supplementary

section line 172-173, this is not correct. Kill and dormancy are totally different. If some one use this to regulate phenology, they must have made mistakes. line 175-181, these discussions should be presented in your Introduction, or the Discussion section line 183, what are the data source of P and PET? line 192-193, remove these two sentences line 201-207, do you have any reference to support your rules? line 238, a ... site? line 242, why lai of 1.5 is removed? line 245, what break? line 253-254, I do not understand how the LAI patterns match precipitation.

line 259-260, awkward expression. consider to improve line 270, change the term 'flow' to stream flow or river discharge through out the manuscript line 272-273, remove the second period line 285, seasonality of what? line 290-291, I am not convinced that lai reflects changes in rainfall. you need to provide some statistical information here. and what about the correlation between temperature and lai? line 306-308. very confusing line 311 figure 5 only show two land covers. What about the other two? line 316. I do not see seasonal variation from figure 5 line 325, in October line 328, was, first line 329, second line 330, a 8-day scale line 339-341. do not understand what does this mean line 344, consider to revise line 351-353, not clear. consider to revise line 361, I do not see improvement in runoff line 387-389, remove this sentence line 396-406. It is surprising that the authors did not evaluate SWAT ET simulations line 412. Is this a sentence?

---

## Referee Comment (RC2) · Anonymous Referee #2 · 27 Mar 2017

Review of hess-2017-104 by Alemayehu et al.: "Improved SWAT vegetation growth module for tropical ecosystem"

The authors develop a new vegetation growth module for tropical ecosystems in SWAT. In particular, they use a soil moisture index to initiate a new growing cycle within two pre-defined months. They evaluate the growth module with regard to LAI, ET, and river discharge with satisfactory results. The topic is of current scientific interest, as several authors have previously outlined that the default vegetation growth for e.g. forests in SWAT is not applicable in the tropics. The manuscript is mostly well prepared. However, the paper would benefit a lot if it was more structured according to the evaluation of the vegetation growth module. In particular, this applies to the results and discussion part. This focus should be set very clearly in a revised version. Moreover, some parts of the manuscript require further, more detailed, or more precise information. The pro-

vided comments should be addressed before accepting this manuscript for publication.

General comments:

1) The authors should take a decision on the aim of the manuscript. Do they aim at providing an improved plant growth module for SWAT in the tropics? Or do they aim at showing their adjustment of the model to a specific catchment? Currently, the title reads like the first is the case. However, in many parts the paper reads like the latter is the case. For the first aim the authors need a stronger focus on evaluation of the plant growth model and they need to program the module as flexible and transferable as possible. Right now there are parts that are not immediately transferable to other catchments. The authors are encouraged to sharpen the paper with regard to the first aim. However, if this was not their aim they may also go for the second aim and adjust the title accordingly.

2) Clear separation of calibration and validation period is required. This also applies to the calibration of plant parameters.

3) A clear calibration and validation strategy regarding the different parameters used for calibration and validation is needed. E.g. right now, it is not clear which parameter was calibrated first and why?

4) LAI is prescribed based on satellite data (l.256). Why is this necessary? This makes the model less flexible and non-transferable to other catchments without following a similar approach.

5) The third chapter "results and discussion" is sometimes hard to understand as discussion of model parameters and results is mixed with model validation. I strongly suggest to rework the structure and separate the "results" from the "discussion" part.

6) Why is it necessary to prescribe the two month in which the growing season starts?

7) It would be very good, if you could validate the modeled begin of the growing season using independent data. Is there any data that you have available to do this?

Line specific comments:

l.7: SWAT is a hydrologic model. The term "simulator" is not very common for SWAT in the literature. Suggest to replace this by "model" in the whole manuscript.

l.13: "uses of a simple..." Please improve the language.

l.15: Would be good to include information here, how the dry season is defined.

l.18: "flow" – The authors probably refer to stream flow. Should be more precise throughout the manuscript.

l.19: Please include information, which RS-ET was used.

l.20: "could be..." Please be more precise. In which situations is it useful?

l.44: Please be more precise, i.e. "dormancy, which is defined as a function of daylength and latitude".

l.47: As I read it, they do not report a shift, but shifted the dormancy period to a prescribed dry season (see p.1786). Please improve the statement.

l.49-55: You are reviewing tropical regions. However the Kalahari has a subtropical climate. Please improve.

l.73: "phonological"

l.67-77: These lines include a lot of information on methodology. Please shift the methodological parts to the methodology section.

l.103: "SWAT uses a GIS based interface". Not precise. You can use GIS to prepare input files for SWAT. Please improve.

l.126 following: Please add citations for formulas.

l.128: Grammar.

l.134: "endo"

l.170-173: This passage is not quiet to the point. SWAT does not offer heat unit scheduling to solve the issue of plant growth in the tropics. In fact, both scheduling options will not help, as long as the temperature dependant dormancy period is still activated. Please improve.

l.190-197: How are SOS1 and SOS2 defined? By a threshold, or by the increase of the SMI? Are they set by the user? If they are set by the user, the model is not as flexible – is this necessary? It should be highlighted in the other parts of the manuscript that the start of the growing season is not fully dynamic but triggered within a pre-defined period.

l.211: Please add a reference for the DEM (also in table 1 & add time period for river discharge in table 1).

l.217: Please add a reference for the SWAT land use codes, so that non-SWAT users can look these up.

l.222: Abbreviation "TMPA" is not explained at first mentioning. Please improve.

l.226-27: Please add some (short) reasoning for this adjustment, so that the reader can understand the idea of this approach without reading the referred paper.

l.237: Which forest biomes? What about others?

l.238: What do you mean by "land cover mix"?

l.238: Grammar.

l.238: Please add the sizes of the homogenous sites.

l.242: Please provide reasoning for selecting the threshold value 1.5.

l.242: Are these gaps resulting from the previous masking?

l.243-245: Sentence not clear. Please improve.

l.266: Is this measured NDVI or remote sensing based? If it is remote sensing derived,

are the two products independent from each other?

l.271-273: Please include how large the gauged headwater area is. Also, add information on when the gaps happen, e.g., at similar times in the year or mainly in one year?

l.275: This is not precise and could be misleading. As I understand it, the SMI triggers the growing season within a predefined period of 2 months. Please improve.

l.275-281: The model calibration and validation strategy is not clear. The authors use stream flow, LAI and ET. However, it is not clear which parameter is used first – or are they combined? Please improve this section.

l.287: Peaks in April and August are not shown in Figure 4. Please clarify.

l.291: Please add the "drier months" you are referring to in brackets.

l.285-299: Why are you showing tea in Fig. 4? It is shown but not mentioned here.

l.302: Again, this is not precise and could be misleading. As I understand it, the SMI triggers the growing season within a predefined period of 2 months. Please improve.

l.308-310: The authors use long-term MODIS LAI to parameterize the model. It would be much better if these values were derived from the calibration period, so that calibration and validation data are strictly independent. Even though, I do not expect a pronounced change in LAI values in calibration and validation period, I would recommend to only use data from the calibration period for model setup.

l.313: Not clear why the authors express the amplitude of simulated LAI as a percentage of the average annual MODIS LAI. Please clarify your validation strategy.

l.316: Why are simulated and remote sensing based LAI not directly compared and shown as later presented in figure 10? That is what the reader would expect at this point. A scatter plot is also useful in this context.

l.320-323: This passage can be shifted to a discussion part.

l.323-325: This is only an effect of your modeling. Can you use this for validation? E.g. by comparing to independent data (e.g. satellite derived) on the beginning of the growing season?

l.337: But with a correct definition of ALAI_MIN it would not be 0, right? Please make clear, what you are validating. This improvement is due to satellite-based improvement of the parameter ALAI_MIN and not because of the improvement of the plant growth module.

l.332-341: You need to explain your different model setups in more detail. E.g. IGRO=1, will not be understandable for non-SWAT users. Moreover, please use only meaningful model parameterizations. It does not make sense to compare SWAT-T to a model that does not work properly. Many different model setups are irritating. I would suggest to compare SWAT-T to the best possible model parameterization achieved without code changes.

l.342-347: Not sure, why this paragraph is provided here. Figure 5 was already discussed before. Moreover, the comparison to an uncalibrated model is not a fair evaluation (see comment above).

l.352: "standard SWAT" Please define this and provide a model evaluation for this setup. If this model does not work it is not useful for comparison.

l.353-354: "better realism". Please improve the language.

l.354: Did you test for significance?

l.368: Quantify where possible.

l.363-369: This short paragraph presents some of the most valuable results. Please provide further details here. E.g. in which periods and why do MODIS LAI and simulated LAI not match well, as shown in Fig. 10?

l.374-375: It is hard to see from Fig. 11 whether ET values match well as the lines overlap. Please add a scatter plot.

l.384: Contradiction to the previous sentence. Do you mean "grassland" instead of "forest"?

l.385: But Figure 5 shows a seasonality for both? This is contradictory to the previous sentences. Please clarify.

l.387-395: Again the structure is not clear. The authors evaluate parameter values at this point. Why is this needed and presented here? Please provide some justification or remove the paragraph.

l.404: Sentence not clear. Please improve.

l.405: Please quantify the spatial variability.

l.396-406: Why do you not compare the spatial distribution of simulated ET and LAI to the spatial distribution of MODIS based ET and LAI? That could be another valuable comparison that might be more useful than a presentation of modeled values.

l.417-452: The conclusion should be shortened so that it only includes the most important conclusion drawn from your study.

l.432: This sentence is misleading ("default parameters"). As I understand your setup SWAT-T parameters were calibrated. Please clarify.

l.434: What do you mean by "potential transpiration"? Potential evapotranspiration?

l.435: The results with the other PET method were not shown. Please focus on what you have shown in this paper. If it is important, please include it, if not please remove.

l.438: Misleading statement regarding SMI initiation of growing season, see also earlier comments. Please improve.

l.441: "conformed"

l.447-452: This is more suitable for a discussions section. Please shorten or remove.

l.451: "could be. . ." Please be more precise. In which situations is it useful?

l.451: "carbon fluxes" Not shown. Please remove.

---

## Author Comment (AC2) · 22 Jun 2017

**Response to Anonymous Referee #1**

We thank the referee for this thorough review and for the numerous constructive suggestions. We agree with the need of further improvements in the manuscript organization. We therefore have addressed all the comments below point-by-point (italics). We have attached as supplementary material the revised manuscript with changes in blue.

In this manuscript, the authors modify the phenology algorithm of SWAT in simulating tropical vegetation. This work provided some interesting discussion of limitations in SWAT plant module. I have several major concerns about this work. First, writing of this work should be further improved to make it publishable. Second, organization of the introduction and method sections should be changed following requirement of a scholarly journal.

*Response: We appreciate the anonymous referee the constructive and valuable comments. We agree on the need of further improvement on the write up and organization. Therefore, we have substantially improved the write up and the organization in the revised manuscript.*

line 47, what does this mean? do you mean that SWAT could not represent changes deciduous forest?

*Response: No, we are referring to the mismatch between the SWAT dormancy period and the dry season, where dormancy is a function of water and temperature stress. According to Wagner et al. (2011) the SWAT simulated LAI for deciduous forest in India is not realistic because the dormancy in SWAT is related to daylength and latitude. As a result the authors shifted the dormancy period in SWAT to the dry season months and hence improved the LAI simulation. As pointed by referee #2, we have further improved this text in the revised manuscript (lines 50-52).*

line 55, extra space.

*Response: removed.*

line 58, for -> of. To simulate

*Response: Modified accordingly.*

line 59, a rainy season

*Response: Updated.*

line 67-68 in this sentence should be moved to the method section

*Response: We agree with this suggestion. We have moved that to the methodology section in the modified manuscript.*

line 71-81, I expected to see objectives of this work, but the authors are describing their methodology, which should be moved to the next section (method)

*Response: The authors agree with the referee. We have clearly stated objective in the revised manuscript (line 74-77) that reads as "..The objective this study is to improve the vegetation growth module of SWAT model for trees and perennials in the tropics. Towards this the use of the SMI within a predefined transition months as a dynamic trigger for new vegetation growth cycle will be explored. The modified SWAT (SWAT-T) model will be evaluated using 8-day MODIS LAI and Remote sensing-based ET. Additionally, the model will be evaluated using observed daily streamflow. "*

line 92, a long rainy season

*Response: Updated.*

line 93, across what?

*Response: across the basin. We have added this in the revised manuscript.*

section 2.2.1 was copied from the SWAT manual. I suggest to condense this part significantly, or move it to a supplementary section

*Response: This is a summary of the vegetation growth module in SWAT based on the manual and published literatures. As suggested, we have considerably reduced the summary of the SWAT vegetation growth module in the revised manuscript (lines 135-165).*

line 172-173, this is not correct. Kill and dormancy are totally different. If some one use this to regulate phenology, they must have made mistakes.

*Response: We agree that dormancy and kill are different concepts in SWAT. Dormancy is a function of daylength and latitude, and dormancy starts at the shortest day of the year for trees and perennials in tropics. Whereas, the kill operation in SWAT stops plant growth at a specified time using either date calendar or fractional potential heat unit (Nietch et al. 2011). The default management setting in SWAT has planting/beginning of the growing season and kill/end of the growing season at 0.15 and 1.2 $FR_{PHU}$, respectively for trees and perennials. The management setting are important for several agroforestery operations..*

line 175-181, these discussions should be presented in your Introduction, or the Discussion section

*Response: We agree on this suggestion. We excluded part of this paragraph in the revised manuscript.*

line 183, what are the data source of P and PET?

*Response: The P is based on historical local gauge observations in and around the basin. PET is based on observation based global PET from Trabscaou and Zomer (2009). We have added the sources in the revised manuscript (line 187).*

line 192-193, remove these two sentences

*Response: We disagree with the referee. These sentences are important because in larger basins there could be variation in climate. As a result, there could be difference in seasonality of rainfall across the watersheds, thereby SOS changes.*

line 201-207, do you have any reference to support your rules?

*Response: the fundamental rule for the start of new growing season is the SMI (i.e. P/PET). When the SMI exceeds a user defined threshold a new growth cycle is triggered within a predefined period annually. This concept is somewhat similar to the Water Requirement Satisfaction Index (WRSI), which is a ratio of actual evapotranspiration to PET (Verdin and Kalver 2002). The threshold for the SMI to trigger new growing season is set to 0.5, meaning the rainfall satisfies 50% of the atmospheric water demand (PET). Even though this threshold could vary from place to place, growing season in general is defined as the time when average precipitation is greater than half of the average PET (Mcnally et al. 2015).*

line 238, a... site?

*Response: The 8-day MODIS LAI has 1000 m spatial resolution and therefore to avoid land –cover class mix during aggregation of pixels, we used selected sample sites (shown in Figure 1b) to mask the MODIS LAI for each representative land-cover classes. We believe such approaches improve the reliability of the LAI estimates for each representative land-cover types.*

line 242, why lai of 1.5 is removed?

*This is based on the long term LAI timeseries during the rainy season (i.e. the peak growing season), whereby the LAI values are above 2.0 $m^2/m^2$. Therefore, LAI values during the rainy season below 1.5 are replaced with interpolated values. We have provided this information in the revised manuscript (line 233).*

line 245, what break?

*Response: This break is referring to unrealistically low LAI values due to noise and cloud contamination. Sudden break in LAI could also happen due to anthropogenic effects (land use change. Fire, etc.).*

line 253-254, I donot understand how the LAI patterns match precipitation.

*Response: For the dominant part of the basin, the long rainy season is from March to May with a peak in April. Also, the basin gets short rain from October to December. On the other hand, we note the LAI seasonal pattern, whereby the lower LAI values are observed in the dry months (July to Sept). This is supported by a correlation of 0.66 seasonally in the humid part. This is discussed further in the seasonality of LAI and its association with rainfall in the revised manuscript (lines 411-416).*

line 259-260, awkward expression. consider to improve

*Response: We have improved this in the revised manuscript (lines 253-254) that reads as "ET is one of the major components in a basin water balance, which is influenced by the seasonal vegetation growth cycle."*

line 270, change the term 'flow'to stream flow or river discharge through out the manuscript

*Response: Modified accordingly.*

line 272-273, remove the second period

*Response: Removed.*

line 285, seasonality of what?

*Response: LAI is missing here. We have included this in the revised manuscript (line 404).*

line 290-291, I am not convinced that lai reflects changes in rainfall. you need to provide some statistical information here. And what about the correlation between temperature and lai?

*Response: There is a fair association between seasonal rainfall and LAI in the humid part with correlation up to 0.66. The correlation (0.81) is even stronger for the lower part of the basin due to the clear seasonality of rainfall, albeit with one month lag.  Since temperature is not a limiting factor on vegetation growth in tropics, we did not consider that in our analysis.*

line 306-308. very confusing

*Response: The minimum LAI for a land cover could vary inter-annually depending on the climatic condition. However, this minimum LAI for each land cover need to be provided in SWAT plant database, meaning the minimum LAI during dormancy for a specific land cover does not change inter-annually and hence it is constant.*

line 311 figure 5 only show two land covers. What about the other two?

*Response: The plot for Tea and RNGB are excluded since the observed patterns  are somewhat similar to FRSE and RNGE, respectively.*

line 316. I do not see seasonal variation from figure 5

*Response: Figure 5 depicts the seasonal pattern. The seasonal variation is depicted as the range between the maximum and minimum LAI values. In this regard note that the range in LAI that is normalized with the annual average MODIS LAI is more considerable (up to 82%) for grass and shrub cover types.*

line 325, in October

*Response: Updated accordingly.*

line 328, was, first

*Response: Modified.*

 line 329, second

*Response: Modified.*

line 330, a 8-day scale

*Response: updated accordingly.*

line 339-341. do not understand what does this mean

*Response: We have further improved the clarity in the revised manuscript (lines 326-332) that reads as "With this management setting, the simulated LAI is zero at the beginning of each simulation year for all types of vegetation cover. Mwang et al. (2016) improved the SWAT LAI simulation with this management setting using $FR_{PHU}$ of 0.001 to start the growing season and minimum LAI of 3.0 for evergreen forest. Yet, this change is region specific and cannot be transferred. As shown in Figure 4 and Figure 5, this can also be partly improved using a date scheduling (Date) for the start and the end of the vegetation growth cycle (i.e. instead of heat unit). Additionally, all the management setting can be removed (no mgt) and vegetation is growing since the start of the simulation (i.e. IGRO=1)."*

line 344, consider to revise

*Response: We have modified this in the revised manuscript.*

line 351-353, not clear. consider to revise

*Response: We have revised this in the modified manuscript (lines 352-356)*

line 361, I do not see improvement in runoff

*Response: we did show improvements in simulated LAI but we highlighted the SWAT-T is able to reproduce the observed streamflow.*

line 387-389, remove this sentence

*Response: We have revised this paragraph.*

line 396-406. It is surprising that the authors did not evaluate SWAT ET simulations

*Response: We in fact evaluated the SWAT-T simulated ET against ET-RS at 8-day. Since SWAT is not a fully distributed and gridded model, we have not done a pixel level spatial evaluation. Nevertheless, we have depicted the agreement between LAI and ET qualitatively using one rainy month and one dry month from year 2002 at HRU level. Also, we have shown the temporal dynamics of ET using 8 years average monthly ET.*

line 412. Is this a sentence?

*Response: We have updated this sentence in the revised manuscript (lines 478-479).*

---

## Author Comment (AC3) · 22 Jun 2017

**Response to Anonymous Referee #2**

We thank the referee for this thorough review and for the numerous constructive suggestions. We agree with the need of further improvements in the manuscript organization. We therefore have addressed all the comments below point-by-point (italics). We have attached as supplementary material the revised manuscript with changes in blue.

The authors develop a new vegetation growth module for tropical ecosystems in SWAT. In particular, they use a soil moisture index to initiate a new growing cycle within two pre-defined months. They evaluate the growth module with regard to LAI, ET, and river discharge with satisfactory results. The topic is of current scientific interest, as several authors have previously outlined that the default vegetation growth for e.g. forests in SWAT is not applicable in the tropics. The manuscript is mostly well prepared. However, the paper would benefit a lot if it was more structured according to the evaluation of the vegetation growth module. In particular, this applies to the results and discussion part. This focus should be set very clearly in a revised version. Moreover, some parts of the manuscript require further, more detailed, or more precise information. The provided comments should be addressed before accepting this manuscript for publication.

*Response: We thank the anonymous referee for his/her time and the valuable comments and suggestions. We have addressed properly all the comments and suggestions in the revised manuscript (in blue). Below we provided our point-by-point response (in italics).*

General comments:

1) The authors should take a decision on the aim of the manuscript. Do they aim at providing an improved plant growth module for SWAT in the tropics? Or do they aim at showing their adjustment of the model to a specific catchment? Currently, the title reads like the first is the case. However, in many parts the paper reads like the latter is the case. For the first aim the authors need a stronger focus on evaluation of the plant growth model and they need to program the module as flexible and transferable as possible. Right now there are parts that are not immediately transferable to other catchments. The authors are encouraged to sharpen the paper with regard to the first aim. However, if this was not their aim they may also go for the second aim and adjust the title accordingly.

*Response: This is an interesting point, thank you! The main goal of this manuscript is to present and demonstrate a methodology on an improved SWAT vegetation growth module for tropical condition. The modified growth module can be applied anywhere in the tropics and a user can also add region specific information such as the transition months ($SOS_1$ and $SOS_2$), the number of days for rainfall (P) and reference evapotranspiration (PET) aggregation to compute the soil moisture index ($SMI = P/PET$) and the minimum SMI threshold for triggering new growth cycle within a predefined period. We therefore have sharpened our revised manuscript in this regard.*

2) Clear separation of calibration and validation period is required. This also applies to

the calibration of plant parameters. 3) A clear calibration and validation strategy regarding the different parameters used for calibration and validation is needed. E.g. right now, it is not clear which parameter was calibrated first and why?

*Response: For the sake of convenience we combined the responses for comments 2 and 3. As noted by the referee, the calibration and evaluation approach was not stated clearly in the manuscript. The revised manuscript has now included a dedicated section (section 2.8.2 lines 290-305) that elaborates on the calibration and evaluation approach.*
*"...The main purpose of this study is to explore the potential of the SMI to trigger new vegetation growth cycle for the tropical ecosystem within a predefined period annually. We initially evaluated the effects of the vegetation growth module modification by comparing against the standard SWAT model growth module with varying management settings. This analysis involved uncalibrated simulations of the SWAT models with the default SWAT model parameters, meaning the models differs only with how vegetation growth is simulated. It is worth noting that the aim of these simulations is mainly to expose the inconsistencies in the vegetation growth module structure. Afterwards, we calibrated the parameters related to the simulation of the LAI, the evapotranspiration and the streamflow manually by trial-and-error and expert knowledge for the SWAT-T model. Firstly, SWAT parameters that control the shape, the magnitude and the temporal dynamics of LAI were adjusted to reproduce the MODIS LAI at 8-day for each land cover classes. Then, we adjusted parameters that mainly control streamflow and evapotranspiration (ET) simulation simultaneously using the daily observed streamflow and 8-day ET-RS. Perhaps, the manual adjustment may not be as robust as an automatic calibration as the latter explores a larger parameters space. However, the manual calibration is sufficient to illustrate the impact of the modification on the vegetation growth cycle and its effect on the water balance components. The SWAT-T model calibration and validation was done for 2002-2005 and 2006-2009, respectively."*

4) LAI is prescribed based on satellite data (l.256). Why is this necessary? This makes the model less flexible and non-transferable to other catchments without following a similar approach.

*Response: The results from LAI prescription is not shown in the manuscript. Therefore, we have removed in the revised manuscript.*

5) The third chapter "results and discussion" is sometimes hard to understand as discussion of model parameters and results is mixed with model validation. I strongly suggest reworking the structure and separating the "results" from the "discussion" part.

*Response: We accepted this suggestion. As a result, we substantially restructured the "results and discussion" part in the revised manuscript (line 314-483). Briefly, we have presented uncalibrated LAI simulation results from SWAT growth module with and without modification. The purpose of this section is to highlight the limitations of LAI simulation with the existing SWAT vegetation growth module and the added value of the new SMI based modifications. Afterwards, we have presented calibration and evaluation results for LAI, ET and streamflow are presented back-to back using SWAT-T model.*

6) Why is it necessary to prescribe the two month in which the growing season starts?

*Response: This is interesting question, thank you! The two months are assumed to represent the transition from the end of the dry season to the beginning of the rainy season. These months are determined based the climatological rainfall (P) and reference evapotranspiration (PET) data. The main purpose of these months is to avoid false starts during the dry season short rainfall episodes as well as during the rainy season short dry spells. These months are in fact defined a priori and varies geographically depending on the climate.*

7) It would be very good, if you could validate the modeled begin of the growing season using independent data. Is there any data that you have available to do this?

*Response: In fact, the SOS dates are mainly controlled by the SMI variations and the effect of setting the transition months a priori is rather minimal given the season change is not immediately occurring with the start of the first transition month (.i.e. $SOS_1$). This dates can be verified with SOS dates extracted from remote sensing-based NDVI timeseries. However, we believe it is sufficient to show the simulated inter-annual dynamics of the SOS dates for two reasons: i) Since the SWAT-T model is calibrated against MODIS LAI, using the NDVI derived SOS dates may not be considered as independent data and ii) our study area is a typical data basin and hence such detailed verification would be more interesting in a basins with better forcing data. Therefore, we acknowledge the need of further research in this regard. We have added in the revised manuscript the need of verification of the SOS dates (line 427-428) that reads as"…..Yet, we acknowledge the need of further verification studies in basins with sufficient forcing data and field measurements."*

Line specific comments:
l.7: SWAT is a hydrologic model. The term "simulator" is not very common for SWAT in the literature. Suggest to replace this by "model" in the whole manuscript.

*Response: Modified accordingly throughout the revised manuscript.*

l.13: "uses of a simple..." Please improve the language.

*Response: We have improved the language in the revised manuscript that reads as (line 12-13) "…..we present a modified SWAT version for the tropics (SWAT-T) that uses a straightforward but robust soil moisture index (SMI)…".*

l.15: Would be good to include information here, how the dry season is defined.

*Response:  Given the word limit in the abstract, we could not include extra information on how the transition months are defined. However, we have further elaborated the rationale on how to determine the transition months ($SOS_1$ and $SOS_2$) using long-term climatological P and PET data in the revised manuscript (line 180-205). In short, the dry season is defined based on climatological P and PET data, whereby the PET is considerably higher than the P (shown in Figure 2 page 8).*

l.18: "flow" – The authors probably refer to stream flow. Should be more precise throughout the manuscript.

*Response: Updated accordingly throughout in the revised manuscript.*

l.19: Please include information, which RS-ET was used.

*Response: We have slightly modified the text to reflect the type of ET source in the revised manuscript (line 18)".... a thermal-based evapotranspiration (ET-RS) estimate……".* Nevertheless, *We have provided a brief description in the data in section 2.5 (line 254-264) about the ET-RS data based on Alemayehu, T., Griensven, A. van, Senay, G. B. and Bauwens, W.: Evapotranspiration Mapping in a Heterogeneous Landscape Using Remote Sensing and Global Weather Datasets: Application to the Mara Basin, East Africa, Remote Sens., 9(4), 390, doi:10.3390/rs9040390, 2017.*

l.20: "could be: : :" Please be more precise. In which situations is it useful?

*Response: We agree with the referee suggestion. We have updated this in the revised manuscript with information about the applicability of the tool (line 19-20) that reads as "The SWAT-T model with the proposed improved vegetation growth module for tropical ecosystem can be a robust tool for simulating the vegetation growth dynamics consistently in hydrologic model applications including land use and climate change impact studies."*

l.44: Please be more precise, i.e. "dormancy, which is defined as a function of daylength and latitude".

*Response: Thank you. We have updated this in the revised manuscript.*

l.47: As I read it, they do not report a shift, but shifted the dormancy period to a prescribed dry season (see p.1786). Please improve the statement.

*Response: Thank you for spotting this. We have corrected this in the revised manuscript (line 50-52) that reads "Likewise, Wagner et al. (2011) reported a mismatch between the growth cycle of deciduous forest in the Western Ghats  (India) and the SWAT dormancy period, and they subsequently shifted the dormancy period to the dry season. "*

l.49-55: You are reviewing tropical regions. However the Kalahari has a subtropical climate. Please improve.

*Response: We disagree on this comment with the referee. Jolly and Running (2004) used two site Maun and Tshane sites, respectively located at $-19.93^0$ and $24.17^0$ latitude to evaluate the BIOME-BGC simulated phenological development. The authors reported tropical climate for the study area (P.307 reference in the manuscript).*

l.73: "phonological"

*Response: Corrected in the revised manuscript.*

l.67-77: These lines include a lot of information on methodology. Please shift the methodological parts to the methodology section.

*Response: We agree with this suggestion, thank you. We have moved that to the methodology section in the modified manuscript.*

l.103: "SWAT uses a GIS based interface". Not precise. You can use GIS to prepare input files for SWAT. Please improve.

*Response: We have rephrased this in the revised manuscript.*

l.126 following: Please add citations for formulas.

*Response: We have added Neitsch et al. ( 2011) in the revised manuscript.*
l.128: Grammar.

*Response: We have corrected this in the revised manuscript.*

l.134: "endo"

Response: Thank you for spotting this. We have corrected this.

l.170-173: This passage is not quiet to the point. SWAT does not offer heat unit scheduling to solve the issue of plant growth in the tropics. In fact, both scheduling options will not help, as long as the temperature dependant dormancy period is still activated. Please improve.

*Response: We agree with the referee suggestion. We have further improved this in the revised manuscript (line 173-176). This reads as ".....SWAT offers several management settings for the start and the end of growing season based on either heat units (the default) or calendar date scheduling. The default management setting in SWAT is scheduled using heat unit fractions, whereby planting (start of growing season) and kill (end of growing season) occur at $FR_{PHU}$ values of 0.15 and 1.2, respectively. In fact, the limitation with plant growth dynamics cannot be solved using SWAT management settings as far as the latitude and daylength dependent dormancy is activated...... "*

l.190-197: How are SOS1 and SOS2 defined? By a threshold, or by the increase of the SMI? Are they set by the user? If they are set by the user, the model is not as flexible – is this necessary? It should be highlighted in the other parts of the manuscript that the start of the growing season is not fully dynamic but triggered within a pre-defined period.

*Response: We appreciate these important questions from the referee. The transition months (i.e. $SOS_1$ and $SOS_2$) indicate the end of the dry season and the beginning of the rainy season. These months should be determined using the climatological monthly P and PET ratio (i.e. the SMI). In principle during the dry season months the SMI values are low since the PET exceeds the P considerably. In contrast, during the rainy months the SMI values are relatively higher compared to the dry months. Therefore, the user should select the transition months guided by the climatological SMI values. We acknowledge some degree of subjectivity in fixing the months and yet, the climatological transition months from the dry to the rainy season are often known. The aim of fixing the $SOS_1$ and $SOS_2$ is to avoid false starts during the dry season due to short spell rainfall episodes. The new growth cycle is triggered dynamically when the SMI exceeded and/or equaled a user defined threshold within these pre-defined months. We have further clarified this in the revised manuscript.*

l.211: Please add a reference for the DEM (also in table 1 & add time period for river discharge in table 1).

*Response. We have provided this information in the revised manuscript.*

l.217: Please add a reference for the SWAT land use codes, so that non-SWAT users can look these up.

*Response: We have provided reference in the revised manuscript.*

l.222: Abbreviation "TMPA" is not explained at first mentioning. Please improve.
*Response: Updated in the revised manuscript.*

l.226-27: Please add some (short) reasoning for this adjustment, so that the reader can understand the idea of this approach without reading the referred paper.

*Reponse: Thank you for the suggestions. We have provided information on the adjustment in the revised manuscript. The new text read as (line 286-289) "....To improve the spatially and temporally consistency of the PET estimates, we adjusted at sub-basin level the solar radiation for each month by comparing against observation based, long-term (1950-2000) seasonal PET (Trabucco and Zomer, 2009), which is similar to the method suggested in Alemayehu et al. (2017)."*

l.237: Which forest biomes? What about others?

*Response: This is referring to the validation study on MOD15A2 LAI at Budongo Forest (Uganda) and Kakamega Forest (Kenya) by Kraus (2008). This study did not include other biome types.*

l.238: What do you mean by "land cover mix"? l.238: Grammar. l.238: Please add the sizes of the homogenous sites.

*Response: land cover mix is referring to the mix of different land cover classes (i.e. forest, grassland...) within 1000m grid resolution of MODIS LAI. We have further improved this part and provided more*

*information in the revised manuscript (line 228-231). This read as "…We selected a representative relatively homogenous sample sites (i.e. polygons) for evergreen forest (174 km², tea (123 km²), savanna grassland (136 km²) and shrubland (130 km²) (see Figure 1b) using the Africover classes and Google Earth images. This is useful to reduce the effect of land cover mix while averaging coarse scale (i.e. 1 km) LAI and hence improve the reliability of the LAI timeseries. "…*

l.242: Please provide reasoning for selecting the threshold value 1.5.

*Response: This is based on the long term LAI timeseries during the rainy season (i.e. the peak growing season), whereby the LAI values are above 2.0 m²/m². Therefore, LAI values during the rainy season below 1.5 are replace with interpolated values. We have provided this information in the revised manuscript (line 233).*

l.242: Are these gaps resulting from the previous masking?
*Response: The gaps are mainly due to cloud contaminations during April mostly.*

l.243-245: Sentence not clear. Please improve.

*Response: We have further improved this part in the revised manuscript (line 235-237).*

l.266: Is this measured NDVI or remote sensing based? If it is remote sensing derived, are the two products independent from each other?

*Response: The NDVI is referring the MODIS product. The thermal-based ET from Alemayehu et al. (2017) is independent of the MODIS NDVI data. Alemayehu et al. (2017) estimated ET using mainly MODIS land surface temperature based on the Operational Simplified Surface Energy Balance (SSEBop) (Senay et al. 2013) algorithm.*

l.271-273: Please include how large the gauged headwater area is. Also, add information on when the gaps happen, e.g., at similar times in the year or mainly in one year?

*Response: We have provided this information in the revised manuscript (line 266).*

l.275: This is not precise and could be misleading. As I understand it, the SMI triggers the growing season within a predefined period of 2 months. Please improve.

*Response: Right, the SMI is used to indicate (i.e. trigger) the start of the new growing cycle within a predefined period. We have further clarified this throughout the revised manuscript.*

l.275-281: The model calibration and validation strategy is not clear. The authors use stream flow, LAI and ET. However, it is not clear which parameter is used first – or are they combined? Please improve this section.

*Response: We have addressed this properly and we provided the response for major comments 2 and 3.*

l.287: Peaks in April and August are not shown in Figure 4. Please clarify.

*Response: We agree with the referee. Even though the LAI magnitudes are relatively high in April and August, they are not considerably high compared to the rest of the rainly months. This is mainly due to ample rainfall distribution throughout the year. We therefore further improved this in the revised manuscript.*

l.291: Please add the "drier months" you are referring to in brackets.

*Response: We have modified this accordingly in the revised manuscript.*

l.285-299: Why are you showing tea in Fig. 4? It is shown but not mentioned here.

*Response: The tea and the forest are located in the humid and mountainous part of the basin. Therefore, we note comparable seasonal LAI dynamics and hence presented results for forest only. We have updated this in the revised manuscript.*

l.302: Again, this is not precise and could be misleading. As I understand it, the SMI triggers the growing season within a predefined period of 2 months. Please improve.

*Response: We have updated this accordingly in the revised manuscript.*

l.308-310: The authors use long-term MODIS LAI to parameterize the model. It would be much better if these values were derived from the calibration period, so that calibration and validation data are strictly independent. Even though, I do not expect a pronounced change in LAI values in calibration and validation period, I would recommend to only use data from the calibration period for model setup.

*Response: This suggestion seems a misunderstanding. We used the MODIS LAI timeseries from 2002-2005 for the calibration and the rest (2006-2009) for validation. Only the minimum LAI and the Potential Heat Unit (PHU) are adjusted based on long-term values and the remain SWAT parameters related to vegetation growth are adjusted by comparing the with the 8-day LAI time series. As mentioned earlier, we substantially restructured the results and discussion part in the revised manuscript.*

l.313: Not clear why the authors express the amplitude of simulated LAI as a percentage of the average annual MODIS LAI. Please clarify your validation strategy.

*Response: We think the way the manuscript structured created this confusion. As we stated already, the results and discussion has been restructured substantially. We have clarified in the revised manuscript (line 407-419) that the average seasonal pattern of LAI was computed using the calibrated SWAT-T simulation and MODIS LAI from 2002-2009. The average seasonal LAI amplitude is the difference*

*between the peak and the trough LAI values (i.e. the range) and we normalize this value with the mean annual MODIS LAI.*

l.316: Why are simulated and remote sensing based LAI not directly compared and shown as later presented in figure 10? That is what the reader would expect at this point. A scatter plot is also useful in this context.

*Response: We agreed with the referee suggestion. We have plotted together the seasonal LAI for from MODIS and SWAT together. We have also included a pooled scatter plot of MODIS and SWAT-T simulated seasonal LAI for FRSE,Tea, RNGE and RNGB in the revised manuscript.*

[Figure]

**Figure 8  The long-term (2002-2009) average seasonal LAI pooled scatter plot (left) and temporal dynamics (right).  FRSE: evergreen forest; RNGE: grassland; RNGB: shrubland.**

 l.320-323: This passage can be shifted to a discussion part.

*Response: We think discussing the season change dates after LAI seasonality is appropriate.*

l.323-325: This is only an effect of your modeling. Can you use this for validation? E.g. by comparing to independent data (e.g. satellite derived) on the beginning of the growing season?

*Response: Addressed in the general comment 7 response.*

l.337: But with a correct definition of ALAI_MIN it would not be 0, right? Please make clear, what you are validating. This improvement is due to satellite-based improvement of the parameter ALAI_MIN and not because of the improvement of the plant growth module.

*Response: Yes, correct ALAI_MIN value to the SWAT plant database would not solve this problem. Yet, the zero LAI values, for instance the forest (FRSE), as simulated in SWAT is not due to wrong prescription of ALAI_MIN in the plant database, rather due to the management settings. The ALAI_MIN is used only when dormancy occurs.*

l.332-341: You need to explain your different model setups in more detail. E.g. IGRO=1, will not be understandable for non-SWAT users. Moreover, please use only meaningful model parameterizations. It does not make sense to compare SWAT-T to a model that does not work properly. Many different model setups are irritating. I would suggest to compare SWAT-T to the best possible model parameterization achieved without code changes.

*Response: We agree on the fact that comparing different SWAT model LAI simulation with varying management setting and vegetation growth module is not appealing. Nevertheless, such uncalibrated comparisons of LAI simulation is important in shedding light on the inconsistencies of SWAT growth module for trees and perennials in the tropics. To clarify further, the uncalibrated LAI simulations are outputs from the same SWAT model with four different settings: 1) standard SWAT vegetation growth module with the default management settings based on heat unit scheduling*
*    2) standard SWAT vegetation growth module with the default management settings*
*      based of calendar date scheduling*
*    3) standard SWAT vegetation growth module with no management setting*
*    4) with the modified vegetation growth module with no management setting*
*The first thee option, of course, do not require a code change. Note that there is no difference in the SWAT model parameters for this comparison, their differences are mainly due to the management setting and the modification in the vegetation growth module. We have further sharpened this in the revised manuscript.*

l.342-347: Not sure, why this paragraph is provided here. Figure 5 was already discussed before. Moreover, the comparison to an uncalibrated model is not a fair evaluation (see comment above).
*Response: The aim of this paragraph is to provide supportive information on rainfall distribution in the basin, which is the dominant factor in vegetation phenology. The redundancy with Figure 5 has been improved with the new rearrangement of the results and discussion part in the revised manuscript.*

l.352: "standard SWAT" Please define this and provide a model evaluation for this setup. If this model does not work it is not useful for comparison.

*Response: The "standard SWAT" model is referring to SWAT 2012 revision 627 as stated in line 272 in the revised manuscript. As discussed in the earlier comments, the uncalibrated model comparison is to minimize the effect of model parameters in highlighting the inherent limitations with the vegetation growth cycle in tropics.*

l.353-354: "better realism". Please improve the language.

*Response: We have improved the language in the revised manuscript.*

l.354: Did you test for significance?

*Response: No significance test but that is to show the considerable reduction in zero potential transpiration. We have changed the wording in the revised manuscript.*

l.368: Quantify where possible.

*Response: We have improved the result presentations in this part using bias and correlation values in the revised manuscript.*

l.363-369: This short paragraph presents some of the most valuable results. Please provide further details here. E.g. in which periods and why do MODIS LAI and simulated LAI not match well, as shown in Fig. 10?

*Response: As pointed by the referee, this section is indeed the important part of the manuscript. We have further provided information on the results in the revised manuscript (line 368-3888).*

l.374-375: It is hard to see from Fig. 11 whether ET values match well as the lines overlap. Please add a scatter plot.

*Response: Thank you for the suggestion. We have improved the figures in the revised manuscript (shown below).*

[Figure]

***Figure 10 The comparison of remote sensing-based evapotranspiration (ET-RS) and SWAT-T simulated ET (ET-SWAT-T) aggregated per land cover classes. Note that for SWAT-T HRU level ET is aggregated per landcover. The vertical black line marks the end of the calibration period and the beginning of the validation period.***

l.384: Contradiction to the previous sentence. Do you mean "grassland" instead of "forest"?

*Response: Thank you for spotting this. This has been corrected in the revised manuscript.*

l.385: But Figure 5 shows a seasonality for both? This is contradictory to the previous sentences. Please clarify.

*Response: Thank you again for pointing this out. We have clarified this in the revised manuscript (line 443-444).*

l.387-395: Again the structure is not clear. The authors evaluate parameter values at this point. Why is this needed and presented here? Please provide some justification or remove the paragraph.

*Response: The purpose of this paragraph is to provide information on the calibrated SWAT parameters and the higher water use (i.e. ET) by FRSE compared to the other land cover classes. We have moved up this to the first part of section 3.2.3. (Page 20, line 431-436) in the revised manuscript.*

l.404: Sentence not clear. Please improve.

*Response: We have rephrased this in the revised manuscript.*

l.405: Please quantify the spatial variability.

*Response: Since SWAT assumes a uniform, single plant species community per land cover class, thus we do not expect substantial spatial heterogeneity. Therefore, we have provided a qualitative visual information using spatial maps on LAI and ET as shown in Figure 11 (page 22).*

l.396-406: Why do you not compare the spatial distribution of simulated ET and LAI to the spatial distribution of MODIS based ET and LAI? That could be another valuable comparison that might be more useful than a presentation of modeled values.

*Response: This is a good point. However, since SWAT is not a fully distributed model and the HRUs are not square grids direct comparison will not be effective. Furthermore, getting cloud free MODIS LAI for the whole study area is not feasible. Therefore, the aim of the graphs two show LAI and ET spatial variation (qualitatively) for one rainy month and one dry month.*

l.417-452: The conclusion should be shortened so that it only includes the most important conclusion drawn from your study.

*Response: Thank you for the suggestion. We have shortened the conclusion in the revised manuscript.*

l.432: This sentence is misleading ("default parameters"). As I understand your setup

SWAT-T parameters were calibrated. Please clarify.

*Response: To demonstrate the modification in the plant growth module, the comparison were shown for simulations with default SWAT parameters (i.e. uncalibrated SWAT and SWAT-T results). Then the SWAT-T were calibrated and evaluated using MODIS LAI, RS-ET and observed streamflow. We noted in general that our manuscript need improvement in structures. Therefore, the revised manuscript will address clarity issues.*

l.434: What do you mean by "potential transpiration"? Potential evapotranspiration?

*Response: Potential transpiration is referring to the maximum transpiration rate by plants at optimal conditions, i.e. no water, temperature and nutrient stress. On the other hand, the potential evapotranspiration is defined as "the amount of water transpired by alfalfa, completely shading the ground, of uniform 40cm height and never short of water.*

l.435: The results with the other PET method were not shown. Please focus on what you have shown in this paper. If it is important, please include it, if not please remove.

*Response: We have excluded this from the conclusion of the revised manuscript.*
.
l.438: Misleading statement regarding SMI initiation of growing season, see also earlier comments. Please improve.

*Response: Improved.*

l.441: "conformed"

*Response: We have improved the language.*

l.447-452: This is more suitable for a discussions section. Please shorten or remove.

*Response: We removed this from the conclusion.*

l.451: "could be: : :" Please be more precise. In which situations is it useful?

*Response: Improved.*

l.451: "carbon fluxes" Not shown. Please remove.

Response: *Removed.*

---

## Author Comment (AC4) · 22 Jun 2017

**An improved SWAT vegetation growth module for tropical ecosystem**

Alemayehu T.[1,2*], van Griensven A.[1,2] and Bauwens W.[1]

[1]Vrije Universiteit Brussel (VUB), Department of Hydrology and Hydraulic Engineering, Brussel, Belgium
[2]IHE Delft Institute for Water Education, Department of Water Science and Engineering, Delft, the Netherlands

*Correspondence: t.abitew@un-ihe.org; Tel.: +31-621381512

**Abstract.** The Soil and Water Assessment Tool (SWAT) is a globally applied river basin eco-hydrological model in a wide spectrum of studies, ranging from land use change and climate change impacts studies to research for the development of best water management practices. However, SWAT has limitations in simulating the seasonal growth cycles for trees and perennial vegetation in the tropics, where rainfall (via soil moisture) is the major plant growth controlling factor than temperature. Our goal is to improve the vegetation growth module of the SWAT

model for simulating the vegetation variables such as the leaf area index (LAI) for tropical ecosystem. Therefore, we present a modified SWAT version for the tropics (SWAT-T) that uses a straightforward but robust soil moisture index (SMI) - a quotient of the rainfall (P) and reference evapotranspiration (PET) - to initiate a new growth cycle dynamically within a pre-defined period. Our results for the Mara Basin (Kenya/Tanzania) show that the SWAT-T

simulated LAI corresponds well with the Moderate Resolution Imaging Spectroradiometer (MODIS) LAI for ever- green forest, savanna grassland and shrubland, indicating that the SMI is reliable for triggering new growth cycle annually. The water balance components (evapotranspiration and streamflow) simulated by the SWAT-T exhibit a good agreement with a thermal-based evapotranspiration (ET-RS) estimate and observed streamflow. The SWAT-T

model with the proposed improved vegetation growth module for tropical ecosystem can be a robust tool for simu- lating the vegetation growth dynamics consistently in hydrologic model applications including land use and climate change impact studies.

**1. Introduction**

[revised manuscript text omitted]

SWAT model simulated soil moisture in the top soil layers with a certain minimum threshold after a defined dry season to indicate the start of a rainy season (SOS) and thus new vegetation growth cycle. Their results showed improvements in the SWAT simulated LAI seasonal dynamics and reproduced well the Moderate Resolution Imag- ing Spectroradiometer (MODIS) 8-day LAI. However, such approach requires calibrating the SWAT parameters for a realistic representation of the soil water balance dynamics often using observed streamflow. Recently, Yu *et al.*

(2016)  concluded uncertainty in soil moisture is significantly greater than streamflow simulations of a calibrated hydrologic model.

The objective this study is to improve the vegetation growth module of SWAT model for trees and perennials in the tropics. Towards this the use of the SMI within a predefined transition months as a dynamic trigger for new vegeta- tion growth cycle will be explored. The modified SWAT (SWAT-T) model will be evaluated using 8-day MODIS

LAI and thermal-based ET. Additionally, the model will be evaluated using observed daily streamflow.

**2. Materials and methods**

**2.1. The study area**

The Mara River, a transboundary river shared by Kenya and Tanzania, drains an area of 13,750 km$^2$ (Figure 1a).

This river originates from the forested Mau Escarpment (about 3000 m.a.s.l.) and meander through diverse agroeco- systems and subsequently crosses the Masai-Mara Game Reserve in Kenya and the Seregenti National Park in Tan- zania and finally feeds the Lake Victoria. The Amala River and the Nyangores River are the only perennial tributar- ies draining the head water region. The Talek River and the Sand River are the two most notable seasonal rivers stemming from Loita Hills.

Rainfall varies spatially mainly due to its equatorial location and its topography. The rainfall pattern in most part of the basin is bimodal, with a short rainy season (October-December) driven by convergence and southward migration of the Intertropical Convergence Zone (ITCZ) and a long rainy season (March-May) driven by southeasterly trades.

In general, rainfall decreases from west to east across the basin while temperature increases southwards. The Mara basin is endowed with significant biodiversity features through a sequence of zones from moist montane forest on the escarpment through dry upland forest to scattered woodland and then the extensive savanna grasslands (Figure

1b). Dark volcanic origin soils are common on the escarpment and rangelands while shallow soils that drain freely are found lower down. Poorly drained soils cover the plateau and the plains.

[Figure]

(a)

[Figure]

**Figure 1 Location of the Mara Basin (a) and its land cover classes (b). Note the sample sites location for the major natural vegetation classes that are used to mask the Moderate Resolution Imaging Spectroradiometer ( MODIS) Leaf Area Index (LAI).**

**2.2. SWAT model description**

The SWAT (Arnold et al., 1998, 2012; Neitsch et al., 2011) is a comprehensive, process-oriented and physically-based eco-hydrological model at a river basin scale. SWAT requires specific information about weather, soil proper-ties, topography, vegetation, and land management practices occurring in the watershed to directly model physical processes associated with water movement, sediment movement, crop growth, nutrient cycling, etc. In SWAT a
basin is partitioned into several sub-basins using topographic information and the sub-basins, in turn, are subdivided
into several Hydrological Response Units (HRUs) with a unique combination of land use, soil and slope class. Each
hydrologic processes are simulated at HRU level on a daily or sub-daily time step and aggregated into sub-basin
level for routing into a river network (Neitsch et al., 2011). SWAT considers five storages: snow, canopy storage,
the soil profile with up to ten layers, a shallow aquifer and a deep aquifer to calculate the water balance (Neitsch et
al., 2011) using the following equation:

$$\Delta S = \sum_{i=1}^{t} \left( P - Q_{total} - ET - Losses \right)$$

( 1 )

where $\Delta S$ is the change in water storage (mm) and $t$ is time in days. $P$, $Q_{total}$, $ET$ and $Losses$ are the daily amounts of
precipitation (mm), the total water yield (mm), the evapotranspiration (mm) and the groundwater losses (mm), re-
spectively. The total water yield represents an aggregated sum of the surface runoff, the lateral flow and the return
flow. In this study, the surface runoff is computed using the Soil Conservation Service (SCS) Curve Number (CN)
method (USDA SCS, 1972).

SWAT provides three options for estimating PET: Hargreaves (Hargreaves et al., 1985), Priestley-Taylor (Priestley
and Taylor, 1972), and Penman-Monteith (Monteith, 1965) (Neitsch et al., 2011). The model simulates evaporation
from soil and plants separately as described in Ritchie (1972). The potential soil evaporation is simulated as a func-
tion of PET and leaf area index (LAI) and further reduced with high plant water use while the actual soil water
evaporation is estimated by using exponential functions of soil depth and water content (Neitsch et al., 2011).
SWAT simulated LAI is also required to calculate the potential plant transpiration with formulations that varies
depending on the PET method selection (Alemayehu et al., 2015; Neitsch et al., 2011). The actual plant transpiration
(i.e. the plant water uptake) is reduced exponentially for soil water below filed capacity. Therefore, actual evapo-
transpiration in SWAT refers to the sum of evaporation from the canopy, the soil as well as plant transpiration.

In this study, we use the Penman-Monteith method (Monteith, 1965) to compute the PET for alfalfa (Neitsch et al.,
2011) as:

$$PET = \frac{\Delta.(H_{net} - G) + \rho_{air}.c_p.\left[e_z^o - e_z\right]/r_a}{\Delta + \gamma.\left(1 + {r_c}/{r_a}\right)}$$

( 2 )

where PET is the maximum transpiration rate (mm d$^{-1}$), $\Delta$ is the slope of the saturation vapour pressure-temperature
curve (kPa °C$^{-1}$), $H_{net}$ is the net radiation (MJ m$^{-2}$ d$^{-1}$), G is the heat flux density to the ground (MJ m$^{-2}$ d$^{-1}$), $\rho_{air}$ is
the air density (kg m$^{-3}$), $C_p$ is the specific heat at constant pressure (MJ kg$^{-1}$ °C$^{-1}$), $e_z^0$ is the saturation vapour pres-
sure of air at height z (kPa), $e_z$ is the water vapor pressure of air at height z (kPa), $\gamma$ is the psychrometric constant (kPa $^\circ$C$^{-1}$), $r_c$ is the plant canopy resistance (s m$^{-1}$), and $r_a$ is the diffusion resistance of the air layer (aerodynamic resistance) (s m$^{-1}$). The plant growth module simulates LAI and canopy height, which are required to determine the canopy and aerodynamic resistance.

**2.3. The vegetation growth and Leaf Area Index modeling in SWAT**

SWAT simulates the annual vegetation growth based on the simplified version of the EPIC plant growth model (Neitsch et al., 2011). The potential plant phenological development is hereby simulated on the basis of daily accu- mulated heat units under optimal conditions; however, the actual growth is constrained by temperature, water, nitro- gen or phosphorous stress (Arnold et al., 2012; Neitsch et al., 2011).

Plant growth is primarily based on temperature and hence each plant has its own temperature requirements (i.e.

minimum, maximum and optimum). The fundamental assumption in the heat unit theory is plants have a heat unit requirements that can be quantified and linked to the time of planting and maturity (Neitsch et al., 2011). The total number of heat units required for a plant to reach maturity must be provided by the user. The plant growth modeling includes simulation of the leaf area development, light interception and conversion of intercepted light into biomass assuming a plant species-specific radiation-use efficiency (Neitsch et al., 2011). The optimal leaf area development during the initial period of the growth is modeled (Neitsch et al., 2011) as:

$$fr_{LA\text{Im}x} = \frac{fr_{PHU}}{fr_{PHU} + \exp(l_1 - l_2 . fr_{PHU})} \tag{3}$$

where $fr_{LAImx}$ is the fraction of the plant's maximum leaf area index corresponding to a given fraction of potential heat units for the plant, , and $l_1$ and $l_2$ are shape coefficients. Once the maximum leaf area index is reached, LAI will remain constant until the leaf senescence begins to exceed the leaf growth. Afterwards, the leaf senescence becomes the dominant growth process and hence the LAI follows a linear decline (Neitsch et al., 2011). However, Strauch and Volk (2013) showed the advantage of using a logistic decline curve, to avoid that the LAI drops to zero before dormancy occurs. Therefore, we adopted this change to SWAT2012 whereby the LAI during leaf senescence for trees and perennials is calculated as (Strauch and Volk, 2013):

$$LAI = \frac{LAI_{mx} - LAI_{\min}}{1 + \exp(-t)} \tag{4}$$

$$with \ \ t = 12(r - 0.5) \quad and \quad r = \frac{1 - fr_{PHU}}{1 - fr_{PHU,sen}} \quad , fr_{PHU} \ge fr_{PHU,sen}$$

where the term used as exponent is a function of time and t varies from 6 to -6, *LAI* is the leaf area for a given day and declines using r as a decline rate, $LAI_{mx}$ and $LAI_{min}$ are the maximum and minimum (i.e. during dormancy) leaf area index, respectively, $fr_{PHU,sen}$ is the fraction of growing season (PHU) at which senescence becomes the domi- nant growth process.

As detailed in Neitsch *et al.* (2011), the daily LAI calculation for perennials and trees are slightly different, for the
latter the years of development is considered.

For perennials, the leaf on day *i* is calculated as:

$$\Delta LAI_i = \left(fr_{LA\,\mathrm{Im}\,x,i} - fr_{LA\,\mathrm{Im}\,x,i-1}\right)LAI_{mx}.$$
$$\left(1 - \exp(5.(LAI_{i-1} - LAI_{mx}))\right) \tag{5}$$

The total leaf area index, the area of green leaf per unit area of land, is calculated:

$$LAI_i = LAI_{i-1} + \Delta LAI_i \tag{6}$$

where $\Delta LAI_i$ is the leaf area added on day *i*, $LAI_i$ and $LAI_{i-l}$ are the leaf area indices for day *i* and *i-1* respectively,
$fr_{LAImx,i}$ and $fr_{LAImx,i-1}$ are the fraction of the plant's maximum leaf area index for day *i* and *i-1*, $LAI_{mx}$ is the maximum
leaf area index for the plants, $yr_{cur}$ is the age of the tree (years), and $yr_{fulldev}$ is the number of years for tree species to
reach full development (years).

**2.4. SWAT annual vegetation growth cycle limitation for the tropics**

SWAT assumes that trees and perennial vegetation can go dormant as the daylength nears the minimum daylength
for the year. Dormancy, which is a function of latitude and daylength during which plants do not grow, is used to
repeat the growth cycle each year for trees and perennials. At the beginning of the dormant period, a fraction of the
biomass is converted to residue and the leaf area index is set to the minimum value (Neitsch et al., 2011). In the
tropics, however, plants growth dormancy is primarily controlled by precipitation (Bobée et al., 2012; Jolly and
Running, 2004; Lotsch, 2003; Zhang et al., 2010; Zhang, 2005) and hence the standard SWAT growth module can-
not realistically represent the seasonal growth dynamics for trees and perennials. SWAT offers several management
settings for the start and the end of growing season based on either heat units (the default) or calendar date schedul-
ing. In fact, the limitation with plant growth dynamics cannot be solved using SWAT management settings as far as
the latitude and daylength dependent dormancy is activated.

**2.5. A soil moisture index-based vegetation growth cycle for the tropics**

As several studies demonstrated (Jolly and Running, 2004; Zhang, 2005; Zhang et al., 2006), the water availability
in the soil profile is one of the primary governing factors of vegetation growth in tropics. Thus, we propose a soil
moisture index (SMI) to trigger new growth cycle for tropical ecosystem in SWAT model within a predefined peri-
od. The SMI is computed as:

$$SMI = \frac{\sum\limits_{i=1}^{N} P}{\sum\limits_{i=1}^{N} PET} \qquad\qquad (7)$$

where P and PET denotes daily rainfall and potential evapotranspiration (mm d$^{-1}$), N is the number of days of aggre- gation. In this study we used five days aggregated P and PET (i.e. pentad) to determine the SMI to assure sufficient soil moisture availability to initiate new growth cycle. The SMI is somewhat similar to the Water Requirement Sat- isfaction Index (WRSI) (Verdin and Kalver 2002), which is a ratio of ET to PET.

Figure 2 presents the SMI seasonal pattern based on long-term climatological P for several gauge stations and PET

from Trabucco and Zomer (2009) across the Mara Basin. It is apparent from Figure 2 that the dry season (mostly from June - September) shows low SMI values (less than 0.5). Additionally, these patterns resemble well the long- term monthly average LAI for the savanna ecosystem (the dominant cover in the mid-section of the Mara Basin). In areas with a humid climate (i.e. the head water regions of the basin), the SMI values are high and the rainfall regime is different, yet in the relatively drier months (January and February) the SMI is low. As shown in Figure 2, the LAI

and the SMI seasonal dynamics match well with approximately one month lag, indicating the reliability of the SMI

as a proxy for the SOS and hence to trigger  the annual vegetation growth cycle. This approach enables SWAT

growth module not only to simulate the vegetation cycle dynamically within a predefined period, also avoids the need for management setting ("plant" and "kill").

[Figure]

**Figure 2 The climatological moisture index (SMI) derived from historical gauge observation across the Mara Basin and Trabucco and Zomer** (2009) **global reference evapotranspiration data. Leaf Area Index (LAI) for the savanna ecosystem (dotted line). SOS$_1$ and SOS$_2$ represent the start-of-rainy season (SOS) transition months to trigger growth.**

To avoid false starts during the dry season, the end of the dry season and the beginning of the rainy season ($SOS_1$

and $SOS_2$, respectively) are determined using a long-term monthly climatological P to PET ratio (Figure 2). For river basins with a single rainfall regime, a single set of SOS months can be used across the basin. However, in ba- sins with different rainfall regimes, different SOS months need to be set at sub-basin level. In our study area two distinct rainfall regimes are observed and therefore two different SOS values were needed. For the major part of the sub-basins October ($SOS_1$) and November ($SOS_2$) were used as transitions (Figure 2).

**2.6. SWAT-T: the adaptation of the SWAT plant growth module**

Based on the rationale elaborated in the preceding sections, we modified the standard SWAT2012 (version 627)
plant growth subroutine for basins located between $20^0$ N and $20^0$ S:

i)      If the simulation day is within $SOS_1$ and $SOS_2$ for a given HRU and a new growing cycle is not initiated
  yet, the SMI is calculated as the ratio of the pentad P to PET.

ii)     If the SMI exceeds or equals 0.5, a new growing cycle for trees and perennials is initiated. Subsequent-
  ly, $FR_{PHU}$ is set to 0 and the LAI is set to the minimum value. Plant residue decomposition and nutrient
  release is calculated as if dormancy would occur.

iii)    In case the SMI is still below the threshold (i.e. 0.5) at the end of month $SOS_2$, a new growing cycle is
  initiated immediately after the last date of $SOS_2$.

It is worth noting that SMI threshold could be raised or lowered depending on the climatic condition of the basin.

**2.7. Data for model evaluation**

*The Leaf Area Index*

The remote sensing LAI data used in this study is based on the MODIS TERRA sensor (Table 1). The LAI product
retrieval algorithm is based on the physics of radiative transfer in vegetation canopies (Myneni et al., 2002) and
involves  several constants (leaf angle distribution, optical properties of soils and wood, and canopy heterogeneity)
(Bobée et al., 2012). The theoretical basis of the MODIS LAI product algorithm and the validation results are de-
tailed in Myneni et al. (2002). Kraus (2008) validated the MOD15A2 LAI data at Budongo Forest (Uganda) and
Kakamega Forest (Kenya) sites and reported an accuracy level comparable to the accuracy of field measurements,
indicating the reliability of MOD15A2 LAI for evaluating SWAT simulated LAI for the study area.

Table 1 Summary of the inputs of the SWAT model and the evaluation datasets.

|  | Spatial/temporal resolution | Source | Description |
| --- | --- | --- | --- |
| Rainfall | 5 km / 1-day | Roy *et al.* (2017) | Bias-corrected satellite rainfall for Mara basin |
| Climate | 25 km / 3-hour | Rondell *et al.* (2004) | Max. and min. temperature, relative |
| Land cover classes | 30 m | FAO (2002) | Land cover classes for East Africa |

| | | | |
|---|---|---|---|
| DEM | 30 m | NASA (2014) | Digital elevation model |
| Soil classes | 1 km | FAO (2009) | Global soil classes |
| Discharge | Daily (2002-2008) | WRMA (Kenya) | River discharge at Bomet |
| ET | 1 km / 8-day | Alemayehu *et al.* (2017) | ET maps for Mara basin |
| MOD15A2 | 1 km / 8-day | LPDAAC(2014) | Global leaf area index |

We selected a representative relatively homogeneous sample sites (i.e. polygons) for evergreen forest (174 km$^2$), tea (123 km$^2$), savanna grassland (136 km$^2$) and shrubland (130 km$^2$) (see Figure 1b) using the Africover classes and

Google Earth images. This is useful to reduce the effect of land cover mix while averaging coarse scale (i.e. 1 km)

LAI and hence improve the reliability of the LAI timeseries. Subsequently, the MOD15A2 LAI was masked using the polygons of the sample covers and pixels with only quality flag 0, which indicates good quality, were used. Also, pixels with LAI values less than 1.5 during the peak growing months (i.e. period with LAI values mostly above 2.0)

were removed. Finally, we extracted the 8-day median LAI time series for each land cover for 2002-2009 and few gaps in the LAI time series were filled using linear interpolation. Notwithstanding with all the quality control efforts, due to the high variability and the inevitable signal noise, we noted breaks and high temporal variation in the LAI

[revised manuscript text omitted]

627) model and SWAT-T model. At this stage, the models were uncalibrated (i.e. based on default SWAT parame- ters). This is useful to explore the effect of the vegetation growth module structural modification on the consistency of simulated LAI annual cycle. We note from the simulation results considerable inconsistencies in the growth cycle of the simulated daily LAI mainly due to the vegetation growth model structure and management settings. For in- stance, Figure 4 and Figure 5 present the simulated daily LAI for FRSE and RNGE based on the standard SWAT

model under different management settings and the SWAT-T model. Strauch and Volk (2013), Kilonzo (2014) and

Mwang et al. (2016) reported similar observations. The default management setting in the standard SWAT model for starting the new growth cycle (i.e. planting) and ending the growth cycle is scheduled using the $FR_{PHU}$ (Heat unit). Thus, the start and the end of the vegetation growth cycle management settings occurs at $FR_{PHU}$ 0.15 and 1.2, respectively. With this management setting, the simulated LAI is zero at the beginning of each simulation year for all types of vegetation cover. Mwang et al. (2016) improved the SWAT LAI simulation with this management set- ting using $FR_{PHU}$ of 0.001 to start the growing season and  minimum LAI  of 3.0 for evergreen forest. Yet, this change is region specific and cannot be transferred.  As shown in Figure 4 and Figure 5, this can also be partly im- proved using a date scheduling (Date) for the start and the end of the vegetation growth cycle (i.e. instead of heat unit). Additionally, all the management setting can be removed (no mgt) and vegetation is growing since the start of the simulation (i.e. IGRO=1).

The forested head-water region experiences a unimodal rainfall regime, with March-August being the rainy season.

In contrast, a bimodal rainfall regime prevails (March-May and October-December) on the remaining part of the basin. Despite the changes in the management settings, it is apparent that the standard SWAT model has inherent limitation to simulate vegetation growth cycle for tropics that are consistent with seasonal rainfall distribution (Figure 4 and Figure 5). Also, the vegetation growth cycle resets annually on 28[th] June due to dormancy.

In contrast, the simulated LAI cycles for FRSE, tea, RNGE and RNGB  cover types using the SWAT-T model (i.e.

the modified vegetation growth module) reveal a consistent annual cycle and are associated with the seasonal rain- fall pattern (see Figure 4 and Figure 5).

[Figure]

**Figure 4 The LAI as simulated by the SWAT-T and the standard SWAT models for different management settings for**
**evergreen forest (FRSE) using default SWAT parameter. See management settings explanations in the texts.**

[Figure]

**Figure 5 The LAI as simulated by the SWAT-T and the standard SWAT models for different management settings for grassland (RNGE) using default SWAT parameter. See management settings explanations in the texts.**

**3.1.2. Implication of inconsistent LAI simulation**

The LAI is required to compute potential transpiration, potential soil evaporation and plant biomass, among others in SWAT (Neitsch et al., 2011). For instance, to compute the daily potential plant transpiration in SWAT, the canopy resistance and the aerodynamic resistance are determined using the simulated actual daily LAI and canopy height, respectively (Neitsch et al., 2011). Therefore, the aforementioned limitations of the annual vegetation growth cycle in the standard SWAT model growth module also influence directly the accuracy of transpiration. For instance, Figure 6 depicts the comparison of the standard SWAT and the SWAT-T simulated daily potential transpiration timeseries for grassland based on the Penman-Monteith approach. We observe 14% (12%) of the standard SWAT simulated daily potential transpiration timeseries (2002-2009) for FRSE (RNGE) being zero, suggesting a considerable inconsistency. However, the SWAT-T reduced considerably (i.e. less than 2% for FRSE and RNGE) the inconsistent zero daily potential transpiration, indicating the improvements in the vegetation growth module. Several studies have shown the effect of PET method selection in SWAT on simulated ET and other water balance components (Alemayehu et al., 2015; Maranda and Anctil, 2015; Wang et al., 2006). Alemayehu et al. (2015) reported significant differences in both potential and actual transpiration with the choice of PET method using calibrated SWAT model, which partly ascribed to the unrealistic LAI growth cycle. We notice the SWAT-T simulated potential transpiration is consistent regardless of the PET method selection in SWAT (results not shown here) and therefore, the improved vegetation growth module in the SWAT-T could reduce the uncertainty arising from the module structure and thus minimize the uncertainties in model simulation outputs.

[Figure]

**Figure 6 Inter-comparison of Penman-Monteith-based daily potential transpiration simulated by the SWAT-T and the standard SWAT models for grassland. Note that the heat unit scheduling is used in the standard SWAT model.**

**3.2. Evaluation of the calibrated SWAT-T model**

**3.2.1. Performance of the LAI simulation**

Table 2 presents the list of SWAT model parameters that are adjusted during the calibration process. Initially, the minimum LAI (ALAI_MIN) for each land cover classes were set based on the long-term MODIS LAI. Also, the PHU was computed using the long-term climatology, as suggested in Strauch and Volk (2013). The shape coefficients for the LAI curve ($FRGW_1$, $FRGW_2$, $LAIMX_1$, $LAIMX_2$ and $DLAI$) and the remaining parameters were adjusted during the calibration period by a trail-and-error process such that the SWAT-T simulated 8-day LAI mimics the MODIS 8-day LAI.

Figure 7 presents the comparison of 8-day MODIS LAI with the calibrated SWAT-T simulated LAI aggregated over several land cover classes for the calibration and validation period. We evaluated the degree of agreement qualitatively -by visual comparison- and quantitatively -by statistical measures. From the visual inspection it is apparent that the intra-annual LAI dynamics (and hence the annual growth cycle of each land cover class) from the SWAT-T model correspond well with the MODIS LAI data. This observation is supported by correlation as high as 0.94 (FRSE) and 0.92 (RNGB) during the calibration period (Table 3). As shown in Table 3, the model also shows similar performance during the validation period with low average biases and correlation as high as 0.93 (FRSE). Overall, the results indicate that the SMI can indeed be used to dynamically trigger a new growing season within a predefined period.

Despite the overall good performance of SWAT-T in simulating LAI, we observed biases for FRSE and Tea mainly during the rainy season over the calibration and validation period (see Figure 7 top row). This is partly attributed to the cloud contamination of the MODIS LAI, as shown in Figure 3a and Figure 3b, in the mountainous humid part of the basin , as note in Krause (2008). Also, the senescence seems to occur slightly early for Tea, as shown in Figure

3b, thereby we note a mismatch between SWAT simulated LAI and MODIS LAI. This indicate the need to further adjust the Fraction of total PHU when leaf area begins to decline (DLAI).

[revised manuscript text omitted]

---

## Author Comment (AC5) · 22 Jun 2017

**1 An improved SWAT vegetation growth module for tropical ecosystem**

Alemayehu T.1,2\*, van Griensven A.1,2 and Bauwens W.1

1Vrije Universiteit Brussel (VUB), Department of Hydrology and Hydraulic Engineering, Brussel, Belgium

2IHE Delft Institute for Water Education, Department of Water Science and Engineering, Delft, the Netherlands

\*Correspondence: t.abitew@un-ihe.org; Tel.: +31-621381512

Abstract. The Soil and Water Assessment Tool (SWAT) is a globally applied river basin eco-hydrological model in 7 a wide spectrum of studies, ranging from land use change and climate change impacts studies to research for the 8 development of best water management practices. However, SWAT has limitations in simulating the seasonal 9 growth cycles for trees and perennial vegetation in the tropics, where rainfall (via soil moisture) is the major plant 10 growth controlling factor than temperature. Our goal is to improve the vegetation growth module of the SWAT 11 model for simulating the vegetation variables such as the leaf area index (LAI) for tropical ecosystem. Therefore, we present a modified SWAT version for the tropics (SWAT-T) that uses a straightforward but robust soil moisture 12 13 index (SMI) - a quotient of the rainfall (P) and reference evapotranspiration (PET) - to initiate a new growth cycle 14 dynamically within a pre-defined period. Our results for the Mara Basin (Kenya/Tanzania) show that the SWAT-T 15 simulated LAI corresponds well with the Moderate Resolution Imaging Spectroradiometer (MODIS) LAI for ever-16 green forest, savanna grassland and shrubland, indicating that the SMI is reliable for triggering new growth cycle 17 annually. The water balance components (evapotranspiration and streamflow) simulated by the SWAT-T exhibit a 18 good agreement with a thermal-based evapotranspiration (ET-RS) estimate and observed streamflow. The SWAT-T 19 model with the proposed improved vegetation growth module for tropical ecosystem can be a robust tool for simu-20 lating the vegetation growth dynamics consistently in hydrologic model applications including land use and climate change impact studies.

**22 **1. Introduction**

The Soil and Water Assessment Tool (SWAT; Arnold et al., 1998) is a process-oriented, spatially semi-distributed 24 and time-continuous river basin model. SWAT is one of the most widely applied eco-hydrological models for simu-25 lating hydrological and biophysical processes under a range of climate and management conditions (Arnold et al., 26 2012; Bressiani et al., 2015; Gassman et al., 2014; van Griensven et al., 2012; Krysanova and White, 2015). Many 27 studies used SWAT in tropical Africa, to investigate the basin hydrology (e.g. Dessu and Melesse, 2012; Easton et 28 al., 2010; Mwangi et al., 2016; Setegn et al., 2009) as well as to study the hydrological impacts of land use change 29 (e.g. Gebremicael et al., 2013; Githui et al., 2009; Mango et al., 2011) and climate change (Mango et al., 2011; 30 Mengistu and Sorteberg, 2012; Setegn et al., 2011; Teklesadik et al., 2017). Notwithstanding the high number of 31 SWAT model applications in tropical catchments, only a few studies underscored the limitation of its plant growth module for simulating the growth cycles of trees, perennials and annuals in this region of the world (Mwangi et al.,
 2016; Strauch and Volk, 2013; Wagner et al., 2011).

It is worthwhile to note that phenological changes in vegetation affect the biophysical and hydrological processes in 35 the basin hydrology and thus play a key role in integrated hydrologic and ecosystem modeling (Jolly and Running, 36 2004; Shen et al., 2013; Strauch and Volk, 2013; Yang and Zhang, 2016; Yu et al., 2016). The Leaf Area Index 37 (LAI), a vegetation attribute commonly used in eco-hydrological modeling, strongly correlates with a vegetation 38 phenological development. Thus, an enhanced representation of the LAI dynamics can improve the predictive capa-39 bility of hydrologic models, as noted in several studies (Andersen et al., 2002; Yu et al., 2016; Zhang et al., 2009). 40 Arnold et al. (2012) also underscored the need for a realistic representation of the local and regional plant growth 41 processes to simulate reliably the water balance, the erosion, and the nutrient yields using SWAT. For instance, the 42 LAI and canopy height are needed to determine the canopy resistance and the aerodynamic resistance to subsequent-43 ly compute the potential plant transpiration in SWAT. Therefore, inconsistencies in the vegetation growth could result in uncertain ET estimates as noted in Alemayehu *et al.* (2015).

SWAT utilizes a simplified version of the Environmental Policy Impact Climate (EPIC) crop growth module to 46 simulate the phenological development of plants, based on accumulated heat units (Arnold et al., 1998; Neitsch et 47 al., 2011). SWAT uses dormancy, which is a function of daylength and latitude, to repeat the annual growth cycle 48 for trees and perennials. Admittedly, this approach is suitable for temperate region. However, Strauch and Volk 49 (2013) showed that the LAI temporal dynamics are not well represented for perennial vegetation (savanna and 50 shrubs) and evergreen forest in Brazil. Likewise, Wagner et al. (2011) reported a mismatch between the growth 51 cycle of deciduous forest in the Western Ghats (India) and the SWAT dormancy period, and they subsequently 52 shifted the dormancy period to the dry season.

Unlike temperate regions where the vegetation growth dynamics are mainly controlled by the temperature, the pri-54 mary controlling factor in tropical regions is the rainfall (i.e. the water availability) (Jolly and Running, 2004; 55 Lotsch, 2003; Pfeifer et al., 2012, 2014; Zhang, 2005). A study of Zhang et al. (2005) explored the relationship be-56 tween the rainfall seasonality and the vegetation phenology across Africa. They showed that the onset of the vegeta-57 tion green-up can be predicted using the cumulative rainfall as a criterion to indicate the season change. Jolly and 58 Running (2004) determined the timing of leaf flush in an ecosystem process simulator (BIOME-BGC) after a de-59 fined dry season in the Kalahari, using events where the daily rainfall (P) exceeded the reference evapotranspiration 60 (PET). They showed that the modeled leaf flush dates compared well with the leaf flush dates estimated from the 61 Normalized Difference Vegetation Index (NDVI), indicating the reliability of a proxy derived from P and PET to 62 pinpoint a season change of tropical ecosystems. Sacks et al. (2010) studied the relationships between crop planting 63 dates and temperature, P and PET globally, using 30-year average climatological values. They noted that in rainfall 64 limited regions the ratio of P to PET is a better proxy for the soil moisture status than is P alone. Using soil mois-65 ture index (SMI) derived from the ratio of P to PET to trigger new growth cycle annually in hydrological modeling is appealing because as the SMI can be determined a priori. On the other hand, Strauch and Volk (2013) used 66

SWAT model simulated soil moisture in the top soil layers with a certain minimum threshold after a defined dry season to indicate the start of a rainy season (SOS) and thus new vegetation growth cycle. Their results showed improvements in the SWAT simulated LAI seasonal dynamics and reproduced well the Moderate Resolution Imag- ing Spectroradiometer (MODIS) 8-day LAI. However, such approach requires calibrating the SWAT parameters for

- 71 a realistic representation of the soil water balance dynamics often using observed streamflow. Recently, Yu *et al.*
- 72 (2016) concluded uncertainty in soil moisture is significantly greater than streamflow simulations of a calibrated
- 73 hydrologic model.

The objective this study is to improve the vegetation growth module of SWAT model for trees and perennials in the

- tropics. Towards this the use of the SMI within a predefined transition months as a dynamic trigger for new vegeta-
- tion growth cycle will be explored. The modified SWAT (SWAT-T) model will be evaluated using 8-day MODIS

**78 2. Materials and methods**

**79 2.1. The study area**

The Mara River, a transboundary river shared by Kenya and Tanzania, drains an area of 13,750 km2 (Figure 1a). This river originates from the forested Mau Escarpment (about 3000 m.a.s.l.) and meander through diverse agroecosystems and subsequently crosses the Masai-Mara Game Reserve in Kenya and the Seregenti National Park in Tanzania and finally feeds the Lake Victoria. The Amala River and the Nyangores River are the only perennial tributaries draining the head water region. The Talek River and the Sand River are the two most notable seasonal rivers stemming from Loita Hills.

Rainfall varies spatially mainly due to its equatorial location and its topography. The rainfall pattern in most part of 87 the basin is bimodal, with a short rainy season (October-December) driven by convergence and southward migration 88 of the Intertropical Convergence Zone (ITCZ) and a long rainy season (March-May) driven by southeasterly trades. 89 In general, rainfall decreases from west to east across the basin while temperature increases southwards. The Mara 90 basin is endowed with significant biodiversity features through a sequence of zones from moist montane forest on 91 the escarpment through dry upland forest to scattered woodland and then the extensive savanna grasslands (Figure 92 1b). Dark volcanic origin soils are common on the escarpment and rangelands while shallow soils that drain freely 93 are found lower down. Poorly drained soils cover the plateau and the plains.

Figure 1 Location of the Mara Basin (a) and its land cover classes (b). Note the sample sites location for the major natural
 vegetation classes that are used to mask the Moderate Resolution Imaging Spectroradiometer (MODIS) Leaf Area Index
 (LAI).

**101 2.2. SWAT model description**

The SWAT (Arnold et al., 1998, 2012; Neitsch et al., 2011) is a comprehensive, process-oriented and physically- based eco-hydrological model at a river basin scale. SWAT requires specific information about weather, soil proper- ties, topography, vegetation, and land management practices occurring in the watershed to directly model physical processes associated with water movement, sediment movement, crop growth, nutrient cycling, etc. In SWAT a basin is partitioned into several sub-basins using topographic information and the sub-basins, in turn, are subdivided into several Hydrological Response Units (HRUs) with a unique combination of land use, soil and slope class. Each hydrologic processes are simulated at HRU level on a daily or sub-daily time step and aggregated into sub-basin level for routing into a river network (Neitsch et al., 2011). SWAT considers five storages: snow, canopy storage, the soil profile with up to ten layers, a shallow aquifer and a deep aquifer to calculate the water balance (Neitsch et al., 2011) using the following equation:

$$\Delta S = \sum_{i=1}^{t} \left( P - Q_{total} - ET - Losses \right)$$
(1)

where  $\Delta S$  is the change in water storage (mm) and *t* is time in days. *P*,  $Q_{total}$ , *ET* and *Losses* are the daily amounts of precipitation (mm), the total water yield (mm), the evapotranspiration (mm) and the groundwater losses (mm), respectively. The total water yield represents an aggregated sum of the surface runoff, the lateral flow and the return flow. In this study, the surface runoff is computed using the Soil Conservation Service (SCS) Curve Number (CN) method (USDA SCS, 1972).

- 117 SWAT provides three options for estimating PET: Hargreaves (Hargreaves et al., 1985), Priestley-Taylor (Priestley 118 and Taylor, 1972), and Penman-Monteith (Monteith, 1965) (Neitsch et al., 2011). The model simulates evaporation 119 from soil and plants separately as described in Ritchie (1972). The potential soil evaporation is simulated as a func-120 tion of PET and leaf area index (LAI) and further reduced with high plant water use while the actual soil water 121 evaporation is estimated by using exponential functions of soil depth and water content (Neitsch et al., 2011). 122 SWAT simulated LAI is also required to calculate the potential plant transpiration with formulations that varies 123 depending on the PET method selection (Alemayehu et al., 2015; Neitsch et al., 2011). The actual plant transpiration 124 (i.e. the plant water uptake) is reduced exponentially for soil water below filed capacity. Therefore, actual evapo-125 transpiration in SWAT refers to the sum of evaporation from the canopy, the soil as well as plant transpiration.
- In this study, we use the Penman-Monteith method (Monteith, 1965) to compute the PET for alfalfa (Neitsch et al.,2011) as:

$$PET = \frac{\Delta (H_{net} - G) + \rho_{air} c_p [e_z^o - e_z] / r_a}{\Delta + \gamma (1 + \frac{r_c}{r_a})}$$
(2)

where PET is the maximum transpiration rate (mm d-1),  $\Delta$  is the slope of the saturation vapour pressure-temperature curve (kPa °C-1),  $H_{net}$  is the net radiation (MJ m-2 d-1), G is the heat flux density to the ground (MJ m-2 d-1),  $\rho_{air}$  is the air density (kg m-3),  $C_p$  is the specific heat at constant pressure (MJ kg-1 °C-1),  $e_z^0$  is the saturation vapour pressure of air at height z (kPa),  $e_z$  is the water vapor pressure of air at height z (kPa),  $\gamma$  is the psychrometric constant 132 (kPa °C-1),  $r_c$  is the plant canopy resistance (s m-1), and  $r_a$  is the diffusion resistance of the air layer (aerodynamic 133 resistance) (s m-1). The plant growth module simulates LAI and canopy height, which are required to determine the 134 canopy and aerodynamic resistance.

**135 2.3. The vegetation growth and Leaf Area Index modeling in SWAT**

SWAT simulates the annual vegetation growth based on the simplified version of the EPIC plant growth model (Neitsch et al., 2011). The potential plant phenological development is hereby simulated on the basis of daily accumulated heat units under optimal conditions; however, the actual growth is constrained by temperature, water, nitrogen or phosphorous stress (Arnold et al., 2012; Neitsch et al., 2011).

Plant growth is primarily based on temperature and hence each plant has its own temperature requirements (i.e. minimum, maximum and optimum). The fundamental assumption in the heat unit theory is plants have a heat unit requirements that can be quantified and linked to the time of planting and maturity (Neitsch et al., 2011). The total number of heat units required for a plant to reach maturity must be provided by the user. The plant growth modeling includes simulation of the leaf area development, light interception and conversion of intercepted light into biomass assuming a plant species-specific radiation-use efficiency (Neitsch et al., 2011). The optimal leaf area development during the initial period of the growth is modeled (Neitsch et al., 2011) as:

$$fr_{LA\,\mathrm{Im}\,x} = \frac{fr_{PHU}}{fr_{PHU} + \exp(l_1 - l_2.fr_{PHU})} \tag{3}$$

where  $fr_{LAImx}$  is the fraction of the plant's maximum leaf area index corresponding to a given fraction of potential heat units for the plant, , and  $l_1$  and  $l_2$  are shape coefficients. Once the maximum leaf area index is reached, LAI will remain constant until the leaf senescence begins to exceed the leaf growth. Afterwards, the leaf senescence becomes the dominant growth process and hence the LAI follows a linear decline (Neitsch et al., 2011). However, Strauch and Volk (2013) showed the advantage of using a logistic decline curve, to avoid that the LAI drops to zero before dormancy occurs. Therefore, we adopted this change to SWAT2012 whereby the LAI during leaf senescence for trees and perennials is calculated as (Strauch and Volk, 2013):

$$LAI = \frac{LAI_{mx} - LAI_{min}}{1 + \exp(-t)}$$
(4)

with
$$t = 12(r - 0.5)$$
 and  $r = \frac{1 - fr_{PHU}}{1 - fr_{PHU,sen}}$ ,  $fr_{PHU} \ge fr_{PHU,sen}$

where the term used as exponent is a function of time and t varies from 6 to -6, LAI is the leaf area for a given day and declines using r as a decline rate, *LAImx* and *LAImin* are the maximum and minimum (i.e. during dormancy) leaf area index, respectively,  $fr_{PHU,sen}$  is the fraction of growing season (PHU) at which senescence becomes the domi- nant growth process.

- As detailed in Neitsch *et al.* (2011), the daily LAI calculation for perennials and trees are slightly different, for the latter the years of development is considered.
- 160 For perennials, the leaf on day *i* is calculated as:

$$\Delta LAI_{i} = \left(fr_{LA\operatorname{Im} x,i} - fr_{LA\operatorname{Im} x,i-1}\right)LAI_{mx}.$$

$$\left(1 - \exp(5.(LAI_{i-1} - LAI_{mx}))\right)$$
(5)

The total leaf area index, the area of green leaf per unit area of land, is calculated:

$$LAI_i = LAI_{i-1} + \Delta LAI_i \tag{6}$$

where  $\Delta LAI_i$  is the leaf area added on day *i*,  $LAI_i$  and  $LAI_{i-1}$  are the leaf area indices for day *i* and *i-1* respectively, *frLAImx,i* and *frLAImx,i-1* are the fraction of the plant's maximum leaf area index for day *i* and *i-1*,  $LAI_{mx}$  is the maximum leaf area index for the plants, *yrcur* is the age of the tree (years), and *yrfulldev* is the number of years for tree species to reach full development (years).

**166 2.4. SWAT annual vegetation growth cycle limitation for the tropics**

SWAT assumes that trees and perennial vegetation can go dormant as the daylength nears the minimum daylength 168 for the year. Dormancy, which is a function of latitude and daylength during which plants do not grow, is used to 169 repeat the growth cycle each year for trees and perennials. At the beginning of the dormant period, a fraction of the 170 biomass is converted to residue and the leaf area index is set to the minimum value (Neitsch et al., 2011). In the 171 tropics, however, plants growth dormancy is primarily controlled by precipitation (Bobée et al., 2012; Jolly and 172 Running, 2004; Lotsch, 2003; Zhang et al., 2010; Zhang, 2005) and hence the standard SWAT growth module can-173 not realistically represent the seasonal growth dynamics for trees and perennials. SWAT offers several management 174 settings for the start and the end of growing season based on either heat units (the default) or calendar date schedul-175 ing. In fact, the limitation with plant growth dynamics cannot be solved using SWAT management settings as far as the latitude and daylength dependent dormancy is activated.

**177 **2.5.** A soil moisture index-based vegetation growth cycle for the tropics**

As several studies demonstrated (Jolly and Running, 2004; Zhang, 2005; Zhang et al., 2006), the water availability in the soil profile is one of the primary governing factors of vegetation growth in tropics. Thus, we propose a soil

- 180 moisture index (SMI) to trigger new growth cycle for tropical ecosystem in SWAT model within a predefined peri-
- 181 od. The SMI is computed as:

$$SMI = \frac{\sum_{i=1}^{N} P}{\sum_{i=1}^{N} PET}$$

where P and PET denotes daily rainfall and potential evapotranspiration (mm d-1), N is the number of days of aggregation. In this study we used five days aggregated P and PET (i.e. pentad) to determine the SMI to assure sufficient
soil moisture availability to initiate new growth cycle. The SMI is somewhat similar to the Water Requirement Sat- isfaction Index (WRSI) (Verdin and Kalver 2002), which is a ratio of ET to PET.

Figure 2 presents the SMI seasonal pattern based on long-term climatological P for several gauge stations and PET 187 from Trabucco and Zomer (2009) across the Mara Basin. It is apparent from Figure 2 that the dry season (mostly 188 from June - September) shows low SMI values (less than 0.5). Additionally, these patterns resemble well the long-189 term monthly average LAI for the savanna ecosystem (the dominant cover in the mid-section of the Mara Basin). In 190 areas with a humid climate (i.e. the head water regions of the basin), the SMI values are high and the rainfall regime 191 is different, yet in the relatively drier months (January and February) the SMI is low. As shown in Figure 2, the LAI 192 and the SMI seasonal dynamics match well with approximately one month lag, indicating the reliability of the SMI 193 as a proxy for the SOS and hence to trigger the annual vegetation growth cycle. This approach enables SWAT 194 growth module not only to simulate the vegetation cycle dynamically within a predefined period, also avoids the 195 need for management setting ("plant" and "kill").

---

## Author Response (AR1)

**Point-by-point response to the Editor (Xuesong Zhang), T. Brussée and two anonymous referee comments and suggestions**

**General authors note: We very much appreciated all the comments from the editor and the three reviewers. We think the comments help to substantially improve the clarity and the structure of the revised manuscript.**

**Important changes: Improved methodology section and restructured Results and Discussion part. Additionally, we have added two new Figures (Figure 8). Overall, we have further improved the language throughout the revised manuscript.**

**Below we provided to a point-by-point response in blue (italics).**
* * *
**Response to the Editor**

Comments to the Author:
Overall, I think the manuscript is adding new contributions to improved representation of growth cycle and LAI in the SWAT model.

I think the authors have tried to revise the manuscript according to the comments of the three referees. Many places throughout the manuscript have been improved and clarified. Given the revised form of the manuscript, I recommend the authors further improve the language and make further clarifications. Specifically, I have several comments that I hope the authors will address before re-submission:

*Response: We thank the editor for comments and encouraging assessment of the revised manuscript. We further sharpened the revised manuscript language for better readability and clarity. The point-by-point responses are provided below in italics and the changes in the revised manuscript are in blue.*

(1) The current title is "An improved SWAT vegetation growth module for tropical ecosystem". As the study only tested SWAT-T for evergreen forest, Tea, and two types of rangeland/grassland in one river basin, it would be better to make the tile more specific. Please think about "An improved SWAT vegetation growth module and its evaluation for three tropical ecosystems".

*Response: The title of the revised manuscript is modified accordingly and it reads as "An improved SWAT vegetation growth module and its evaluation for four tropical ecosystems"*

(2) I also agree with T. Brussée and referee #2 that there are still grammar errors and confusing narrative in the manuscript. For example, line 294: "…models differs"; line 11 "…for tropical ecosystem"; line 13 "…reference evapotranspiration (PET)"; line 18: "… with a thermal-based evapotranspiration (ET-RS) estimate and observed streamflow". It would be helpful if the manuscript will be further proofread.

*Response: We further improved the language throughout and we think the revised manuscript is more clear and readable in its current form.*

(3) Please consider change the last sentence of the abstract to "The SWAT-T 19 model with the proposed improved vegetation growth module for tropical ecosystem can be a robust tool for simulating vegetation growth dynamics in hydrologic model applications in tropic regions".

*Response: The last sentence of the abstract is updated accordingly that reads as "…The SWAT-T model, with the proposed vegetation growth module for tropical ecosystems, can be a robust tool for simulating the vegetation growth dynamics in hydrologic models in tropical regions."*

(4) Pleased double check how you addressed comment 1.226-27 from referee 2.

*Response: We rephrased the sentence for better clarity and it reads as "To remove the biases in SWAT computed $ET_r$ compared to the observation-based monthly average (1950-2000) $ET_r$ from Trabucco and Zomer (2009), the GLDAS solar radiation were adjusted relatively per month and per sub-basin. "*

(5) In caption of Figure 8: change "seasonal LAI" to "monthly LAI".

*Response: Changed accordingly.*

(6) The authors' response to referee #1's comment "line 175-181…" is not clear. Please specify if "kill operation" is used in the improved growth module.

*Response: The response to the referee is further clarified and it reads as "We have moved the literature reviews on the role of soil moisture availability for vegetation growth dynamics in the tropics to the introduction section in the revised manuscript (line 53-71). ".*

*There is no any management setting used in the improved growth module. This is specified in the revised manuscript (line 332-333 and in the caption of Figure 4 and Figure 5).*

(7) Did the authors make any clarification according to T. Brussée's comment "Line 97 'poorly drained soils cover the plateau' …"?

*Response: We further clarified the soil type distribution in the study area in the revised manuscript (line 89-91).*
* * *
**Response to T. Brussée**

*The authors appreciate very much T. Brussée for his time and forwarding interesting reflections and questions.*

General reflection:

I liked reading your paper very much, I think your assumptions given for your theory on LAI behaviour are valid and an added value to the SWAT model. They align with my findings on the weaknesses of SWAT in modelling in the tropics.

*Response: We are glad to hear this encouraging remark.*

What is your opinion on how SWAT calculates the maximum transpiration using the Hargreaves or P-T method, at times when the LAI > 3? Is it realistic that the maximum transpiration remains equal when the LAI is 3 and when the LAI is 4? I refer to the formula for calculating maximum transpiration as you also give in eq.5 of Alemayehu et al (2015):

$E_t = E'_0$ if LAI > 3.0 (m2/m2)

*Response: These are interesting questions. In SWAT when either Hargreaves (Hargreaves et al., 1985) or Priestley-Taylor (Priestley and Taylor, 1972) reference evapotranspiration (PET) method is used, unlike the Penman-Monteith (Monteith, 1965) method, the potential transpiration is computed empirically as function of adjusted daily $ET_r$ and LAI. For land covers with LAI above 3, the daily potential transpiration is equal under similar atmospheric condition. However, the actual plant transpiration is further limited by the actual soil water availability. As far as our experience, if the LAI is represented realistically these $ET_r$ methods can reliably estimate the maximum plant water demand under optimal condition on a given day.*

I found it very interesting that you initiate the LAI by using the ratio of P to PET, instead of soil moisture and reading the argument this is a good alteration to what Strauch and Volk (2013) did in their research. Also nice that SWAT-T can better account for climatic variations.

*Response: We are happy that you have pointed out the main part of the manuscript.*

Line 10-11 *"where the major plant growth controlling factor is the rainfall (via soil moisture) rather than temperature."* – it seems as if you mean to say that temperature is the preferred plant growth controlling factor, maybe you can cut the sentences up into two sentences: **1)** However, SWAT has limitations in simulating the seasonal growth cycles for trees and perennial vegetation in tropics. **2)** In the tropics plant growth is mainly controlled by rainfall (via soil moisture), whereas in SWAT plant growth is temperature controlled.

*Response: We have rephrased this sentence for better readability in the revised manuscript (lines 8-11) and reads as "However, SWAT has limitations in simulating the seasonal growth cycles for trees and perennial vegetation in the tropics, where rainfall rather than temperature is the dominant plant growth controlling factor."*

Line 57 "Normalized Vegetation Index (NDVI)" – shouldn't this be: "Normalized Vegetation Difference Index (NDVI)"?

*Response: Well spotted. Modified accordingly.*

Line 97 "poorly drained soils cover the plateau" I was wondering what you meant with "plateau". I guess the Mau escarpment?

*Response: No, that is referring the landscape in the lower section of the basin. We further improved soil type distribution in the study area for better clarity in the revised manuscript (lines 89-91) that reads as "The upper forested basin is dominated by well drained volcanic origin soils, while the middle and the lower part of the basin is dominated by poorly drained soil types with high clay content."*

Line 192-193 Does this mean that there can be set two starts of the rainfall seasons (SOS) for a bimodal rainfall regime? : So there is an end of the dry season [SOS1] and a beginning of the rainy season [SOS2] for the long rains (for the Mara for example) and there is another end of the dry season [SOS3] and a beginning of the rainy season for the short rains [SOS4] ?

*Response: No, there is one phenological cycle per year regardless of the rainfall pattern. Therefore, we need to predefine only two months for the transition months.*

Line 196 "*pentad ratio*" – I had never heard of this, I don't know whether it is a common term (maybe it's because I am a non-native speaker of English), but to make it easier to read you might also just say " five day ratio".

*Response: Pentad is a conventional term that refers to five days aggregate. We have now included for further clarity '…5 days…' in the revised manuscript (line 181).*

Line 302-306 This trial-and-error process was it done manually or with for example SWAT-CUP? And if so, did you have some sort of a steps that you followed in this procedure? I am curious because personal experience taught me that altering these five LAI parameters in SWAT-CUP or directly in the input .mgt or .plant files, could give pretty random outcomes in terms of LAI curves or PET, and results of altering multiple LAI parameters at the same time are difficult to predict

*Response: We did the calibration manually. It is true that manual calibration of distributed hydrological models like SWAT with many parameters is not a trivial task. SWAT plant parameters related to the vegetation growth dynamics were calibrated by comparing with 8-day MODIS LAI timeseries. It is not uncommon to come up with good model performance for the wrong reason and therefore, we used our expert knowledge to guide the trial and error process while adjusting the parameters.*

Line 308-309 Do you know why Kilonzo (2014) [Penmann-Monteith] and Mwangi (2016)[Hargreaves] recommend using a minimum LAI for FRSE of respectively 3 and 4? For Mwangi this worked very well. For tropical forest in Brasil this is reasonable, but looking at the mean annual LAI in the FRSE of the Mau escarpment of 2.6 this seems too high of an estimate. I also saw in **figure 7** That you had set the minimum LAI for SWAT-T to about 2.2 and maximum LAI to 5. Was this just for the purpose of giving an example at the same setting as the default or was this also the value as used in your simulations?

*Response: Both Kilonzo (2014) and Mwangi et al. (2016) stated that they used literature values for forest LAI. In this study, we used 8-day filtered LAI from 2002-2009 to determine the minimum LAI. Indeed Mwangi et al. (2016) improved SWAT simulated LAI by reducing the Fraction heat unit ($FR_{PHU}$) to very low values (0.001) and setting the minimum LAI to 3. These changes improved the LAI simulation at the start of new simulation in January every year, however, the minimum LAI during the summer months due to the latitude and daylength dependent dormancy needs to march with the dry season months. Our simulation in Figure 7 was with default SWAT parameters to explore the effect of the modification on the vegetation growth module.*

Line 334-336 "We also notice the SWAT-T simulated potential transpiration is consistent while changing the PET method to Hargreaves method in SWAT (results not shown here)." Interesting! Is this also the case for the PET at times where LAI > 3 ? Did you also try using the Priestley-Taylor (P-T)? Personal modelling experience in the region taught me that the annual PET using the P-T method is often lower then when using the Hargreaves or P-M, thus giving a lower AET, thus implicating that there is more water in the catchment system to "play with" as in comparison to the P-M or Hargreaves.

*Response: We appreciate the fact that you are sharing your experience. The potential transpiration calculation requires simulated actual LAI and canopy height to compute the canopy resistance and the aerodynamic resistance while using P-M method. The P-T and the HG methods require only simulated LAI to compute the daily potential transpiration. Therefore, realistic representation of the temporal LAI dynamics in SWAT is crucial for reliable potential transpiration. We have shown the inconsistencies in the simulated potential transpiration (i.e. considerable zero values) due to unrealistic LAI simulations using the PM (i.e. data intensive) and the HG (i.e. less data intensive)*

*methods. We believe that if the seasonal dynamics of the LAI is represented well the potential transpiration computed by one of the PET methods would work well. In fact, the underestimation with the actual ET would affect the water balance. We have reflected these points in the revised manuscript.*
* * *
**Response to Anonymous referee #1**

*The authors appreciate the anonymous referee for the constructive and valuable comments.*

In this manuscript, the authors modify the phenology algorithm of SWAT in simulating tropical vegetation. This work provided some interesting discussion of limitations in SWAT plant module. I have several major concerns about this work. First, writing of this work should be further improved to make it publishable. Second, organization of the introduction and method sections should be changed following requirement of a scholarly journal.

*Response: We agree on the need for further improvement on the language and organization. Therefore, we have substantially improved the language and the organization in the revised manuscript. The new changes are marked blue throughout in the revised manuscript.*

line 47, what does this mean? do you mean that SWAT could not represent changes deciduous forest?

*Response: No, we are referring to the mismatch between the SWAT dormancy period and the dry season, where dormancy is a function of water and temperature stress. According to Wagner et al. (2011) the SWAT simulated LAI for deciduous forest in India is not realistic because the dormancy in SWAT is related to daylength and latitude. As a result the authors shifted the dormancy period in SWAT to the dry season months and hence improved the LAI simulation. As pointed by referee #2, we have further improved this text in the revised manuscript (lines 50-52).*

line 55, extra space.

*Response: removed.*

line 58, for -> of. To simulate

*Response: Modified accordingly.*

line 59, a rainy season

*Response: Updated.*

line 67-68 in this sentence should be moved to the method section

*Response: We agree with this suggestion. The use of soil moisture index (SMI) to trigger growth in tropics is described in the methodology section of the modified manuscript (page 8, section 2.5).*

line 71-81, I expected to see objectives of this work, but the authors are describing their methodology, which should be moved to the next section (method)

*Response: We agree the objectives were not clearly stated. The objectives of the research are presented clearly in the revised manuscript (lines 72-74) that reads as "...The objective this study is to improve the vegetation growth module of SWAT model for trees and perennials in the tropics. Towards this the use of the SMI within a predefined transition months as a dynamic trigger for new vegetation growth cycle will be explored. ..."*

line 92, a long rainy season

*Response: Updated.*

line 93, across what?

*Response: across the basin. We have added this in the revised manuscript (line 87).*

section 2.2.1 was copied from the SWAT manual. I suggest to condense this part significantly, or move it to a supplementary section

*Response: This is a brief summary of the vegetation growth modelling in SWAT based on the manual (Neitsch et al., 2011) and published literatures. As suggested, we have considerably reduced the summary in the revised manuscript (lines 135-164).*

line 172-173, this is not correct. Kill and dormancy are totally different. If some one use this to regulate phenology, they must have made mistakes.

*Response: We agree that "dormancy" and "kill" are different concepts in SWAT. Dormancy is a function of daylength and latitude, and dormancy starts at the shortest day of the year for trees and perennials in tropics. Whereas, the kill operation in SWAT stops plant growth at a specified time using either date calendar or fractional potential heat unit (Neitsch et al. 2011). The default management setting in SWAT has planting/beginning of the growing season and kill/end of the growing season at 0.15 and 1.2 $FR_{PHU}$, respectively for trees and perennials. The management setting are important for several agroforestery operations.*

line 175-181, these discussions should be presented in your Introduction, or the Discussion section

*Response:  We have moved the literature reviews on the role of soil moisture availability for vegetation growth dynamics in the tropics to the introduction section of the revised manuscript (lines 53-71).*

line 183, what are the data source of P and PET?

*Response: The P is based on historical local gauge observations in and around the basin. PET is observation based global data from Trabscaou and Zomer (2009). We have added the sources in the revised manuscript (line 187). Note that also in the revised the potential evapotranspiration (PET) term is changed to reference evapotranspiration ($ET_r$).*

line 192-193, remove these two sentences

*Response: We disagree with the referee. Because in larger basins there is a variation in climate across the the sub-basins (i.e. watersheds).  As a result, there could be difference in seasonality of rainfall and start of the growing season across the watersheds.*

line 201-207, do you have any reference to support your rules?

*Response: the fundamental rule for the start of new growing season is the SMI (i.e. P/ET$_r$). When the SMI exceeds a user defined threshold a new growth cycle is triggered within a predefined period annually. This concept is somewhat similar to the Water Requirement Satisfaction Index (WRSI), which is a ratio of actual evapotranspiration to ET$_r$ (Verdin and Kalver 2002). The threshold for the SMI to trigger new growing season is set to 0.5, meaning the rainfall satisfies 50% of the atmospheric water demand (ET$_r$). Even though this threshold could vary from place to place, growing season in general is defined as the time when average P is greater than half of the average ET$_r$ (Mcnally et al. 2015). These references are included in the revised manuscript.*

line 238, a... site?

*Response: The 8-day MODIS LAI has 1000 m spatial resolution and therefore to avoid and/or reduce land –cover class mix during aggregation of pixels, we used selected sample sites (shown in Figure 1b) to mask the MODIS LAI for each representative land-cover classes. We think such approaches improve the reliability of the LAI estimates for each representative land-cover types.*

line 242, why lai of 1.5 is removed?

*This is based on the long term LAI timeseries during the rainy season (i.e. the peak growing season), whereby the LAI values are above 2.0 $m^2/m^2$. Therefore, LAI values during the rainy season below 1.5 are replaced with interpolated values. We have provided this information in the revised manuscript (line 234).*

line 245, what break?

*Response: This break is referring to unrealistically low LAI values due to noise and cloud contamination. Sudden break in LAI could also happen due to anthropogenic effects (land use change, fire, etc.).*

line 253-254, I donot understand how the LAI patterns match precipitation.

*Response: For the dominant part of the basin, the long rainy season is from March to May with a peak in April. Also, the basin gets short rain from October to December. On the other hand, we note the LAI seasonal pattern, whereby the lower LAI values are observed in the dry months (July to Sept).*

*This is supported by a correlation of 0.66 seasonally in the humid part. This is discussed further in the seasonality of LAI and its association with rainfall in the revised manuscript (lines 403-418).*

line 259-260, awkward expression. consider to improve

*Response: We have improved this in the revised manuscript (lines 256-257) that reads as "ET is one of the major components of a basin water balance that is influenced by the seasonal vegetation growth cycle."*

line 270, change the term 'flow' to stream flow or river discharge through out the manuscript

*Response: Modified accordingly.*

line 272-273, remove the second period

*Response: Removed.*

line 285, seasonality of what?

*Response: LAI is missing here. We have included this in the revised manuscript (line 403).*

line 290-291, I am not convinced that lai reflects changes in rainfall. you need to provide some statistical information here. And what about the correlation between temperature and lai?

*Response: There is a fair association between seasonal rainfall and LAI in the humid part with correlation up to 0.66. The correlation (0.81) is even stronger for the lower part of the basin due to the clear seasonality of rainfall, albeit with one month lag. Since temperature is not a limiting factor on vegetation growth in tropics, we did not consider that in our analysis.*

line 306-308. very confusing

*Response: The minimum LAI for a land cover could vary inter-annually depending on the climatic condition. However, this minimum LAI for each land cover need to be provided in SWAT plant database, meaning the minimum LAI during dormancy for a specific land cover does not change inter-annually and hence it is constant.*

line 311 figure 5 only show two land covers. What about the other two?

*Response: The plot for Tea and RNGB are excluded since the observed patterns are similar to FRSE and RNGE, respectively.*

line 316. I do not see seasonal variation from figure 5

*Response: Figure 5 depicts the seasonal pattern. The seasonal variation is depicted as the range between the maximum and minimum LAI values. In this regard note that the range in LAI that is normalized with the annual average MODIS LAI is more considerable (up to 82%) for grass (RNGE) and shrub cover (RNGB) types.*

line 325, in October

*Response: Updated accordingly.*

line 328, was, first

*Response: Modified.*

 line 329, second

*Response: Modified.*

line 330, a 8-day scale

*Response: updated accordingly.*

line 339-341. do not understand what does this mean

*Response: We have further sharpened the language in the revised manuscript (lines 328-332) that reads as "Alternatively, all the management setting can be removed ("No mgt") and vegetation is growing since the start of the simulation. It is worthwhile noting the low LAI values during and following the rainy months (i.e. March -May), suggesting unrealistic growth cycle simulation. Additionally, regardless of the management setting, the vegetation growth cycle resets annually on 28th June due to dormancy."*

line 344, consider to revise

*Response: We have modified this in the revised manuscript (lines 313-316).*

line 351-353, not clear. consider to revise

*Response: We have revised this in the modified manuscript (lines 352-356)*

line 361, I do not see improvement in runoff

*Response: we did show improvements in simulated LAI but we highlighted the SWAT-T is able to reproduce the observed streamflow.*

line 387-389, remove this sentence

*Response: We have revised this paragraph.*

line 396-406. It is surprising that the authors did not evaluate SWAT ET simulations

*Response: We in fact evaluated the SWAT-T simulated ET against ET-RS at 8-day. Since SWAT is not a fully distributed and gridded model, we have not done a pixel level spatial evaluation. Nevertheless, we have depicted the agreement between simulated LAI and ET qualitatively using one rainy month and one dry month from year 2002 at HRU level. Also, we have shown the temporal dynamics of ET using 8 years average monthly ET. However, an ongoing research is investigating on evaluating SWAT simulated ET using spatially distributed remote sensing-based ET.*

line 412. Is this a sentence?

*Response: We have updated this sentence in the revised manuscript (lines 477-479).*
* * *
**Response to Anonymous referee #2**

*The authors appreciate the anonymous referee for the constructive and valuable comments. The detailed comments helped to improve the revised manuscript substantially.*

The authors develop a new vegetation growth module for tropical ecosystems in SWAT. In particular, they use a soil moisture index to initiate a new growing cycle within two pre-defined months. They evaluate the growth module with regard to LAI, ET, and river discharge with satisfactory results. The topic is of current scientific interest, as several authors have previously outlined that the default vegetation growth for e.g. forests in SWAT is not applicable in the tropics. The manuscript is mostly well prepared. However, the paper would benefit a lot if it was more structured according to the evaluation of the vegetation growth module. In particular, this applies to the results and discussion part. This focus should be set very clearly in a revised version. Moreover, some parts of the manuscript require further, more detailed, or more precise information. The provided comments should be addressed before accepting this manuscript for publication.

*Response: The referee summarized succinctly our work and the authors very much appreciate that. We have addressed properly all the comments and suggestions in the revised manuscript and improved the manuscript organization. Below we provided our point-by-point response (blue).*

General comments:

1) The authors should take a decision on the aim of the manuscript. Do they aim at providing an improved plant growth module for SWAT in the tropics? Or do they aim at showing their adjustment of the model to a specific catchment? Currently, the title reads like the first is the case. However, in many parts the paper reads like the latter is the case. For the first aim the authors need a stronger focus on evaluation of the plant growth model and they need to program the module as flexible and transferable as possible. Right now there are parts that are not immediately transferable to other catchments. The authors are encouraged to sharpen the paper with regard to the first aim. However, if this was not their aim they may also go for the second aim and adjust the title accordingly.

*Response: This is an interesting point, thank you! The main goal of this manuscript is to present and demonstrate a methodology on an improved SWAT vegetation growth module for tropical condition. The modified growth module can be applied anywhere in the tropics and a user can also add region specific information such as the transition months ($SOS_1$ and $SOS_2$), the number of days for rainfall (P) and reference evapotranspiration ($ET_r$) aggregation to compute the soil moisture index ($SMI = P/ET_r$) and the minimum SMI threshold for triggering new growth cycle within a predefined period. We therefore have sharpened our revised manuscript in this regard.*

2) Clear separation of calibration and validation period is required. This also applies to the calibration of plant parameters. 3) A clear calibration and validation strategy regarding the different parameters used for calibration and validation is needed. E.g. right now, it is not clear which parameter was calibrated first and why?

*Response: For the sake of convenience we combined the responses for comments 2 and 3. As noted by the referee, the calibration and evaluation approach was not stated clearly in the manuscript. The revised manuscript has now included a dedicated section (section 2.8.2 lines 289-303) that elaborates on the calibration and evaluation approach.*

*"...The main purpose of this study is to explore the potential of the SMI to trigger a new vegetation growth cycle for tropical ecosystems. To evaluate the effect of the modification on the SWAT vegetation growth module, we initially inter-compared simulated LAI from the modified (i.e. SWAT-T) and the standard plant growth module with varying management settings. This analysis involved uncalibrated simulations with the default SWAT model parameters, whereby the models thus only differ regarding the way the vegetation growth is simulated and the management settings. It is worth noting that the aim of these simulations is mainly to expose the inconsistencies in the vegetation growth module structure of the original SWAT model. Afterwards, we calibrated the parameters related to the simulation of the LAI, the ET and the streamflow by trial-and-error and expert knowledge for the SWAT-T model. Firstly, the SWAT parameters that control the shape, the magnitude and the temporal dynamics of LAI were adjusted to reproduce the 8-day MODIS LAI for each land cover class. Then, we adjusted the parameters that mainly control the streamflow and ET simulation, simultaneously using the daily observed streamflow and the 8-day ET-RS. One may put forward that the manual adjustment may not be as robust as an automatic calibration as the later explores a larger parameters space. However, the manual calibration is believed to be apt to illustrate the impact of the modification of the vegetation growth cycle and its effect on the water balance components. The SWAT-T model calibration and validation was done for 2002-2005 and 2006-2009, respectively."*

4) LAI is prescribed based on satellite data (l.256). Why is this necessary? This makes the model less flexible and non-transferable to other catchments without following a similar approach.

*Response: The results based on LAI prescription is not shown in the manuscript and therefore, we have removed the sentence in the revised manuscript.*

5) The third chapter "results and discussion" is sometimes hard to understand as discussion of model parameters and results is mixed with model validation. I strongly suggest reworking the structure and separating the "results" from the "discussion" part.

*Response: We accepted this suggestion. As a result, we substantially restructured the "results and discussion"part in the revised manuscript (lines 310-483). Briefly, we have presented uncalibrated LAI simulation results from SWAT growth module with and without modification. The purpose of this section is to highlight the limitations of LAI simulation with the existing SWAT vegetation growth module and the added value of the new SMI based modifications. Afterwards, we have presented calibration and evaluation results for LAI, ET and streamflow are presented back-to back using the calibrated SWAT-T model.*

6) Why is it necessary to prescribe the two month in which the growing season starts?

*Response: This is interesting question, thank you! The two months are assumed to represent the transition from the end of the dry season to the beginning of the rainy season. These months are determined based the climatological rainfall (P) and reference evapotranspiration ($ET_r$) data. The main purpose of these months is to avoid false starts during the dry season short rainfall episodes. These months are in fact defined a priori and varies geographically depending on the climate.*

7) It would be very good, if you could validate the modeled begin of the growing season using independent data. Is there any data that you have available to do this?

*Response: In fact, the SOS dates are mainly controlled by the SMI variations and the effect of setting the transition months a priori is rather minimal given the season change is not immediately occurring with the start of the first transition month (.i.e. $SOS_1$). These dates can be verified using field data or with SOS dates extracted from remote sensing-based NDVI timeseries. We think it is sufficient to show the simulated inter-annual dynamics of the SOS dates since our study area is a typical data scarce basin and hence such detailed verification would be more interesting in a basins with better forcing data. Therefore, we acknowledge the need for further research in this regard. Yet, we speculate the SOS dates derived using SMI would be reasonable compared to SOS dates derived based on MODIS NDVI since the calibrated SWAT-T model reproduced well the MODIS LAI.*

*We have added in the revised manuscript the need of verification of the SOS dates (lines 426-427) that reads as".....Yet, we acknowledge the need for further verification studies in basins with sufficient forcing data and field measurements."*

Line specific comments:

l.7: SWAT is a hydrologic model. The term "simulator" is not very common for SWAT in the literature. Suggest to replace this by "model" in the whole manuscript.

*Response: Modified accordingly throughout the revised manuscript.*

l.13: "uses of a simple..." Please improve the language.

*Response: We have improved the language in the revised manuscript that reads as (lines 12-13) ".....we present a modified SWAT version for the tropics (SWAT-T) that uses a straightforward but robust soil moisture index (SMI)…".*

l.15: Would be good to include information here, how the dry season is defined.

*Response: Given the word limit in the abstract, we could not include extra information on how the transition months are defined. However, we have further elaborated the rationale on how to determine the transition months ($SOS_1$ and $SOS_2$) using long-term climatological P and $ET_r$ data in the revised manuscript (line 170-206). In short, the dry season is defined based on climatological P and $ET_r$ data, whereby the $ET_r$ is considerably higher than the P (shown in Figure 2 page 8). Even though the threshold could vary from place to place, growing season in general is defined as the time when average P is greater than half of the average $ET_r$ (Mcnally et al. 2015).*

l.18: "flow" – The authors probably refer to stream flow. Should be more precise throughout the manuscript.

*Response: Updated accordingly throughout in the revised manuscript.*

l.19: Please include information, which RS-ET was used.

*Response: We have slightly modified the text to reflect the type of ET source in the revised manuscript (lines 17-18)".... a remote sensing-based evapotranspiration (ET-RS) estimate……".  Nevertheless, We have provided a brief description in the data in section 2.5 (lines 254-263) about the ET-RS data based on Alemayehu, T., Griensven, A. van, Senay, G. B. and Bauwens, W.: Evapotranspiration Mapping in a Heterogeneous Landscape Using Remote Sensing and Global Weather Datasets: Application to the Mara Basin, East Africa, Remote Sens., 9(4), 390, doi:10.3390/rs9040390, 2017.*

l.20: "could be: : :" Please be more precise. In which situations is it useful?

*Response: We have updated this in the revised manuscript with information about the applicability of the tool (lines 19-20) that reads as "The SWAT-T model, with the proposed vegetation growth module for tropical ecosystems, can be a robust tool for simulating the vegetation growth dynamics in hydrologic models in tropical regions."*

l.44: Please be more precise, i.e. "dormancy, which is defined as a function of daylength and latitude".

*Response: Thank you. We have updated this in the revised manuscript.*

l.47: As I read it, they do not report a shift, but shifted the dormancy period to a prescribed dry season (see p.1786). Please improve the statement.

*Response: Thank you for spotting this. We have corrected this in the revised manuscript (lines 50-52) that reads "Likewise, Wagner et al. (2011) reported a mismatch between the growth cycle of deciduous forest in the Western Ghats (India) and the SWAT dormancy period, and they subsequently shifted the dormancy period to the dry season. "*

l.49-55: You are reviewing tropical regions. However the Kalahari has a subtropical climate. Please improve.

*Response: We disagree on this comment with the referee. Jolly and Running (2004) used two site Maun and Tshane sites, respectively located at -19.93$^0$ and 24.17$^0$ latitude to evaluate the BIOME-BGC simulated phenological development. The authors reported tropical climate for the study area (P.307 reference in the manuscript).*

l.73: "phonological"

*Response: Corrected in the revised manuscript.*

l.67-77: These lines include a lot of information on methodology. Please shift the methodological parts to the methodology section.

*Response: We agree with this suggestion. We have moved the description of the SMI as a trigger to new growth cycle in the tropics to the methodology section in the modified manuscript.*

l.103: "SWAT uses a GIS based interface". Not precise. You can use GIS to prepare input files for SWAT. Please improve.

*Response: We have rephrased this in the revised manuscript.*

l.126 following: Please add citations for formulas.

*Response: We have added Neitsch et al. ( 2011) in the revised manuscript.*

l.128: Grammar.

*Response: We have corrected this in the revised manuscript.*

l.134: "endo"

*Response: Thank you for spotting this. We have corrected this.*

l.170-173: This passage is not quiet to the point. SWAT does not offer heat unit scheduling to solve the issue of plant growth in the tropics. In fact, both scheduling options will not help, as long as the temperature dependant dormancy period is still activated. Please improve.

*Response: We agree with the referee suggestion. We have further improved the discussion with SWAT management setting and the vegetation growth cycle in the revised manuscript (lines 313-345).*

l.190-197: How are SOS1 and SOS2 defined? By a threshold, or by the increase of the SMI? Are they set by the user? If they are set by the user, the model is not as flexible – is this necessary? It should be highlighted in the other parts of the manuscript that the start of the growing season is not fully dynamic but triggered within a pre-defined period.

*Response: We appreciate these important questions from the referee. The transition months (i.e. $SOS_1$ and $SOS_2$) indicate the end of the dry season and the beginning of the rainy season. These months should be determined using the climatological monthly P and $ET_r$ ratio (i.e. the SMI). In principle during the dry season months the SMI values are low since the $ET_r$ exceeds the P considerably. In*

*contrast, during the rainy months the SMI values are relatively higher compared to the dry months. Mcnally et al. (2015) defined growing season as the period of time when average P is greater than half of the average $ET_r$. Therefore, the user should select the transition months guided by the climatological SMI values. We acknowledge some degree of subjectivity in fixing the months and yet, the climatological transition months from the dry to the rainy season are often known. The aim of fixing the $SOS_1$ and $SOS_2$ is to avoid false starts during the dry season due to short spell rainfall episodes. The new growth cycle is triggered dynamically when the SMI exceeded and/or equaled a user defined threshold within these pre-defined months. We have further clarified this in the revised manuscript (lines 176-206).*

l.211: Please add a reference for the DEM (also in table 1 & add time period for river discharge in table 1).

*Response. We have provided this information in the revised manuscript.*

l.217: Please add a reference for the SWAT land use codes, so that non-SWAT users can look these up.

*Response: We have provided reference in the revised manuscript.*

l.222: Abbreviation "TMPA" is not explained at first mentioning. Please improve.

*Response: Updated in the revised manuscript.*

l.226-27: Please add some (short) reasoning for this adjustment, so that the reader can understand the idea of this approach without reading the referred paper.

*Reponse: We have provided information on the adjustment of SWAT computed $ET_r$ using GLDAS weather data in the revised manuscript. The new text read as (lines 285-287) "….To remove the biases in SWAT computed $ET_r$ compared to the observation-based monthly average (1950-2000) $ET_r$ data from Trabucco and Zomer (2009), the GLDAS solar radiation were adjusted relatively per month and per sub-basin."*

l.237: Which forest biomes? What about others?

*Response: This is referring to the validation study on MOD15A2 LAI at Budongo Forest (Uganda) and Kakamega Forest (Kenya) by Kraus (2008). This study did not include other biome types.*

l.238: What do you mean by "land cover mix"? l.238: Grammar. l.238: Please add the sizes of the homogenous sites.

*Response: land cover mix is referring to the mix of different land cover classes (i.e. forest, grassland…) within 1000m grid resolution of MODIS LAI. We have further improved this part and provided more information in the revised manuscript (lines 229-234). This read as "…We selected relatively homogeneous representative sample sites (i.e. polygons) for evergreen forest (174 km$^2$), tea (123 km$^2$), savanna grassland (136 km$^2$) and shrubland (130 km$^2$) (see* **Error! Reference source not found.***b) using the Africover classes and Google Earth images. This is useful to reduce the effect of mixed LAI values from different land cover classes while averaging the coarse scale (i.e. 1 km) MODIS LAI. The MOD15A2 pixels with quality flag 0 (i.e. indicating good quality) were masked using the polygons of the sample covers "…*

l.242: Please provide reasoning for selecting the threshold value 1.5.

*Response: This is based on the long term LAI timeseries during the rainy season (i.e. the peak growing season), whereby the LAI values are above 2.0 m$^2$/m$^2$. Therefore, LAI values during the rainy season below 1.5 are replaced with interpolated values. We have provided this information in the revised manuscript (line 234).*

l.242: Are these gaps resulting from the previous masking?

*Response: The gaps are mainly due to cloud contaminations during April mostly.*

l.243-245: Sentence not clear. Please improve.

*Response: We have further improved this part in the revised manuscript (lines 234-237).*

l.266: Is this measured NDVI or remote sensing based? If it is remote sensing derived, are the two products independent from each other?

*Response: The NDVI is referring the MODIS product. The thermal-based ET from Alemayehu et al. (2017) is independent of the MODIS NDVI data. Alemayehu et al. (2017) estimated ET using mainly MODIS land surface temperature based on the Operational Simplified Surface Energy Balance (SSEBop) (Senay et al. 2013) algorithm.*

l.271-273: Please include how large the gauged headwater area is. Also, add information on when the gaps happen, e.g., at similar times in the year or mainly in one year?

*Response: We have provided this information in the revised manuscript (line 266).*

l.275: This is not precise and could be misleading. As I understand it, the SMI triggers the growing season within a predefined period of 2 months. Please improve.

*Response: Right, the SMI is used to indicate (i.e. trigger) the start of the new growing cycle within a predefined period. We have further clarified this throughout the revised manuscript.*

l.275-281: The model calibration and validation strategy is not clear. The authors use stream flow, LAI and ET. However, it is not clear which parameter is used first – or are they combined? Please improve this section.

*Response: We have addressed this properly and we provided the response for major comments 2 and 3.*

l.287: Peaks in April and August are not shown in Figure 4. Please clarify.

*Response: We agree with the referee. Even though the LAI magnitudes are relatively high in April and August, they are not considerably high compared to the rest of the rainy months. This is mainly due to ample rainfall distribution throughout the year. We therefore further improved this in the revised manuscript.*

l.291: Please add the "drier months" you are referring to in brackets.

*Response: We have modified this accordingly in the revised manuscript.*

l.285-299: Why are you showing tea in Fig. 4? It is shown but not mentioned here.

*Response: The tea and the forest are located in the humid and mountainous part of the basin. Therefore, we note comparable seasonal LAI dynamics and hence presented results for forest only. We have updated this in the revised manuscript.*

l.302: Again, this is not precise and could be misleading. As I understand it, the SMI triggers the growing season within a predefined period of 2 months. Please improve.

*Response: We have updated this accordingly in the revised manuscript.*

l.308-310: The authors use long-term MODIS LAI to parameterize the model. It would be much better if these values were derived from the calibration period, so that calibration and validation data are strictly independent. Even though, I do not expect a pronounced change in LAI values in calibration and validation period, I would recommend to only use data from the calibration period for model setup.

*Response: This suggestion seems a misunderstanding. We used the MODIS LAI timeseries from 2002-2005 for the calibration and the rest (2006-2009) for validation. Only the minimum LAI and the Potential Heat Unit (PHU) are adjusted based on long-term values and the remaining SWAT parameters related to vegetation growth are adjusted by comparing with the 8-day LAI time series. As mentioned earlier, we substantially restructured the results and discussion part in the revised manuscript.*

l.313: Not clear why the authors express the amplitude of simulated LAI as a percentage of the average annual MODIS LAI. Please clarify your validation strategy.

*Response: We think the way the manuscript structured created this confusion. As we stated already, the results and discussion has been restructured substantially. We have clarified in the revised manuscript (line 403-418) that the average seasonal pattern of LAI was computed using the calibrated SWAT-T simulation and MODIS LAI from 2002-2009. The average seasonal LAI amplitude is the difference between the peak and the trough LAI values (i.e. the range) and we normalize this value with the mean annual MODIS LAI.*

l.316: Why are simulated and remote sensing based LAI not directly compared and shown as later presented in figure 10? That is what the reader would expect at this point. A scatter plot is also useful in this context.

*Response: We agreed with the referee suggestion. We have plotted together the seasonal LAI for from MODIS and SWAT together. We have also included a pooled scatter plot of MODIS and SWAT-T simulated seasonal LAI for FRSE,Tea, RNGE and RNGB in the revised manuscript.*

[Figure]

**Figure 8 The long-term (2002-2009) average monthly LAI pooled scatter plot (left) and temporal dynamics (right). FRSE: evergreen forest; RNGE: grassland; RNGB: shrubland.**

l.320-323: This passage can be shifted to a discussion part.

*Response: We think discussing the season change dates after LAI seasonality is appropriate.*

l.323-325: This is only an effect of your modeling. Can you use this for validation? E.g. by comparing to independent data (e.g. satellite derived) on the beginning of the growing season?

*Response: Addressed in the general comment 7 response.*

l.337: But with a correct definition of ALAI_MIN it would not be 0, right? Please make clear, what you are validating. This improvement is due to satellite-based improvement of the parameter ALAI_MIN and not because of the improvement of the plant growth module.

*Response: Yes, correct ALAI_MIN value to the SWAT plant database would not solve this problem. Yet, the zero LAI values, for instance the forest (FRSE), as simulated in SWAT is not due to wrong prescription of ALAI_MIN in the plant database, rather due to the management settings. The ALAI_MIN is used only when dormancy occurs.*

l.332-341: You need to explain your different model setups in more detail. E.g. IGRO=1, will not be understandable for non-SWAT users. Moreover, please use only meaningful model parameterizations. It does not make sense to compare SWAT-T to a model that does not work properly. Many different model setups are irritating. I would suggest to compare SWAT-T to the best possible model parameterization achieved without code changes.

*Response: We agree on the fact that comparing different SWAT model LAI simulation with varying management setting and vegetation growth module is not appealing. Nevertheless, such uncalibrated comparisons of LAI simulation are important in shedding light on the inconsistencies of SWAT growth module for trees and perennials in the tropics. To clarify further, the uncalibrated LAI simulations are outputs from the same SWAT model with four different settings:*

*1) standard SWAT vegetation growth module with the default management settings based on heat unit scheduling*
*2) standard SWAT vegetation growth module with the default management settings based on calendar date scheduling*
*3) standard SWAT vegetation growth module with no management settings*
*4) with the modified vegetation growth module with no management settings*

*The first thee option, of course, do not require a code change. Note that there is no difference in the SWAT model parameters for this comparison, their differences are mainly due to the management settings and the modification in the vegetation growth module. We have further sharpened this in the revised manuscript (section 3.1).*

l.342-347: Not sure, why this paragraph is provided here. Figure 5 was already discussed before. Moreover, the comparison to an uncalibrated model is not a fair evaluation (see comment above).

*Response: The aim of this paragraph is to provide supportive information on rainfall distribution in the basin, which is the dominant controlling factor in vegetation phenology. The redundancy with Figure 5 has been improved with the new rearrangement of the results and discussion part in the revised manuscript.*

l.352: "standard SWAT" Please define this and provide a model evaluation for this setup. If this model does not work it is not useful for comparison.

*Response: The "standard SWAT" model is referring to SWAT 2012 revision 627 as stated in line 271 in the revised manuscript. As discussed in the earlier comments, the uncalibrated model comparison is to minimize the effect of model parameters in highlighting the inherent limitations with the vegetation growth cycle in the tropics.*

l.353-354: "better realism". Please improve the language.

*Response: We have improved the language in the revised manuscript.*

l.354: Did you test for significance?

*Response: No significance test but that is to show the considerable reduction in zero potential transpiration. We have changed the wording in the revised manuscript.*

l.368: Quantify where possible.

*Response: We have improved the result presentations in this part using bias and correlation values in the revised manuscript.*

l.363-369: This short paragraph presents some of the most valuable results. Please provide further details here. E.g. in which periods and why do MODIS LAI and simulated LAI not match well, as shown in Fig. 10?

*Response: As pointed by the referee, this section is indeed the important part of the manuscript. We have further provided information on the results in the revised manuscript (line 370-389).*

l.374-375: It is hard to see from Fig. 11 whether ET values match well as the lines overlap. Please add a scatter plot.

*Response: Thank you for the suggestion. We have improved the figures in the revised manuscript (shown below).*

[Figure]

*Figure 10 The comparison of remote sensing-based evapotranspiration (ET-RS) and SWAT-T simulated ET (ET-SWAT-T) aggregated per land cover classes. Note that for SWAT-T HRU level ET is aggregated per landcover. The vertical black line marks the end of the calibration period and the beginning of the validation period.*

l.384: Contradiction to the previous sentence. Do you mean "grassland" instead of "forest"?

*Response: Thank you for spotting this. This has been corrected in the revised manuscript.*

l.385: But Figure 5 shows a seasonality for both? This is contradictory to the previous sentences. Please clarify.

*Response: Thank you again for pointing this out. We have clarified this in the revised manuscript (lines 442-443).*

l.387-395: Again the structure is not clear. The authors evaluate parameter values at this point. Why is this needed and presented here? Please provide some justification or remove the paragraph.

*Response: The purpose of this paragraph is to provide information on the calibrated SWAT parameters and the higher water use (i.e. ET) by FRSE compared to the other land cover classes. We have moved up this to the first part of section 3.2.3. (Page 20, lines 432-437) in the revised manuscript.*

l.404: Sentence not clear. Please improve.

*Response: We have rephrased this in the revised manuscript.*

l.405: Please quantify the spatial variability.

*Response: Since SWAT assumes a uniform, single plant species community per land cover class, thus we do not expect substantial spatial heterogeneity. Therefore, we have provided qualitative visual information using spatial maps on LAI and ET as shown in Figure 11 (page 22).*

l.396-406: Why do you not compare the spatial distribution of simulated ET and LAI to the spatial distribution of MODIS based ET and LAI? That could be another valuable comparison that might be more useful than a presentation of modeled values.

*Response: This is a good point. However, since SWAT is not a fully distributed model and the HRUs are not square grids direct comparison will not be effective. Furthermore, getting cloud free MODIS LAI for the whole study area is not feasible. Therefore, the aim of the graphs is to show LAI and ET spatial variation (qualitatively) for one rainy month and one dry month.*

l.417-452: The conclusion should be shortened so that it only includes the most important conclusion drawn from your study.

*Response: Thank you for the suggestion. We have shortened the conclusion in the revised manuscript.*

l.432: This sentence is misleading ("default parameters"). As I understand your setup SWAT-T parameters were calibrated. Please clarify.

*Response: To demonstrate the added value of the modification in the plant growth module, the LAI simulation were compared for outputs from uncalibrated SWAT model (i.e. Default parameters) using the standard growth module with varying management settings and the modified growth module. Then the SWAT-T was calibrated and evaluated using MODIS LAI, RS-ET and observed streamflow. We noted in general that our manuscript needs improvement in structures. Therefore, the revised manuscript will address clarity issues.*

l.434: What do you mean by "potential transpiration"? Potential evapotranspiration?

*Response: SWAT computes plant transpiration and soil evaporation separately and their formulation depends on the reference evapotranspiration method. Potential transpiration is referring to the maximum transpiration rate by plants at optimal conditions, i.e. no water, temperature and nutrient stress. However, the potential evapotranspiration (i.e. the reference evapotranspiration in the revised manuscript) is defined as "the amount of water transpired by uniformly grown, 40cm height alfalfa, completely shading the ground and never short of water.*

l.435: The results with the other PET method were not shown. Please focus on what you have shown in this paper. If it is important, please include it, if not please remove.

*Response: We have excluded this from the conclusion of the revised manuscript.*

.

l.438: Misleading statement regarding SMI initiation of growing season, see also earlier comments. Please improve.

*Response: Improved.*

l.441: "conformed"

*Response: We have improved the language.*

l.447-452: This is more suitable for a discussions section. Please shorten or remove.

*Response: We removed this from the conclusion.*

l.451: "could be: : :" Please be more precise. In which situations is it useful?

*Response: Improved.*

l.451: "carbon fluxes" Not shown. Please remove.

Response: *Removed.*
* * *

[revised manuscript text omitted]